# DUAL-LEVEL DISENTANGLEMENT (DL$^2$): TASK-ADAPTIVE DISENTANGLEMENT FOR RESOLVING THE TASK-GENERATION DILEMMA

## ABSTRACT

Multimodal learning faces a task-generation dilemma: discriminative tasks require a purified, task-specific subset of semantics, whereas generative tasks demand the complete shared information, forcing a trade-off in a single model. To resolve this, we propose task-adaptive disentanglement (TADL), a paradigm that dynamically disentangles representations guided by task-specific supervised signals. We instantiate this paradigm with the dual-level disentanglement (DL$^2$) framework, which leverages contrastive signals as a practical and efficient form of weak supervision. DL$^2$ first separates modality private information from shared information (Level-1) and then adaptively decomposes the shared representation into a task-relevant component and a residual component that preserves generative integrity (Level-2). This second-level disentanglement is driven by two regularizers: a virtual modality pair method for positive pairs and a common-cause mutual information (CCMI) metric for negative pairs. Extensive experiments on multimodal clustering demonstrate that DL$^2$ achieves state-of-the-art task performance without compromising generative quality within a single model.

## 1 INTRODUCTION

Multimodal data (e.g., image-text) provide richer semantics than unimodal data, making the extraction of shared information key for cross-modal understanding and generation. Multimodal Variational Autoencoders (MVAEs) are widely used to disentangle this shared information from modality-specific data. However, most existing MVAEs treat the shared information as an indivisible whole (Bouchacourt et al. (2018); Sutter et al. (2020)), overlooking a critical point: different downstream tasks rely on distinct semantic subsets. This leads to the task-generation dilemma.

As shown in Figure 1, this assumption forces a trade-off between task performance and generative capability, limiting model versatility:

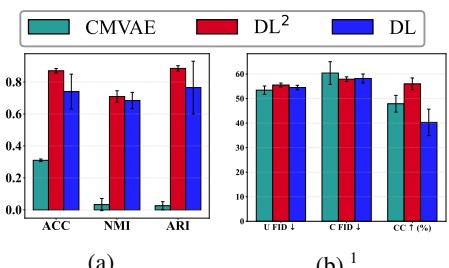

Figure 1: The task-generation dilemma on CelebA-HQ. CMVAE (Palumbo et al. (2024)) captures complete shared information, while DL extracts more purified task-relevant information. DL$^2$ achieves both pure task-relevant information and complete information preservation. See Appendix I.1 for details.

**Task Performance Demands Purity:** Discriminative tasks (e.g., clustering) require highly purified representations of task-relevant features (e.g., facial features without backgrounds). Using the full shared representation introduces noise, degrading performance and manifold structure (Wu et al. (2025)), as evidenced by CMVAE's 56% and 43% ACC decrease versus DL$^2$ and DL.

**Generation Fidelity Demands Completeness:** High-quality generation (e.g., text-to-image) requires complete shared information to preserve semantic integrity. Losing any component com-

---

[1]Part of the Abbreviations Used in This Paper: Unconditional FID (U FID), Conditional FID (C FID), Conditional Coherence (CC), Unconditional Coherence (UCC). U/C FID measure generation quality; UCC/CC measure consistency. These will be used without further explanation.

promises consistency, demonstrated by DL's 15% and 8% reduction in CC, and 4% and 3% decline in UCC relative to DL$^2$ and CMVAE.

We term this conflict the ***task-generation dilemma***. Addressing it enables unified models that achieve: i) superior task specialization through cleaner, more interpretable features; ii) controlled feature editing via task-relevant latent representations; and iii) efficient deployment by eliminating separate task-specific and generative models. A detailed analysis and experimental verification of the Task-Generation Dilemma can be found in Appendix I.1.

Resolving the task-generation dilemma requires disentangling task-relevant information. Although disentangled representation learning (DRL) provides a framework for factorizing data, its mainstream paradigms are unsuitable for this challenge. The fundamental limitation is incomplete information perception—we lack prior knowledge of the full factor set and their relationships. This leads to critical shortcomings in existing approaches: Independence-prior methods (e.g., $\beta$-VAE Higgins et al. (2017)) enforce factor independence at the cost of information loss, harming generation. Causal DRL methods (Yang et al. (2021); Shen et al. (2020)) rely on a predefined causal graph; but an incorrect graph leads to latent space misalignment, compromising both interpretability and generation quality. Therefore, a new paradigm of task-adaptive disentanglement is needed.

Resolving the task-generation dilemma requires disentangling task-relevant information. While disentangled representation learning (DRL) provides a framework for data factorization, its mainstream paradigms remain unsuitable for this challenge due to incomplete information perception—we lack prior knowledge of the full factor set and their relationships. This results in critical limitations: independence-prior methods (e.g., $\beta$-VAE Higgins et al. (2017)) enforce factor independence at the cost of information loss, harming generation; causal DRL methods (Yang et al. (2021); Shen et al. (2020)) depend on predefined causal graphs, where incorrect specifications misalign the latent space and compromise both interpretability and generation quality. These limitations necessitate a new paradigm of task-adaptive disentanglement.

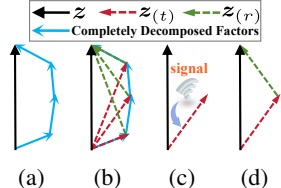

To resolve the task-generation dilemma, We propose a new paradigm: ***Task-Adaptive DisentangLement (TADL)***. Unlike prior approaches, TADL does not seek the full latent factorization depicted in Figure 2a. Instead, it decomposes the shared representation $z$ into a task-specific component $z_{(t)}$ and a residual component $z_{(r)}$. However, as Locatello et al. (2019) notes, unsupervised disentanglement requires inductive biases. Without guidance, $z$ can be decomposed infinitely many ways (Figure 2b), making task-relevant separation challenging. TADL's core principle is that task-derived signals define $z_{(t)}$, with $z_{(r)}$ adapting as its complement. This ensures holistic preservation: $z = (z_{(t)}, z_{(r)})$ (Figures 2c and 2d). The residual acts as a "reservoir" for task-irrelevant information, preventing loss (see Appendix I.1). Thus, $z_{(t)}$ is purified for downstream tasks while $z$ remains complete for generation, resolving the dilemma at its source. A successful implementation of the TADL paradigm should exhibit the following measurable properties:

Figure 2: *Task-Adaptive disentanglement **analogy**.* (a) Complete factorization of $z$. (b) Multiple decompositions are possible. (c) Task signals define $z_{(t)}$ semantics. (d) $z_{(r)}$ complements $z_{(t)}$, preserving $z = z_{(t)} + z_{(r)}$.

**Task-Adaptive Purity:** $z_{(t)}$ should contain *minimal sufficient task information*, achieving performance (e.g., ACC, NMI) comparable or superior to using $z$.

**Information Preservation:** The process must ensure no information loss in $z = (z_{(t)}, z_{(r)})$, guaranteeing high reconstruction fidelity (e.g., low FID).

**Dynamic Functional Allocation:** The information partition between $z_{(t)}$ and $z_{(r)}$ should dynamically adapt to each input and supervisory signal, distinguishing TADL from static methods.

To materialize the task-adaptive disentanglement paradigm, we instantiate the **D**ual-**L**evel **D**isentang**L**ement (DL$^2$) framework upon a MVAE. As Figure 3 shows, DL$^2$ employs a structured approach: it first separates modality-private from shared information (Level-1), then adaptively decomposes the shared representation into task-relevant ($z_{(t)}$) and residual ($z_{(r)}$) components (Level-2). This adaptive decomposition is guided by contrastive signals (CSs), a naturally available form of weak supervision. We introduce two novel regularizations: i) For positive signals (PSs), we treat a positive sample pair as a "virtual modality pair", extending the MVAE's inference framework to induce a regularization ($\mathcal{R}$PS) that identifies consistent semantics while preserving generation. ii) For negative signals (NSs), we propose Common-Cause Mutual Information (CCMI), a new metric

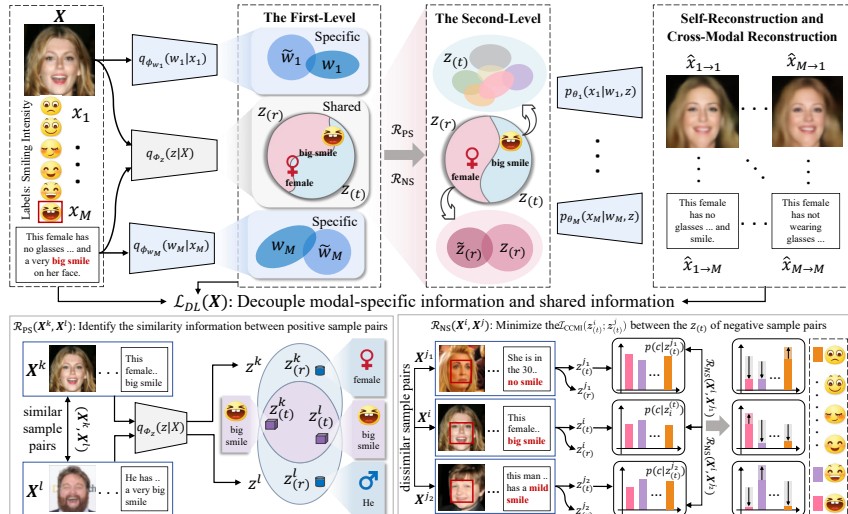

Figure 3: The illustration of $DL^2$. $\mathcal{L}_{\text{DL}}(\boldsymbol{X})$ dominates the first-level disentanglement. It can separate the shared information and the private information from each modality. $\mathcal{R}_{\text{PS}}(\boldsymbol{X}^k, \boldsymbol{X}^l)$ and $\mathcal{R}_{\text{NS}}(\boldsymbol{X}^i, \boldsymbol{X}^j)$ dominate the second-level task-adaptive disentanglement. Among them, $\mathcal{R}_{\text{PS}}(\boldsymbol{X}^k, \boldsymbol{X}^l)$ directly captures the similarity information between $(\boldsymbol{X}^k, \boldsymbol{X}^l)$. Furthermore, $\mathcal{R}_{\text{NS}}(\boldsymbol{X}^i, \boldsymbol{X}^j)$ minimizes the $\mathcal{I}_{\text{CCMI}}(\boldsymbol{z}^i_{(t)}; \boldsymbol{z}^j_{(t)})$ between negative sample pairs, which helps $\boldsymbol{z}^i_{(t)}$ and $\boldsymbol{z}^j_{(t)}$ model the dissimilarity information between $(\boldsymbol{X}^i, \boldsymbol{X}^j)$.

quantifying the probability that two samples share the same semantic cause. A dedicated regularization ($\mathcal{R}$NS) minimizes CCMI to encourage $\boldsymbol{z}_{(t)}$ to encode discriminative information between negative pairs. Integrating these into a unified objective enables $DL^2$ to achieve task-adaptive disentanglement, effectively resolving the dilemma. Our key contributions are as follows:

- **The Task-Generation Dilemma.** We identify and formalize a fundamental conflict in multimodal learning between task performance and generation fidelity.
- **The Task-Adaptive Disentanglement Paradigm.** We propose a novel paradigm that dynamically disentangles representations based on task-derived signals.
- **The $DL^2$ Framework.** We introduce a structured MVAE-based framework with a dual-level latent space that enables fine-grained semantic separation while maintaining generative integrity.
- **Novel Weakly-Supervised Mechanisms.** We develop disentanglement techniques utilizing contrastive signals, including: i) The "virtual modality pair" concept and its regularization for PSs. ii) CCMI metric and its regularization for NSs, effectively capturing discriminative semantics.

## 2 DUAL-LEVEL DISENTANGLEMENT ($DL^2$) FRAMEWORK

To resolve the task-generation dilemma, we propose a novel *task-adaptive disentanglement* paradigm, instantiated via the Dual-Level DisentangLement ($DL^2$) framework. $DL^2$ extends multimodal variational autoencoders with a structured two-level latent space for fine-grained semantic separation. We instantiate and validate $DL^2$ on multimodal clustering—an ideal testbed where contrastive signals naturally arise as pairwise constraints and clustering performance is highly sensitive to representation purity. The following sections detail $DL^2$'s primary loss function and its two regularization terms.

### 2.1 LEVEL-1 DISENTANGLEMENT: SEPARATING PRIVATE AND SHARED INFORMATION

Consider a multimodal dataset comprising $N$ samples and $M$ modalities. The dataset can be represented in two equivalent forms: $\boldsymbol{X} := \left\{ \boldsymbol{x}_m = \{\boldsymbol{x}^i_m\}^N_{i=1} \right\}^M_{m=1}$ (grouped by modality) or $\boldsymbol{X} := \left\{ \boldsymbol{X}^i = \{\boldsymbol{x}^i_m\}^M_{m=1} \right\}^N_{i=1}$ (grouped by sample). The generative process for each data point $\boldsymbol{X}^i = \{\boldsymbol{x}^i_m\}^M_{m=1}$, where $i = 1, \ldots, N$, is defined as follows. First, the cluster assignment $\boldsymbol{c}^i$

follows a discrete distribution with parameters $\boldsymbol{\pi} = \{\pi_1, \pi_2, \ldots, \pi_K\}$. The shared latent code $\boldsymbol{z}^i$ consists of two components: the task-driven component $\boldsymbol{z}^i_{(t)}$, which captures cluster-relevant information, and the residual component $\boldsymbol{z}^i_{(r)}$, which contains unrelated variations, satisfying $\boldsymbol{z}^i = (\boldsymbol{z}^i_{(t)}, \boldsymbol{z}^i_{(r)})$. The task-driven component is generated conditional on the cluster assignment as $\boldsymbol{z}^i_{(t)} \sim p(\boldsymbol{z}_{(t)} \mid \boldsymbol{c}^i)$, while the residual component $\boldsymbol{z}^i_{(r)} \sim p(\boldsymbol{z}_{(r)})$ and modality-specific latent codes $\{\boldsymbol{w}^i_m \sim p(\boldsymbol{w}_m)\}^M_{m=1}$ are sampled from their respective priors. Each data point is then generated as $\boldsymbol{X}^i = \{\boldsymbol{x}^i_m \sim p_{\theta_m}(\boldsymbol{x}^i_m \mid \boldsymbol{w}^i_m, \boldsymbol{z}^i)\}^M_{m=1}$. Typically, both priors and likelihoods are assumed to belong to specific distribution families (e.g., Gaussian, Laplacian, or mixture distributions), with the likelihood parameterized by neural network decoders. Crucially, $\boldsymbol{z}_{(t)}$, $\boldsymbol{z}_{(r)}$, and $\{\boldsymbol{w}^i_m\}^M_{m=1}$ are assumed mutually independent. Under these assumptions, the generative model is formalized as:

$$p_\Theta(\boldsymbol{X}, \boldsymbol{W}, \boldsymbol{z}, \boldsymbol{c}) = p_\pi(\boldsymbol{c}) \cdot p(\boldsymbol{z}_{(r)}) \cdot p(\boldsymbol{z}_{(t)} \mid \boldsymbol{c}) \cdot \prod^M_{m=1} p(\boldsymbol{w}_m) \cdot p_{\theta_m}(\boldsymbol{x}_m \mid \boldsymbol{w}_m, \boldsymbol{z}).$$

To achieve tractable optimization, we introduce variational encoders $\{q_{\phi_{\boldsymbol{w}_m}}(\boldsymbol{w}_m \mid \boldsymbol{x}_m)\}^M_{m=1}$ and $q_{\Phi_{\boldsymbol{z}}}(\boldsymbol{z} \mid \boldsymbol{X})$ to approximate posterior inference for each latent variable. Consistent with the generative assumption, the shared and modality-specific encoders are conditionally independent given the observed data. We further assume that given a single modality, the task-driven component $\boldsymbol{z}_{(t)}$ and residual component $\boldsymbol{z}_{(r)}$ are independent. To ensure scalability across varying numbers of modalities, we model the joint encoder for $\boldsymbol{z}$ as a mixture-of-experts (MOE):

$$
\begin{aligned}
q_{\Phi_{\boldsymbol{z}}}(\boldsymbol{z} \mid \boldsymbol{X}) = \frac{1}{M} \sum^M_{m=1} q_{\phi_{\boldsymbol{z}_m}}(\boldsymbol{z} \mid \boldsymbol{x}_m) &= \frac{1}{M} \sum^M_{m=1} q_{\phi_{\boldsymbol{z}_m}}(\boldsymbol{z}_{(t)}, \boldsymbol{z}_{(r)} \mid \boldsymbol{x}_m) \\
&= \frac{1}{M} \sum^M_{m=1} q_{\phi_{tm}}(\boldsymbol{z}_{(t)} \mid \boldsymbol{x}_m) \cdot q_{\phi_{rm}}(\boldsymbol{z}_{(r)} \mid \boldsymbol{x}_m).
\end{aligned}
\tag{1}
$$

Marginalizing the joint encoder for $\boldsymbol{z}$ yields two marginal MOE encoders: $q_{\Phi_t}(\boldsymbol{z}_{(t)} \mid \boldsymbol{X}) = \frac{1}{M} \sum^M_{m=1} q_{\phi_{tm}}(\boldsymbol{z}_{(t)} \mid \boldsymbol{x}_m)$ and $q_{\Phi_r}(\boldsymbol{z}_{(r)} \mid \boldsymbol{X}) = \frac{1}{M} \sum^M_{m=1} q_{\phi_{rm}}(\boldsymbol{z}_{(r)} \mid \boldsymbol{x}_m)$. Following Palumbo et al. (2023), we introduce auxiliary distributions $r_1(\boldsymbol{w}_1), \ldots, r_M(\boldsymbol{w}_M)$ to precisely model the shared information $\boldsymbol{z} = (\boldsymbol{z}_{(t)}, \boldsymbol{z}_{(r)})$ and modality-specific information $\boldsymbol{W}$ for each sample.

The primary objective of DL$^2$, denoted $\mathcal{L}_{\text{DL}}(\boldsymbol{X})$, forms a valid evidence lower bound for $\log p_\Theta(\boldsymbol{X})$ under Lemma 1, with proof provided in Appendix C:

$$\mathcal{L}_{\text{DL}}(\boldsymbol{X}) = \frac{1}{M} \sum^M_{m=1} \mathbb{E}_{\substack{q_{\phi_{\boldsymbol{z}_m}}(\boldsymbol{z}_{(t)}, \boldsymbol{z}_{(r)} \mid \boldsymbol{x}_m) \\ q_{\phi_{\boldsymbol{w}_m}}(\boldsymbol{w}_m \mid \boldsymbol{x}_m) \\ \{\widetilde{\boldsymbol{w}}_n \sim r_n(\boldsymbol{w}_n)\}_{n \neq m}}} \left[ J_{\pi, \Phi_t, \Phi_r, \phi_{\boldsymbol{w}_m}, \Theta}(\boldsymbol{X}, \boldsymbol{c}, \boldsymbol{z}_{(t)}, \boldsymbol{z}_{(r)}, \boldsymbol{w}_m) \right], \tag{2}$$

where

$$
\begin{aligned}
&J_{\pi, \Phi_t, \Phi_r, \phi_{w_m}, \Theta}(\boldsymbol{X}, \boldsymbol{c}, \boldsymbol{z}_{(t)}, \boldsymbol{z}_{(r)}, \boldsymbol{w}_m) \\
&= \log p_{\theta_m}(\boldsymbol{x}_m \mid \boldsymbol{z}_{(t)}, \boldsymbol{z}_{(r)}, \boldsymbol{w}_m) + \sum_{n \neq m} \log p_{\theta_n}(\boldsymbol{x}_n \mid \boldsymbol{z}_{(t)}, \boldsymbol{z}_{(r)}, \widetilde{\boldsymbol{w}}_n) \\
&\quad + \beta \log \frac{p(\boldsymbol{z}_{(r)}) p(\boldsymbol{w}_m) \sum_{\boldsymbol{c}} p(\boldsymbol{c}) p(\boldsymbol{z}_{(t)} \mid \boldsymbol{c})}{q_{\Phi_r}(\boldsymbol{z}_{(r)} \mid \boldsymbol{X}) q_{\phi_{w_m}}(\boldsymbol{w}_m \mid \boldsymbol{x}_m) q_{\Phi_t}(\boldsymbol{z}_{(t)} \mid \boldsymbol{X})}.
\end{aligned}
\tag{3}
$$

**Lemma 1.** *The primary objective $\mathcal{L}_{DL}(\boldsymbol{X})$ (Equation 2) is a valid lower bound on $\log p_\Theta(\boldsymbol{X})$.*

## 2.2 Level-2 Task-Adaptive Disentanglement via Contrastive Signals

DL$^2$ integrates information from CSs via two carefully-designed regularizations to facilitate task-adaptive disentanglement at the second level.

### 2.2.1 Regularization of Positive Signals based on Virtual Modality Pairs

Given the set of positive sample pairs (*must-link sample pairs*), which is denoted as $\mathbb{M} = \{(\boldsymbol{X}^k, \boldsymbol{X}^l) \mid \boldsymbol{c}^k = \boldsymbol{c}^l\}$. We hope to guide the model to identify and decouple $\boldsymbol{z}_{(t)}$ through their weakly supervised positive signals.

MVAEs leverage inter-modal matching relationships as weak supervision to disentangle shared and private information. We observe that *the positive signals (PSs) between sample pairs is isomorphic to multimodal matching*, both indicate semantic consistency. This inspires a novel paradigm:

treating positive sample pair as *"virtual modality pair"* to disentangle task-driven semantic subsets via extended variational inference. Specifically, we treat each sample of a positive pair as a "virtual modality". The virtual shared information in this virtual modality pair is $z_{(t)} \approx z_{(t)}^k \approx z_{(t)}^l$, while the virtual modality-specific information corresponds to $(W^k, z_{(r)}^k)$ and $(W^l, z_{(r)}^l)$. Then, for virtual modality pairs $(X^k, X^l)$, we derive $\mathcal{R}_{\mathrm{PS}}(X^k, X^l)$ to maximize $\log p(X^k, X^l)$:

$$\mathcal{R}_{\mathrm{PS}}(X^k, X^l) = \frac{1}{2M} \sum_{h \in \{k,l\}} \sum_{m=1}^{M} \mathbb{E}_{\substack{q_{\phi_{z_m}}(z_{(t)}, z_{(r)} | x_m^h) \\ q_{\phi_{w_m}}(w_m | x_m^h), \widetilde{z}_{(r)} \sim f(z_{(r)}) \\ \{\widetilde{w}_n \sim r_n(w_n)\}_{n \neq m}}} G_{\substack{\pi, \Theta, \\ \Phi_t, \Phi_r, \\ \phi_{w_m}}}(X^k, X^l, c, z, w_m), \tag{4}$$

where $z = (z_{(t)}, z_{(r)})$,

$$G_{\pi, \Theta, \Phi_t, \Phi_r, \phi_{w_m}}(X^k, X^l, c, z, w_m)$$

$$= \sum_{n=1}^{M} \log p_{\theta_n}(x_n^{\overline{h}} \mid z_{(t)}, \widetilde{z}_{(r)}, \widetilde{w}_m) + \sum_{n \neq m} \log p_{\theta_n}(x_n^h \mid z_{(t)}, z_{(r)}, \widetilde{w}_n) \tag{5}$$

$$+ \log p_{\theta_m}(x_m^h \mid z_{(t)}, z_{(r)}, w_m) + \beta \log \frac{p(z_{(r)}) p(w_m) \sum_c p(c) p(z_{(t)} \mid c)}{q_{\Phi_r}(z_{(r)} \mid X^h) q_{\phi_{w_m}}(w_m \mid x^h) q_{\Phi_t}(z_{(t)} \mid X^k, X^l)}$$

and $q_{\Phi_t}(z_{(t)} \mid X_m^k, X_m^l)$ represents the positive sample pair MOE encoder:

$$q_{\Phi_t}(z_{(t)} \mid X^k, X^l) = \frac{1}{2} \left[ q_{\Phi_t}(z_{(t)} \mid X^k) + q_{\Phi_t}(z_{(t)} \mid X^l) \right]$$

$$= \frac{1}{2} \left[ \frac{1}{M} \sum_{m=1}^{M} \left[ q_{\phi_{tm}}(z_{(t)} \mid x^k) + q_{\phi_{tm}}(z_{(t)} \mid x^l) \right] \right]. \tag{6}$$

Notably, analogous to Section 2.1, we introduce an auxiliary distribution $f(z_{(r)})$ for $z_{(r)}$ to achieve precise separation between the consistent information and specific components within positive sample pairs. The following Lemma 2, for which we provide a proof in Appendix D, proves that $\mathcal{R}_{\mathrm{PS}}(X^k, X^l)$ is a valid ELBO of $\log p_\Theta(X^k, X^l)$. Therefore, while traditional MVAE matching operates on cross-modal samples (e.g., image–text pairs), $\mathcal{R}_{\mathrm{PS}}(X^k, X^l)$ operates on virtual modality pairs. Maximizing $\mathcal{R}_{\mathrm{PS}}(X^k, X^l)$ compels the model to extract common semantics from $X^k$ and $X^l$. From an information-theoretic perspective, the first term in $\mathcal{R}_{\mathrm{PS}}(X^k, X^l)$ exerts an effect analogous to maximizing the mutual information between $z_{(t)}^h$ and $X^{\overline{h}}$. The formal proof is provided in Appendix E.

**Lemma 2.** *The similarity regularization $\mathcal{R}_{PS}(X^k, X^l)$ (Equation 4) is a valid lower bound on $\log p_\Theta(X^k, X^l)$.*

### 2.2.2 REGULARIZATION OF NEGATIVE SIGNALS BASED ON COMMON-CAUSE MUTUAL INFORMATION

Given the set of negative sample pairs (*cannot-link sample pairs*), which is denoted as $\mathbb{C} = \left\{ (X^i, X^j) \mid c^i \neq c^j \right\}$. We hope that the model can identify and decouple $z_{(t)}$ through the weakly supervised negative signals between them.

Minimizing the joint likelihood $\log p_\Theta(X^i, X^j)$ for dissimilar pairs fails to reliably identify inconsistent semantics, because the model can cheat by collapsing to low-energy states. We thus pivot to mutual information (MI): Minimizing $\mathcal{I}(z_{(t)}^i; z_{(t)}^j)$ directly severs task-semantic correlation. However, the standard mutual information exhibits limitations in the current scenario.

Consider a generative process $z_{(t)} \sim p(z_{(t)} \mid c)$ with latent variables $c \sim p(c)$. When two samples $z_{(t)}^i$ and $z_{(t)}^j$ are independently generated via:

$$c^i \sim p(c), z_{(t)}^i \sim p(z_{(t)} \mid c^i); \ c^j \sim p(c), z_{(t)}^j \sim p(z_{(t)} \mid c^j), \tag{7}$$

they are marginally independent: $p(z_{(t)}^i, z_{(t)}^j) = p(z_{(t)}^i) p(z_{(t)}^j)$. This independence renders standard mutual information $\mathcal{I}(z_{(t)}^i; z_{(t)}^j)$ identically zero, failing to capture their intrinsic relationship.

Thus, we introduce a new metric to quantify the probability that $z^i_{(t)}$ and $z^j_{(t)}$ share the same latent cause $c$ (i.e. $c^i = c^j$), despite the fact that $c^i$ and $c^j$ are sampled independently, and refer to it as common-cause mutual information (CCMI), whose form is given in Definition 1.

**Definition 1** (Common-Cause Mutual Information (CCMI)). *Assume that $z^i_{(t)}$ and $z^j_{(t)}$ are two observations, which are generated conditional on $c^i$ and $c^j$ respectively, where $c^i$ and $c^j$ are derived from the same underlying semantic space. Then, we define the CCMI between $z^i_{(t)}$ and $z^j_{(t)}$ as*

$$\mathcal{I}_{CCMI}(z^i_{(t)}; z^j_{(t)}) = \mathbb{E}_{p(z^i_{(t)})p(z^j_{(t)})} \left[ \log \frac{\mathbb{E}_c \left[ p(z^i_{(t)} \mid c) \cdot p(c \mid z^j_{(t)}) \right]}{\mathbb{E}_c \left[ p(z^i_{(t)} \mid c) \right]} \right]. \tag{8}$$

Moreover, we can deduce that $\mathcal{I}_{\text{CCMI}}(z^i_{(t)}; z^j_{(t)}) = \mathbb{E}_{p(z^i_{(t)})p(z^j_{(t)})} \log P(c^i = c^j \mid z^i_{(t)}, z^j_{(t)})$, which provides a more intuitive *semantic interpretation* for $\mathcal{I}_{\text{CCMI}}(z^i_{(t)}; z^j_{(t)})$. The derivation process is shown in Appendix F. It follows that $\mathcal{I}_{\text{CCMI}}(z^i_{(t)}; z^j_{(t)})$ exhibits symmetry and a higher $\mathcal{I}_{\text{CCMI}}(z^i_{(t)}; z^j_{(t)})$ value indicates stronger evidence that $z^i_{(t)}$ and $z^j_{(t)}$ encode the same underlying semantic $c$.

We can estimate $\mathcal{I}_{\text{CCMI}}(z^i_{(t)}; z^j_{(t)})$ using the variational encoders $p(z^i_{(t)}) \approx q_{\Phi_t}(z_{(t)} \mid X^i)$ and $p(z^j_{(t)}) \approx q_{\Phi_t}(z_{(t)} \mid X^j)$. However, it should be noted that numerical instability may arise during minimization due to the logarithm of zero. Hence, we propose maximizing

$$\mathcal{R}_{\text{NS}}(X^i, X^j) = \mathbb{E}_{\substack{z^i_{(t)} \sim q_{\Phi_t}(z_{(t)}|X^i) \\ z^j_{(t)} \sim q_{\Phi_t}(z_{(t)}|X^j)}} \log \left[ 1 - \frac{\mathbb{E}_c \left[ p(z^i_{(t)} \mid c) \cdot p(c \mid z^j_{(t)}) \right]}{\mathbb{E}_c \left[ p(z^i_{(t)} \mid c) \right]} \right] \tag{9}$$

instead, and calculate it by equation 44 in Appendix F.1.

**Remark 1** (The generality of regularization for negative sample pairs). *While instantiated with clustering, DL$^2$'s core regularization $\mathcal{R}_{NS}$ (based on CCMI) has broad applicability. For non-clustering tasks (e.g., classification/retrieval), modeling $p(z_{(t)})$ as a **mixture distribution** (e.g., Gaussian Mixture) allows this regularizer to capture discriminative semantics between sample pairs. This approach is justified by the **inherently multimodal nature** of real-world semantic spaces.*

**Remark 2** (The design rationale of asymmetrical regularizations). *While applying reversed $\mathcal{R}_{NS}$ to positive signals may seem intuitive, it captures semantic consistency less effectively than our $\mathcal{R}_{PS}$ and significantly impairs generative performance, thereby contradicting the core goal of resolving the task-generation dilemma. In contrast, $\mathcal{R}_{PS}$ is specifically designed to disentangle information within virtual modality pairs while preserving generative capability. See Appendix I.2 for analysis.*

### 2.3 UNIFIED OPTIMIZATION OBJECTIVE OF DL$^2$

DL$^2$'s final $\mathcal{L}_{\text{DL}^2}$ objective integrates the variational lower bound and dual CSs regularization, enabling end-to-end learning of dual-level disentanglement:

$$\mathcal{L}_{\text{DL}^2}(X, \mathbb{M}, \mathbb{C}) = \mathcal{L}_{\text{DL}}(X) + \frac{\lambda}{|\mathbb{M}|} \sum_{(X^k, X^l) \in \mathbb{M}} \mathcal{R}_{\text{PS}}(X^k, X^l) + \frac{\gamma}{|\mathbb{C}|} \sum_{(X^i, X^j) \in \mathbb{C}} \mathcal{R}_{\text{NS}}(X^i, X^j). \tag{10}$$

Among them, $\lambda$ and $\gamma$ represent the hyperparameters of balanced regularization $\mathcal{R}_{\text{PS}}$ and $\mathcal{R}_{\text{NS}}$, respectively. $\mathcal{L}_{\text{DL}}(X)$ dominates the first-level disentanglement. $\mathcal{R}_{\text{PS}}(X^k, X^l)$ and $\mathcal{R}_{\text{NS}}(X^i, X^j)$ dominate the second-level task-adaptive disentanglement.

## 3 EXPERIMENTS

We evaluate DL$^2$ on a semi-synthetic dataset (DDMNISTMM) and two real-world benchmarks (CUBICC and CelebA-HQ), comparing against state-of-the-art methods. As shown in Figure 4, DDM-NISTMM contains three modalities, each displaying two independent MNIST digit sets against modality-specific backgrounds. Crucially, digit labels are shared across modalities, making them the ground-truth shared information. This provides a clean benchmark for multimodal disentanglement. CUBICC, a challenging image-text clustering benchmark (Palumbo et al. (2024)), tests real-world

Table 1: The clustering comparison results (mean(std)) on the three datasets. The best and second best results in all methods are represented by bold value and underline value, respectively.

| Dataset | DDMNISTMM | | | CUBICC | | |
|---|---|---|---|---|---|---|
| Method | ACC(↑) | NMI(↑) | ARI(↑) | ACC(↑) | NMI(↑) | ARI(↑) |
| VaDE | 0.12(0.02) | 0.01(0.02) | 0.00(0.01) | 0.18(0.00) | 0.02(0.00) | 0.00(0.00) |
| DC-GMM | 0.17(0.02) | 0.04(0.01) | 0.02(0.01) | 0.35(0.02) | 0.24(0.03) | 0.15(0.03) |
| SDEC | 0.13(0.00) | 0.01(0.00) | 0.00(0.00) | 0.15(0.01) | 0.01(0.00) | -0.01(0.00) |
| C-IDEC | 0.14(0.00) | 0.01(0.00) | 0.00(0.00) | 0.28(0.01) | 0.40(0.00) | 0.18(0.04) |
| VolMaxDCC | 0.28(0.16) | 0.17(0.20) | 0.14(0.16) | 0.80(0.04) | 0.82(0.04) | 0.74(0.05) |
| CMVAE | 0.11(0.00) | 0.00(0.00) | 0.00(0.00) | 0.55(0.13) | 0.52(0.11) | 0.35(0.16) |
| DL$^2$ | **0.99(0.00)** | **0.99(0.00)** | **0.99(0.00)** | **0.95(0.00)** | **0.90(0.01)** | **0.88(0.02)** |
| Dataset | CelebA-HQ (Smiling Intensity) | | | CelebA-HQ (Gender) | | |
| Method | ACC(↑) | NMI(↑) | ARI(↑) | ACC(↑) | NMI(↑) | ARI(↑) |
| VaDE | 0.51(0.00) | 0.00(0.00) | 0.00(0.00) | 0.64(0.04) | 0.06(0.03) | 0.07(0.04) |
| DC-GMM | 0.56(0.04) | 0.22(0.02) | 0.36(0.05) | 0.98(0.00) | 0.85(0.00) | 0.92(0.00) |
| SDEC | 0.52(0.00) | 0.02(0.00) | 0.01(0.00) | 0.66(0.03) | 0.07(0.07) | 0.05(0.05) |
| C-IDEC | 0.72(0.05) | 0.43(0.03) | 0.58(0.01) | 0.97(0.00) | 0.83(0.02) | 0.90(0.00) |
| VolMaxDCC | 0.67(0.18) | 0.70(0.24) | 0.68(0.33) | 0.63(0.00) | 0.00(0.00) | 0.00(0.00) |
| CMVAE | 0.30(0.00) | 0.03(0.03) | 0.02(0.02) | 0.59(0.05) | 0.04(0.05) | -0.03(0.02) |
| DL$^2$ | **0.86(0.01)** | **0.71(0.03)** | **0.88(0.01)** | **0.99(0.00)** | **0.97(0.00)** | **0.99(0.00)** |

On DDMNIST, unimodal methods show average performance across modalities; on CUBICC and CelebA-HQ, only the best-modal result is shown due to high variance. For fair comparison, CMVAE's $z$ prior uses the number of clusters rather than the original large value, resulting in different CUBICC outcomes.

performance. To assess high-level semantic disentanglement, we use CelebA-HQ (image-text) with gender and smiling intensity as distinct semantic attributes. All results report mean (std) over three random seeds. Technical details and extended results are in Appendices G and I, with metrics in Appendix H.

### 3.1 PERFORMANCE OF DL$^2$ AGAINST THE TASK-GENERATION DILEMMA

We evaluate DL$^2$ on DDMNISTMM, CUBICC, and CelebA-HQ to assess its performance under the task-generation dilemma. For DDM-NISTMM, we focus on left-digit labels with $0.1 \cdot N$ unique CSs. For CUBICC, we use bird species labels to generate $N$ unique CSs. For CelebA-HQ, we employ gender (binary) and smile intensity (6-point scale) labels, with $N$ unique CSs per attribute.

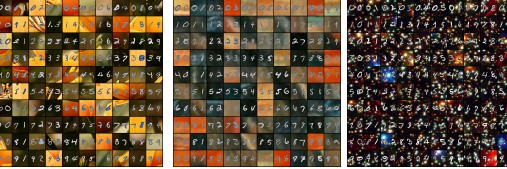

(a) modality 1    (b) modality 2    (c) modality 3

Figure 4: Illustrative samples of DDMNISTMM. Each subplot position corresponds to an individual sample, with a total of 100 samples displayed.

*Clustering performance.* To evaluate DL$^2$'s capability for targeted information disentanglement, we employ representation-sensitive clustering tasks as downstream evaluations, where clustering performance serves as the metric for disentanglement efficacy. Table 1 presents the clustering results. Unsupervised methods such as VaDE and CMVAE yield suboptimal performance due to the absence of weak supervision. More importantly, the encoder-only method VolMaxDCC generally outperforms weakly-supervised encoder-decoder approaches (e.g., DC-GMM, SDEC, C-IDEC). This advantage stems from an inherent conflict in the objective functions of the latter: the reconstruction loss encourages latent variables to encode all information, while the regularization term that incorporates CSs pushes them to capture only task-relevant information. Consequently, these models converge to a suboptimal balance between the two objectives (see Appendix I.1 for detailed analysis). In contrast, DL$^2$ achieves the best clustering performance across all datasets, as its dual-level disentanglement design effectively circumvents this conflict and enables a mutually beneficial solution.

*Generative capabilities.* To investigate whether DL$^2$ maintains generative capabilities without degradation during disentanglement, we perform both unconditional generation (using latent codes from prior distributions) and conditional generation (cross-modal generation). We compare DL$^2$'s generative capabilities against the two most advanced and relevant MVAEs (CMVAE and MM-

Table 2: The generative capabilities of $DL^2$, CMVAE and MMVAE+ on the three datasets. Bold and underline denote best and second-best results, respectively.

| Dataset | DDMNISTMM | | | | CUBICC |
|---|---|---|---|---|---|
| Method | U FID ($\downarrow$) | C FID ($\downarrow$) | CC ($\uparrow$) | UCC ($\uparrow$) | U FID ($\downarrow$) |
| MMVAE+ | **103.82(1.29)** | **99.32(1.76)** | 0.82(0.01) | 0.36(0.01) | 168.32(3.57) |
| CMVAE | 109.77(5.10) | 106.28(3.10) | 0.76(0.04) | 0.26(0.02) | 149.37(10.57) |
| $DL^2$ | 106.00(1.82) | 103.97(2.35) | **0.83(0.01)** | **0.43(0.01)** | **144.54(10.16)** |
| Dataset | CelebA-HQ (Smiling Intensity) | | | | CUBICC |
| Method | U FID ($\downarrow$) | C FID ($\downarrow$) | CC ($\uparrow$) | - | C FID ($\downarrow$) |
| MMVAE+ | 55.58(1.11) | 58.67(1.32) | 0.41(0.03) | - | 164.94 (1.50) |
| CMVAE | **53.43(1.69)** | 60.39(4.62) | 0.47(0.03) | - | 160.13(9.36) |
| $DL^2$ | 55.52(0.80) | **57.89(0.95)** | **0.51(0.06)** | - | **158.75(11.91)** |

VAE+). Results for generation quality and consistency are reported in Table 2 and Table 3 in Appendix I (consistency is not measured on CUBICC due to the lack of reliable labels for residual shared information; on CelebA-HQ, we report only conditional consistency to avoid potential noise from evaluating unconditional generation across highly heterogeneous modalities). Across the three datasets, $DL^2$ achieves the best performance in 5 out of 8 generative quality (FID) comparisons and in all four consistency evaluations, ranking second-best in the remaining tests. These results demonstrate that $DL^2$'s generative capability matches or even slightly surpasses that of CMVAE and MMVAE+. Qualitative results on DDMNISTMM provided in Appendix I.3 are consistent with these quantitative findings.

The experiments demonstrate $DL^2$'s superior task-adaptive disentanglement capability, i.e., effectively isolating target information while maintaining full information integrity.

### 3.2 VERIFYING TASK-ADAPTIVE PURITY OF $z_{(t)}$

A core claim of our task-adaptive disentanglement framework is that the task-driven component $z_{(t)}$ should capture the *minimal sufficient information* for the downstream task. While clustering performance provides initial evidence, we further quantify task-relevant information in each representation component to substantiate $DL^2$'s successful implementation of task-adaptive disentanglement. Specifically,

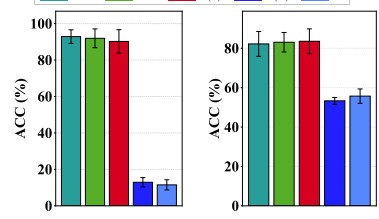

(a) DDMNISTMM (b) CelebA-HQ

Figure 5: Classifier accuracy in each representation on DDMNISTMM and CelebA-HQ (smile intensity).

we train classifiers using five distinct representations: the task-driven encoding $z_{(t)}$, residual encoding $z_{(r)}$, modality-private encoding $w$, full shared representation $z = (z_{(t)}, z_{(r)})$, and full latent encoding $u = (z, w)$. As shown in Figure 5, classifiers using only $z_{(t)}$ achieve accuracy comparable to those using $z$ or $u$, while classifiers based on $z_{(r)}$ or $w$ perform near random chance.

These results provide strong empirical evidence that: i) $z_{(t)}$ successfully captures nearly all task-relevant information present in both the full shared representation and the full latent encoding, satisfying the *sufficiency* criterion; ii) $z_{(r)}$ and $w$ are effectively purified of task-relevant semantics, containing primarily task-irrelevant information and thereby satisfying the *minimality* criterion. This demonstrates that $DL^2$ successfully disentangles modality-specific information from shared information at the first level, and further decouples task-relevant information from residual information at the second level through task-adaptive disentanglement.

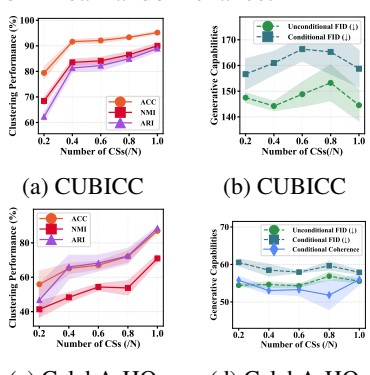

(a) CUBICC (b) CUBICC

(c) CelebA-HQ (d) CelebA-HQ

Figure 6: The TADL performance of $DL^2$ with varying number of CSs.

### 3.3 ANALYSIS OF $DL^2$

This conclusion is further supported by the latent space interpolation and semantic traversals on the DDMNISTMM dataset provided in Appendix I.4, which offer intuitive visualizations of the information encoded within each latent variable.

*Impact of Contrastive Signals Quantity.* We evaluate how the amount of weak supervision affects $DL^2$'s performance by varying the number of CSs as a ratio of total sample size $N$ (0.2, 0.4, 0.6, 0.8, 1.0) on CUBICC and CelebA-HQ datasets. Figure 6 shows that $DL^2$'s representation learning performance (ACC, NMI, ARI) exhibits a positive correlation with CS quantity. On both datasets, these metrics show stable and significant improvement as signal proportion increases, demonstrating $DL^2$'s ability to effectively utilize weak supervision for enhanced disentanglement quality. Concurrently, generation quality (FID) and consistency (Coherence) remain stable across supervision levels, indicating $DL^2$'s generative robustness to weak signal quantity. This highlights a key advantage: $DL^2$ maintains high generation quality while leveraging additional supervisory signals to improve disentanglement performance.

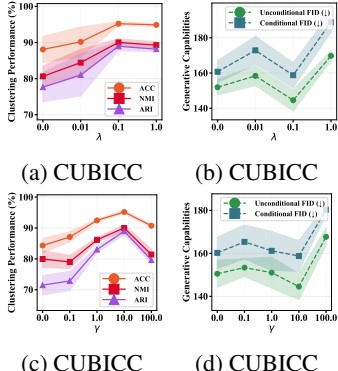

(a) CUBICC     (b) CUBICC

(c) CUBICC     (d) CUBICC

Figure 7: The TADL performance of $DL^2$ with varying values of $\lambda$ and $\gamma$.

*Impact of Regularization Strength on Performance.* $DL^2$ employs hyperparameters $\lambda$ and $\gamma$ to balance the $\mathcal{R}$PS and $\mathcal{R}$NS regularization terms, respectively. To investigate their effects on task-adaptive disentanglement, we vary one hyperparameter while keeping the other fixed, testing $\gamma$ in 0, 0.1, 1, 10, 100 and $\lambda$ in 0, 0.01, 0.1, 1. Results in Figure 7 and Appendix Figure 12 show that both regularization terms are essential for improving representation quality (ACC/NMI/ARI). As either value increases from zero, clustering performance first improves then gradually declines, indicating that moderate regularization enhances disentanglement. However, excessive constraints (e.g., $\lambda = 1$ or $\gamma = 100$ on DDMNISTMM) cause overfitting and degrade representation. Across a wide hyperparameter range (excluding extremes), generation quality (FID) and consistency remain stable, demonstrating $DL^2$'s robustness to regularization strength. This key advantage confirms that $DL^2$ effectively improves disentangled representations without compromising generation quality, validating its success as a task-adaptive disentanglement framework.

*Ablation Study.* We conduct systematic ablation studies to validate the necessity of both $\mathcal{R}_{PS}$ and $\mathcal{R}_{NS}$ regularizers. Figure 8 shows that on DDMNIST and CelebA-HQ (Smile), the variants employing only $\mathcal{R}_{PS}$ or $\mathcal{R}_{NS}$ already achieve significantly higher ACC, NMI, and ARI than the unsupervised CMVAE. Furthermore, the complete $DL^2$ model consistently outperforms these partial variants. This confirms that both regularizers are indispensable and work synergistically for high-quality task-adaptive disentanglement. Generation quality (FID) and consistency remain stable across all variants, consistent with sensitivity analysis results and reaffirming $DL^2$'s inherent robustness in preserving generative capabilities.

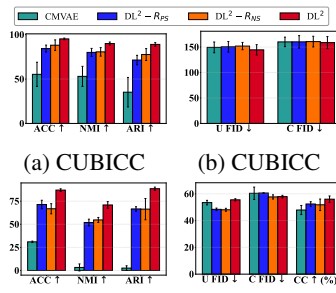

(a) CUBICC     (b) CUBICC

(c) CelebA-HQ     (d) CelebA-HQ

Figure 8: Ablation study results on CUBICC and CelebA-HQ (smile intensity) datasets.

# 4 CONCLUSION

This work identified and addressed a fundamental challenge in multimodal learning: the task-generation dilemma. We introduced Task-Adaptive Disentanglement (TADL), a novel paradigm that moves beyond universal factorization towards dynamic, task-guided separation of representations. The proposed Dual-Level Disentanglement ($DL^2$) framework provides a principled instantiation of this paradigm, demonstrating that the shared information in multimodal data can and should be adaptively decomposed into a task-relevant component for discriminative purity and a residual component for generative completeness. Crucially, $DL^2$ achieves this through innovative use of readily available contrastive signals, introducing the virtual modality pair and Common-Cause Mutual Information (CCMI) regularizers to enable effective weakly-supervised disentanglement. Extensive empirical validation confirms that $DL^2$ not only achieves state-of-the-art performance on challenging tasks like multimodal clustering but also maintains high-generation fidelity, effectively resolving the dilemma within a single model. Our work opens up new possibilities for building versatile and efficient multimodal systems that require no sacrifice between specialization and generality.

## REPRODUCIBILITY STATEMENT

To facilitate the reproducibility of our work, we have made extensive efforts to document our theoretical contributions and experimental procedures. For the theoretical results, detailed proofs of Lemma 1 and Lemma 2 are provided in Appendix C and Appendix D, respectively. Appendix E presents an information-theoretic proof elucidating the role of the first term in $\mathcal{R}_{\mathrm{PS}}(\boldsymbol{X}^k, \boldsymbol{X}^l)$. Furthermore, Appendix F contains the proof of the properties of common-cause mutual information (CCMI) (Appendix F.1) and offers an intuitive semantic interpretation (Appendix F.2).

On the experimental side, Appendix G.1 provides a comprehensive description of all datasets used, including the detailed synthesis process of our proposed semi-synthetic dataset, DDMNISTMM. A brief introduction to the baseline methods is given in Appendix G.2, while Appendix G.3 specifies the detailed experimental settings. The evaluation metrics for clustering performance and generation quality are elaborated in Appendix H. Additional experimental results and analyses are presented in Appendix I, which offers further insights into the task-generation dilemma (Appendix I.1 and Appendix I.2), qualitative comparisons (Appendix I.3), and latent space interpolations (Appendix I.4).

Finally, to support the replication of our methods, we will share the anonymized source code via a private anonymous repository link, which will be made available to the reviewers and Area Chairs in the discussion forum after it opens.

## ETHICS STATEMENT

We have read and adhere to the ICLR Code of Ethics in conducting this research. Our work presents a novel algorithm and is evaluated on both fully public benchmarks and a semi-synthetic dataset. This semi-synthetic dataset is derived entirely from public source data and does not involve any private or sensitive information. The detailed process for creating this dataset is transparently documented in Appendix G.1 to ensure reproducibility and scrutiny. To the best of our knowledge, our work does not raise any immediate or pressing ethical concerns. We will continue to be mindful of the potential societal impacts of our research.

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

# APPENDIX

## APPENDICES CONTENTS

## A    THE USE OF LARGE LANGUAGE MODELS (LLMs)

In the preparation of this manuscript, we utilized large language models (LLMs), specifically GPT-4 and Claude 3, solely as a tool to assist with the writing and polishing of the text.

The use of LLMs was strictly limited to the following aspects:

- Improving grammatical correctness and fluency.
- Rephrasing sentences for better clarity and readability.
- Checking and adjusting the tone of the writing to maintain a formal academic style.

Crucially, the core intellectual content of this work—including the central research ideas, algorithmic development, theoretical derivations, experimental design and interpretation of results—originates entirely from the human authors. The LLM was **not** involved in generating any key ideas, formulating hypotheses, designing experiments, performing mathematical proofs, or drawing scientific conclusions.

The authors have thoroughly reviewed, edited, and verified all content generated by the LLM to ensure its accuracy and alignment with the intended meaning. We take full responsibility for the entire content of this paper, including any portions that were initially drafted or polished with LLM assistance.

In accordance with the ICLR policy, we affirm that no LLM is listed as an author of this work.

## B    RELATED WORK

Our work lies at the intersection of multimodal representation disentanglement and weakly supervised disentangled representation learning. We review key related literature below.

### B.1    MULTIMODAL REPRESENTATION DISENTANGLEMENT

Learning unified representations from multimodal data remains a core challenge. Early MVAEs predominantly modeled latent information within a single shared space. For instance, MVAE (Wu & Goodman (2018)) adopted a product-of-experts encoder to integrate multimodal information into a unified latent space, while MMVAE (Shi et al. (2019)) used a mixture-of-experts structure for joint modeling, both ignoring modality-specific variations. Similarly, MoPoE-VAE (Sutter et al. (2021)) and MVTCAE (Hwang et al. (2021)) retained a single latent space, focusing on improving scalability or adding regularization without disentangling private information. These models assume that a common latent variable can capture all shared information across modalities, but they often fail to model *modality-specific variations*, leading to blurred generations and limited expressiveness.

Consequently, another line of work focuses on disentangling the latent space into *shared* and *modality-specific* components. Bouchacourt et al. (Bouchacourt et al. (2018)) proposed a multi-level VAE to decompose latent spaces into shared and group-specific components. Sutter et al. (Sutter et al. (2020)) extended mixture-based VAEs by introducing modality-specific subspaces alongside a shared subspace, though this led to hyperparameter sensitivity in balancing coherence and quality. MMVAE+ (Palumbo et al. (2023)) further refined this framework using auxiliary priors for cross-modal reconstruction, ensuring robust disentanglement of shared and private information.

Although these methods successfully separate shared from private information, they typically treat the shared representation as an *indivisible unit*. This can be suboptimal, as different semantic aspects within the shared space may be relevant for different downstream tasks. Using the entire shared representation for downstream task may introduce irrelevant features, while neglecting parts of it can harm fidelity. Our method addresses this by further decomposing the shared space into a *task-relevant* subset and residual shared information.

### B.2    DISENTANGLED REPRESENTATION LEARNING

Disentangled Representation Learning (DRL) aims to encode data into representations where distinct latent units correspond to semantically meaningful factors of variation (Bengio et al. (2013)).

A common approach builds upon variational autoencoders (VAEs) by introducing regularizers that encourage statistical independence among latent dimensions. For instance, $\beta$-VAE (Higgins et al. (2017)) strengthens the KL divergence term to promote factorized representations. FactorVAE (Kim & Mnih (2018)) and $\beta$-TCVAE (Chen et al. (2018)) explicitly minimize the total correlation to enhance dimension-wise independence. DIP-VAE (Kumar et al. (2017)) aligns the moments of the aggregate posterior with the prior distribution to encourage disentanglement.

These methods typically assume a *dimension-wise* disentanglement structure, where each scalar latent dimension controls one fine-grained factor (e.g., object color or size in synthetic datasets like Shapes3D (Burgess & Kim (2018))). However, many real-world tasks require *vector-wise* disentanglement, where a group of dimensions encodes a coarse-grained semantic concept (e.g., identity in faces (Tran et al. (2017)) or motion in videos (Denton et al. (2017))).

In this work, we further decompose the shared multimodal space in a vector-wise manner, isolating a *task-relevant* subset $z_{(t)}$ and a residual subset $z_{(r)}$ from the complete shared space $z$.

### B.3    WEAKLY SUPERVISED DISENTANGLED REPRESENTATION LEARNING

While fully unsupervised disentanglement has achieved encouraging progress, recent studies suggest that it remains highly challenging without introducing certain inductive biases (Locatello et al. (2019)). This has motivated the use of weak supervision to guide the disentanglement process with minimal annotation. Weakly supervised methods often leverage:

- **Group-level labels:** DC-IGN (Kulkarni et al. (2015)) restricts variation to one factor per mini-batch, aligning a latent dimension with that factor.

- **Partial labels:** DisUnknown (Xiang et al. (2021)) assumes $N-1$ factors are labeled and distills the remaining unknown factor adversarially.

- **Pairwise similarities:** Some methods use similarity constraints between samples to isolate factors without explicit labels (Locatello et al. (2020)).

These approaches demonstrate that even limited supervisory signals can significantly improve identifiability and disentanglement quality.

In line with this paradigm, we propose using a Contrastive Signals (CSs) —e.g., pairwise semantic similarities—to isolate the task-relevant factors within the shared multimodal space. Unlike contrastive methods that learn general representations (Chen et al. (2020)), we integrate CSs into a MVAE to explicitly disentangle task-specific component $z_{(t)}$ and a residual component $z_{(r)}$ from the complete shared space $z$ without sacrificing generative coherence.

## C  THE PROOF OF LEMMA 1

**Lemma 1.** *The primary objective $\mathcal{L}_{DL}(\boldsymbol{X})$ (Equation 2) is a valid lower bound on $\log p_\Theta(\boldsymbol{X})$.*

*Proof.* Building upon the generative process assumptions, we derive the following:

$$
\begin{aligned}
&\log p_\Theta(\boldsymbol{X}) \\
&= \mathbb{E}_{q_{\Phi_t}(\boldsymbol{z}_{(t)}|\boldsymbol{X})} \log p_\Theta(\boldsymbol{X}) \\
&= \mathbb{E}_{q_{\Phi_t}(\boldsymbol{z}_{(t)}|\boldsymbol{X})} \log \frac{p_\Theta(\boldsymbol{X}\mid\boldsymbol{z}_{(t)})\sum_{\boldsymbol{c}}p(\boldsymbol{c})p(\boldsymbol{z}_{(t)}\mid\boldsymbol{c})}{p(\boldsymbol{z}_{(t)}\mid\boldsymbol{X})} \\
&= \mathbb{E}_{q_{\Phi_t}(\boldsymbol{z}_{(t)}|\boldsymbol{X})} \log \left[ \frac{p_\Theta(\boldsymbol{X}\mid\boldsymbol{z}_{(t)})\sum_{\boldsymbol{c}}p(\boldsymbol{c})p(\boldsymbol{z}_{(t)}\mid\boldsymbol{c})}{p(\boldsymbol{z}_{(t)}\mid\boldsymbol{X})} \cdot \frac{q_{\Phi_t}(\boldsymbol{z}_{(t)}\mid\boldsymbol{X})}{q_{\Phi_t}(\boldsymbol{z}_{(t)}\mid\boldsymbol{X})} \right] \\
&= \mathbb{E}_{q_{\Phi_t}(\boldsymbol{z}_{(t)}|\boldsymbol{X})} \log \frac{p_\Theta(\boldsymbol{X}\mid\boldsymbol{z}_{(t)})\sum_{\boldsymbol{c}}p(\boldsymbol{c})p(\boldsymbol{z}_{(t)}\mid\boldsymbol{c})}{q_{\Phi_t}(\boldsymbol{z}_{(t)}\mid\boldsymbol{X})} \\
&\quad + D_{\mathrm{KL}}\left[ q_{\Phi_t}(\boldsymbol{z}_{(t)}\mid\boldsymbol{X})\|p(\boldsymbol{z}_{(t)}\mid\boldsymbol{X}) \right].
\end{aligned}
\tag{11}
$$

By leveraging the non-negativity property of the Kullback-Leibler divergence, we obtain:

$$
\begin{aligned}
&\log p_\Theta(\boldsymbol{X}) \\
&\geq \mathbb{E}_{q_{\Phi_t}(\boldsymbol{z}_{(t)}|\boldsymbol{X})} \log \frac{p_\Theta(\boldsymbol{X}\mid\boldsymbol{z}_{(t)})\sum_{\boldsymbol{c}}p(\boldsymbol{c})p(\boldsymbol{z}_{(t)}\mid\boldsymbol{c})}{q_{\Phi_t}(\boldsymbol{z}_{(t)}\mid\boldsymbol{X})} \\
&= \mathbb{E}_{q_{\Phi_t}(\boldsymbol{z}_{(t)}|\boldsymbol{X})} \log p_\Theta(\boldsymbol{X}\mid\boldsymbol{z}_{(t)}) + \mathbb{E}_{q_{\Phi_t}(\boldsymbol{z}_{(t)}|\boldsymbol{X})} \log \frac{p(\boldsymbol{c})p(\boldsymbol{z}_{(t)}\mid\boldsymbol{c})}{q_{\Phi_t}(\boldsymbol{z}_{(t)}\mid\boldsymbol{X})} \\
&= \mathbb{E}_{q_{\Phi_t}(\boldsymbol{z}_{(t)}|\boldsymbol{X})} \left[ \mathbb{E}_{q_{\Phi_r}(\boldsymbol{z}_{(r)}|\boldsymbol{X},\boldsymbol{z}_{(t)})} \log p_\Theta(\boldsymbol{X}\mid\boldsymbol{z}_{(t)}) \right] + \mathbb{E}_{q_{\Phi_t}(\boldsymbol{z}_{(t)}|\boldsymbol{X})} \log \frac{p(\boldsymbol{c})p(\boldsymbol{z}_{(t)}\mid\boldsymbol{c})}{q_{\Phi_t}(\boldsymbol{z}_{(t)}\mid\boldsymbol{X})} \\
&= \mathbb{E}_{q_{\Phi_t,\Phi_r}(\boldsymbol{z}_{(t)},\boldsymbol{z}_{(r)}|\boldsymbol{X})} \log p_\Theta(\boldsymbol{X}\mid\boldsymbol{z}_{(t)}) + \mathbb{E}_{q_{\Phi_t}(\boldsymbol{z}_{(t)}|\boldsymbol{X})} \log \frac{p(\boldsymbol{c})p(\boldsymbol{z}_{(t)}\mid\boldsymbol{c})}{q_{\Phi_t}(\boldsymbol{z}_{(t)}\mid\boldsymbol{X})}.
\end{aligned}
\tag{12}
$$

Then, we can use the following two equation relationships:

$$
p_\Theta(\boldsymbol{X}\mid\boldsymbol{z}_{(t)})\cdot p(\boldsymbol{z}_{(r)}\mid\boldsymbol{X}) = p_\Theta(\boldsymbol{X},\boldsymbol{z}_{(r)}\mid\boldsymbol{z}_{(t)})
\tag{13}
$$

and

$$
p_\Theta(\boldsymbol{X}\mid\boldsymbol{z}_{(t)},\boldsymbol{z}_{(r)})\cdot p(\boldsymbol{z}_{(r)}) = p_\Theta(\boldsymbol{X},\boldsymbol{z}_{(r)}\mid\boldsymbol{z}_{(t)}).
\tag{14}
$$

It can be obtained

$$
p_\Theta(\boldsymbol{X}\mid\boldsymbol{z}_{(t)}) = \frac{p_\Theta(\boldsymbol{X}\mid\boldsymbol{z}_{(t)},\boldsymbol{z}_{(r)})\cdot p(\boldsymbol{z}_{(r)})}{p(\boldsymbol{z}_{(r)}\mid\boldsymbol{X})}.
\tag{15}
$$

Then, we have:

$$
\begin{aligned}
&\log p_\Theta(\boldsymbol{X}) \\
&\geq \mathbb{E}_{q_{\Phi_t,\Phi_r}(\boldsymbol{z}_{(t)},\boldsymbol{z}_{(r)}|\boldsymbol{X})} \log p_\Theta(\boldsymbol{X}\mid\boldsymbol{z}_{(t)}) + \mathbb{E}_{q_{\Phi_t}(\boldsymbol{z}_{(t)}|\boldsymbol{X})} \log \frac{p(\boldsymbol{c})p(\boldsymbol{z}_{(t)}\mid\boldsymbol{c})}{q_{\Phi_t}(\boldsymbol{z}_{(t)}\mid\boldsymbol{X})} \\
&= \mathbb{E}_{q_{\Phi_t,\Phi_r}(\boldsymbol{z}_{(t)},\boldsymbol{z}_{(r)}|\boldsymbol{X})} \log \left[ \frac{p_\Theta(\boldsymbol{X}\mid\boldsymbol{z}_{(t)},\boldsymbol{z}_{(r)})p(\boldsymbol{z}_{(r)})}{p(\boldsymbol{z}_{(r)}\mid\boldsymbol{X})} \cdot \frac{q_{\Phi_r}(\boldsymbol{z}_{(r)}\mid\boldsymbol{X})}{q_{\Phi_r}(\boldsymbol{z}_{(r)}\mid\boldsymbol{X})} \right] \\
&\quad + \mathbb{E}_{q_{\Phi_t}(\boldsymbol{z}_{(t)}|\boldsymbol{X})} \log \frac{p(\boldsymbol{c})\cdot p(\boldsymbol{z}_{(t)}\mid\boldsymbol{c})}{q_{\Phi_t}(\boldsymbol{z}_{(t)}\mid\boldsymbol{X})} \\
&= \mathbb{E}_{q_{\Phi_t,\Phi_r}(\boldsymbol{z}_{(t)},\boldsymbol{z}_{(r)}|\boldsymbol{X})} \log p_\Theta(\boldsymbol{X}\mid\boldsymbol{z}_{(t)},\boldsymbol{z}_{(r)}) \\
&\quad + \mathbb{E}_{q_{\Phi_t}(\boldsymbol{z}_{(r)}|\boldsymbol{X})} \log \frac{p(\boldsymbol{z}_{(r)})}{q_{\Phi_r}(\boldsymbol{z}_{(r)}\mid\boldsymbol{X})} + D_{\mathrm{KL}}\left[ q_{\Phi_r}(\boldsymbol{z}_{(r)}\mid\boldsymbol{X})\|p(\boldsymbol{z}_{(r)}\mid\boldsymbol{X}) \right] \\
&\quad + \mathbb{E}_{q_{\Phi_t}(\boldsymbol{z}_{(t)}|\boldsymbol{X})} \log \frac{p(\boldsymbol{c})p(\boldsymbol{z}_{(t)}\mid\boldsymbol{c})}{q_{\Phi_t}(\boldsymbol{z}_{(t)}\mid\boldsymbol{X})}.
\end{aligned}
\tag{16}
$$

By leveraging the non-negativity property of the Kullback-Leibler divergence, we obtain:

$$
\begin{aligned}
&\log p_{\Theta}(\boldsymbol{X}) \\
&\geq \mathbb{E}_{q_{\Phi_t,\Phi_r}(\boldsymbol{z}_{(t)},\boldsymbol{z}_{(r)}|\boldsymbol{X})} \log p_{\Theta}(\boldsymbol{X} \mid \boldsymbol{z}_{(t)}, \boldsymbol{z}_{(r)}) \\
&\quad + \mathbb{E}_{q_{\Phi_t}(\boldsymbol{z}_{(r)}|\boldsymbol{X})} \log \frac{p(\boldsymbol{z}_{(r)})}{q_{\Phi_r}(\boldsymbol{z}_{(r)} \mid \boldsymbol{X})} + D_{\mathrm{KL}}\left[q_{\Phi_r}(\boldsymbol{z}_{(r)} \mid \boldsymbol{X}) \| p(\boldsymbol{z}_{(r)} \mid \boldsymbol{X})\right] \\
&\quad + \mathbb{E}_{q_{\Phi_t}(\boldsymbol{z}_{(t)}|\boldsymbol{X})} \log \frac{p(\boldsymbol{c})p(\boldsymbol{z}_{(t)} \mid \boldsymbol{c})}{q_{\Phi_t}(\boldsymbol{z}_{(t)} \mid \boldsymbol{X})} \\
&\geq \mathbb{E}_{q_{\Phi_t,\Phi_r}(\boldsymbol{z}_{(t)},\boldsymbol{z}_{(r)}|\boldsymbol{X})} \log p_{\Theta}(\boldsymbol{X} \mid \boldsymbol{z}_{(t)}, \boldsymbol{z}_{(r)}) \\
&\quad + \mathbb{E}_{q_{\Phi_t}(\boldsymbol{z}_{(r)}|\boldsymbol{X})} \log \frac{p(\boldsymbol{z}_{(r)})}{q_{\Phi_r}(\boldsymbol{z}_{(r)} \mid \boldsymbol{X})} + \mathbb{E}_{q_{\Phi_t}(\boldsymbol{z}_{(t)}|\boldsymbol{X})} \log \frac{p(\boldsymbol{c})p(\boldsymbol{z}_{(t)} \mid \boldsymbol{c})}{q_{\Phi_t}(\boldsymbol{z}_{(t)} \mid \boldsymbol{X})}.
\end{aligned}
\tag{17}
$$

We can then expand $q_{\Phi_t,\Phi_r}(\boldsymbol{z}_{(t)}, \boldsymbol{z}_{(r)} \mid \boldsymbol{X})$ and $p_{\Theta}(\boldsymbol{X} \mid \boldsymbol{z}_{(t)}, \boldsymbol{z}_{(r)})$ over their modalities to obtain:

$$
\begin{aligned}
&\log p_{\Theta}(\boldsymbol{X}) \\
&\geq \mathbb{E}_{q_{\Phi_t,\Phi_r}(\boldsymbol{z}_{(t)},\boldsymbol{z}_{(r)}|\boldsymbol{X})} \log p_{\Theta}(\boldsymbol{X} \mid \boldsymbol{z}_{(t)}, \boldsymbol{z}_{(r)}) \\
&\quad + \mathbb{E}_{q_{\Phi_t}(\boldsymbol{z}_{(r)}|\boldsymbol{X})} \log \frac{p(\boldsymbol{z}_{(r)})}{q_{\Phi_r}(\boldsymbol{z}_{(r)} \mid \boldsymbol{X})} + \mathbb{E}_{q_{\Phi_t}(\boldsymbol{z}_{(t)}|\boldsymbol{X})} \log \frac{p(\boldsymbol{c})p(\boldsymbol{z}_{(t)} \mid \boldsymbol{c})}{q_{\Phi_t}(\boldsymbol{z}_{(t)} \mid \boldsymbol{X})} \\
&= \underbrace{\frac{1}{M} \sum_{m=1}^{M} \mathbb{E}_{q_{\phi_{\boldsymbol{z}_m}}(\boldsymbol{z}_{(t)},\boldsymbol{z}_{(r)}|\boldsymbol{x}_m)} \log p_{\theta_m}(\boldsymbol{x}_m \mid \boldsymbol{z}_{(t)}, \boldsymbol{z}_{(r)})}_{\mathcal{I}_1: \text{ intra-modal reconstruction likelihood term}} \\
&\quad + \underbrace{\frac{1}{M} \sum_{m=1}^{M} \mathbb{E}_{q_{\phi_{\boldsymbol{z}_m}}(\boldsymbol{z}_{(t)},\boldsymbol{z}_{(r)}|\boldsymbol{x}_m)} \sum_{n \neq m} \log p_{\theta_n}(\boldsymbol{x}_n \mid \boldsymbol{z}_{(t)}, \boldsymbol{z}_{(r)})}_{\mathcal{I}_2: \text{ cross-modal reconstruction likelihood term}} \\
&\quad + \underbrace{\mathbb{E}_{q_{\Phi_t}(\boldsymbol{z}_{(t)}|\boldsymbol{X})} \log \frac{p(\boldsymbol{c})p(\boldsymbol{z}_{(t)} \mid \boldsymbol{c})}{q_{\Phi_t}(\boldsymbol{z}_{(t)} \mid \boldsymbol{X})} + \mathbb{E}_{q_{\Phi_t}(\boldsymbol{z}_{(r)}|\boldsymbol{X})} \log \frac{p(\boldsymbol{z}_{(r)})}{q_{\Phi_r}(\boldsymbol{z}_{(r)} \mid \boldsymbol{X})}}_{\text{Kullback-Leibler term}}.
\end{aligned}
\tag{18}
$$

For *the intra-modal reconstruction likelihood term* $\mathcal{I}_1$, we derive its evidence lower bound (ELBO) via the variational encoders $\{q_{\phi_{\boldsymbol{w}_m}}(\boldsymbol{w}_m \mid \boldsymbol{x}_m)\}_{m=1}^{M}$:

$$
\begin{aligned}
&\frac{1}{M} \sum_{m=1}^{M} \mathbb{E}_{q_{\phi_{\boldsymbol{z}_m}}(\boldsymbol{z}_{(t)},\boldsymbol{z}_{(r)}|\boldsymbol{x}_m)} \log p_{\theta_m}(\boldsymbol{x}_m \mid \boldsymbol{z}_{(t)}, \boldsymbol{z}_{(r)}) \\
&= \frac{1}{M} \sum_{m=1}^{M} \mathbb{E}_{\substack{q_{\phi_{\boldsymbol{z}_m}}(\boldsymbol{z}_{(t)},\boldsymbol{z}_{(r)}|\boldsymbol{x}_m) \\ q_{\phi_{\boldsymbol{w}_m}}(\boldsymbol{w}_m|\boldsymbol{x}_m)}} \log p_{\theta_m}(\boldsymbol{x}_m \mid \boldsymbol{z}_{(t)}, \boldsymbol{z}_{(r)}) \\
&= \frac{1}{M} \sum_{m=1}^{M} \mathbb{E}_{\substack{q_{\phi_{\boldsymbol{z}_m}}(\boldsymbol{z}_{(t)},\boldsymbol{z}_{(r)}|\boldsymbol{x}_m) \\ q_{\phi_{\boldsymbol{w}_m}}(\boldsymbol{w}_m|\boldsymbol{x}_m)}} \left[\log \frac{p_{\theta_m}(\boldsymbol{x}_m \mid \boldsymbol{z}_{(t)}, \boldsymbol{z}_{(r)}, \boldsymbol{w}_m)p(\boldsymbol{w}_m)}{p(\boldsymbol{w}_m \mid \boldsymbol{x}_m)} \cdot \frac{q_{\phi_{\boldsymbol{w}_m}}(\boldsymbol{w}_m \mid \boldsymbol{x}_m)}{q_{\phi_{\boldsymbol{w}_m}}(\boldsymbol{w}_m \mid \boldsymbol{x}_m)}\right] \\
&= \frac{1}{M} \sum_{m=1}^{M} \mathbb{E}_{\substack{q_{\phi_{\boldsymbol{z}_m}}(\boldsymbol{z}_{(t)},\boldsymbol{z}_{(r)}|\boldsymbol{x}_m) \\ q_{\phi_{\boldsymbol{w}_m}}(\boldsymbol{w}_m|\boldsymbol{x}_m)}} \log \frac{p_{\theta_m}(\boldsymbol{x}_m^h \mid \boldsymbol{z}_{(t)}, \boldsymbol{z}_{(r)}, \boldsymbol{w}_m)p(\boldsymbol{w}_m)}{q_{\phi_{\boldsymbol{w}_m}}(\boldsymbol{w}_m \mid \boldsymbol{x}_m)} \\
&\quad + \frac{1}{M} \sum_{m=1}^{M} D_{\mathrm{KL}}(q_{\phi_{\boldsymbol{w}_m}}(\boldsymbol{w}_m \mid \boldsymbol{x}_m) \| p(\boldsymbol{w}_m \mid \boldsymbol{x}_m)).
\end{aligned}
\tag{19}
$$

According to the non-negativity of the Kullback-Leibler divergence, equation 19 can be obtained:

$$
\frac{1}{M} \sum_{m=1}^{M} \mathbb{E}_{q_{\phi_{\boldsymbol{z}_m}}(\boldsymbol{z}_{(t)}, \boldsymbol{z}_{(r)} \mid \boldsymbol{x}_m)} \log p_{\theta_m}(\boldsymbol{x}_m \mid \boldsymbol{z}_{(t)}, \boldsymbol{z}_{(r)})
$$

$$
\geq \frac{1}{M} \sum_{m=1}^{M} \mathbb{E}_{\substack{q_{\phi_{\boldsymbol{z}_m}}(\boldsymbol{z}_{(t)}, \boldsymbol{z}_{(r)} \mid \boldsymbol{x}_m) \\ q_{\phi_{\boldsymbol{w}_m}}(\boldsymbol{w}_m \mid \boldsymbol{x}_m)}} \log \frac{p_{\theta_m}(\boldsymbol{x}_m \mid \boldsymbol{z}_{(t)}, \boldsymbol{z}_{(r)}, \boldsymbol{w}_m) p(\boldsymbol{w}_m)}{q_{\phi_{\boldsymbol{w}_m}}(\boldsymbol{w}_m \mid \boldsymbol{x}_m)}
$$

$$
= \frac{1}{M} \sum_{m=1}^{M} \mathbb{E}_{\substack{q_{\phi_{\boldsymbol{z}_m}}(\boldsymbol{z}_{(t)}, \boldsymbol{z}_{(r)} \mid \boldsymbol{x}_m) \\ q_{\phi_{\boldsymbol{w}_m}}(\boldsymbol{w}_m \mid \boldsymbol{x}_m)}} \log p_{\theta_m}(\boldsymbol{x}_m \mid \boldsymbol{z}_{(t)}, \boldsymbol{z}_{(r)}, \boldsymbol{w}_m)
$$

$$
+ \frac{1}{M} \sum_{m=1}^{M} \mathbb{E}_{q_{\phi_{\boldsymbol{w}_m}}(\boldsymbol{w}_m \mid \boldsymbol{x}_m)} \log \frac{p(\boldsymbol{w}_m)}{q_{\phi_{\boldsymbol{w}_m}}(\boldsymbol{w}_m \mid \boldsymbol{x}_m)}. \tag{20}
$$

For *the cross-modal reconstruction likelihood term* $\mathcal{I}_2$, we establish its lower bound by applying Jensen's inequality:

$$
\frac{1}{M} \sum_{m=1}^{M} \mathbb{E}_{q_{\phi_{\boldsymbol{z}_m}}(\boldsymbol{z}_{(t)}, \boldsymbol{z}_{(r)} \mid \boldsymbol{x}_m)} \sum_{n \neq m} \log p_{\theta_n}(\boldsymbol{x}_n \mid \boldsymbol{z}_{(t)}, \boldsymbol{z}_{(r)})
$$

$$
\geq \frac{1}{M} \sum_{m=1}^{M} \mathbb{E}_{\substack{q_{\phi_{\boldsymbol{z}_m}}(\boldsymbol{z}_{(t)}, \boldsymbol{z}_{(r)} \mid \boldsymbol{x}_m) \\ \{\widetilde{\boldsymbol{w}}_n \sim r_n(\boldsymbol{w}_n)\}_{n=1}^{M}}} \sum_{n \neq m} \log p_{\theta_n}(\boldsymbol{x}_n \mid \boldsymbol{z}_{(t)}, \boldsymbol{z}_{(r)}, \widetilde{\boldsymbol{w}}_n). \tag{21}
$$

By combining equations 18, 20 and 21, we derive the evidence lower bound for $\log p_{\Theta}(\boldsymbol{X})$:

$$
\log p_{\Theta}(\boldsymbol{X})
$$

$$
\geq \underbrace{\frac{1}{M} \sum_{m=1}^{M} \mathbb{E}_{q_{\phi_{\boldsymbol{z}_m}}(\boldsymbol{z}_{(t)}, \boldsymbol{z}_{(r)} \mid \boldsymbol{x}_m)} \log p_{\theta_m}(\boldsymbol{x}_m \mid \boldsymbol{z}_{(t)}, \boldsymbol{z}_{(r)})}_{\mathcal{I}_1 : \text{intra-modal reconstruction likelihood term}}
$$

$$
+ \underbrace{\frac{1}{M} \sum_{m=1}^{M} \mathbb{E}_{q_{\phi_{\boldsymbol{z}_m}}(\boldsymbol{z}_{(t)}, \boldsymbol{z}_{(r)} \mid \boldsymbol{x}_m)} \sum_{n \neq m} \log p_{\theta_n}(\boldsymbol{x}_n \mid \boldsymbol{z}_{(t)}, \boldsymbol{z}_{(r)})}_{\mathcal{I}_2 : \text{cross-modal reconstruction likelihood term}}
$$

$$
+ \underbrace{\mathbb{E}_{q_{\Phi_t}(\boldsymbol{z}_{(t)} \mid \boldsymbol{X})} \log \frac{p(\boldsymbol{c}) p(\boldsymbol{z}_{(t)} \mid \boldsymbol{c})}{q_{\Phi_t}(\boldsymbol{z}_{(t)} \mid \boldsymbol{X})} + \mathbb{E}_{q_{\Phi_t}(\boldsymbol{z}_{(r)} \mid \boldsymbol{X})} \log \frac{p(\boldsymbol{z}_{(r)})}{q_{\Phi_r}(\boldsymbol{z}_{(r)} \mid \boldsymbol{X})}}_{\text{Kullback-Leibler term}}
$$

$$
\geq \underbrace{\frac{1}{M} \sum_{m=1}^{M} \mathbb{E}_{\substack{q_{\phi_{\boldsymbol{z}_m}}(\boldsymbol{z}_{(t)}, \boldsymbol{z}_{(r)} \mid \boldsymbol{x}_m) \\ q_{\phi_{\boldsymbol{w}_m}}(\boldsymbol{w}_m \mid \boldsymbol{x}_m)}} \log p_{\theta_m}(\boldsymbol{x}_m \mid \boldsymbol{z}_{(t)}, \boldsymbol{z}_{(r)}, \boldsymbol{w}_m)}_{\mathcal{I}_1 : \text{intra-modal reconstruction likelihood term}}
$$

$$
+ \underbrace{\frac{1}{M} \sum_{m=1}^{M} \mathbb{E}_{\substack{q_{\phi_{\boldsymbol{z}_m}}(\boldsymbol{z}_{(t)}, \boldsymbol{z}_{(r)} \mid \boldsymbol{x}_m) \\ \{\widetilde{\boldsymbol{w}}_n \sim r_n(\boldsymbol{w}_n)\}_{n=1}^{M}}} \sum_{n \neq m} \log p_{\theta_n}(\boldsymbol{x}_n \mid \boldsymbol{z}_{(t)}, \boldsymbol{z}_{(r)}, \widetilde{\boldsymbol{w}}_n)}_{\mathcal{I}_2 : \text{cross-modal reconstruction likelihood term}}
$$

$$
+ \mathbb{E}_{q_{\Phi_t}(\boldsymbol{z}_{(t)} \mid \boldsymbol{X})} \log \frac{p(\boldsymbol{c}) p(\boldsymbol{z}_{(t)} \mid \boldsymbol{c})}{q_{\Phi_t}(\boldsymbol{z}_{(t)} \mid \boldsymbol{X})} + \mathbb{E}_{q_{\Phi_t}(\boldsymbol{z}_{(r)} \mid \boldsymbol{X})} \log \frac{p(\boldsymbol{z}_{(r)})}{q_{\Phi_r}(\boldsymbol{z}_{(r)} \mid \boldsymbol{X}^h)}
$$

$$
+ \frac{1}{M} \sum_{m=1}^{M} \mathbb{E}_{q_{\phi_{\boldsymbol{w}_m}}(\boldsymbol{w}_m \mid \boldsymbol{x}_m)} \log \frac{p(\boldsymbol{w}_m)}{q_{\phi_{\boldsymbol{w}_m}}(\boldsymbol{w}_m \mid \boldsymbol{x}_m)}. \tag{22}
$$

$\square$

## D  THE PROOF OF LEMMA 2

**Lemma 2.** *The similarity regularization $\mathcal{R}_{PS}(\boldsymbol{X}^k, \boldsymbol{X}^l)$ (Equation 4) is a valid lower bound on* $\log p_\Theta(\boldsymbol{X}^k, \boldsymbol{X}^l)$.

*Proof.* To establish a novel paradigm that treats positive sample pairs as "*virtual modality pairs*" and disentangles task-driven semantic subsets via extended variational inference, we first posit that such pairs could be generated by:

$$
\begin{aligned}
&p_\Theta(\boldsymbol{X}^k, \boldsymbol{X}^l, \boldsymbol{W}^k, \boldsymbol{W}^l, \boldsymbol{z}^k_{(r)}, \boldsymbol{z}^l_{(r)}, \boldsymbol{z}_{(t)}, \boldsymbol{c}) \\
=&p_\Theta(\boldsymbol{X}^k, \boldsymbol{W}^k, \boldsymbol{z}^k_{(r)}, \boldsymbol{z}_{(t)}, \boldsymbol{c}) \cdot p_\Theta(\boldsymbol{X}^l, \boldsymbol{W}^l, \boldsymbol{z}^l_{(r)}, \boldsymbol{z}_{(t)}, \boldsymbol{c}),
\end{aligned}
\tag{23}
$$

where

$$
\begin{aligned}
&p_\Theta(\boldsymbol{X}^k, \boldsymbol{W}^k, \boldsymbol{z}^k_{(r)}, \boldsymbol{z}_{(t)}, \boldsymbol{c}) \\
=&p_\pi(\boldsymbol{c}) \cdot p(\boldsymbol{z}^k_{(r)}) \cdot p(\boldsymbol{z}_{(t)} \mid \boldsymbol{c}) \cdot \prod_{m=1}^{M} p(\boldsymbol{w}^k_m) \cdot p_{\theta_m}(\boldsymbol{x}^k_m \mid \boldsymbol{w}^k_m, \boldsymbol{z}^k_{(r)}, \boldsymbol{z}_{(t)})
\end{aligned}
\tag{24}
$$

and

$$
\begin{aligned}
&p_\Theta(\boldsymbol{X}^l, \boldsymbol{W}^l, \boldsymbol{z}^l_{(r)}, \boldsymbol{z}_{(t)}, \boldsymbol{c}) \\
=&p_\pi(\boldsymbol{c}) \cdot p(\boldsymbol{z}^l_{(r)}) \cdot p(\boldsymbol{z}_{(t)} \mid \boldsymbol{c}) \cdot \prod_{m=1}^{M} p(\boldsymbol{w}^l_m) \cdot p_{\theta_m}(\boldsymbol{x}^l_m \mid \boldsymbol{w}^l_m, \boldsymbol{z}^l_{(r)}, \boldsymbol{z}_{(t)}).
\end{aligned}
\tag{25}
$$

It is obvious that $\boldsymbol{z}_{(t)}$ captures consistent semantics between positive sample pairs. We estimate $\boldsymbol{z}_{(t)}$ through a mixture-of-experts (MOE) encoder for similarity sample pairs, as formalized in equation 6.

Then, we can derive the following.

$$
\begin{aligned}
&\log p_\Theta(\boldsymbol{X}^k, \boldsymbol{X}^l) \\
=&\mathbb{E}_{q_{\Phi_t}(\boldsymbol{z}_{(t)} \mid \boldsymbol{X}^k, \boldsymbol{X}^l)} \log p_\Theta(\boldsymbol{X}^k, \boldsymbol{X}^l) \\
=&\mathbb{E}_{q_{\Phi_t}(\boldsymbol{z}_{(t)} \mid \boldsymbol{X}^k, \boldsymbol{X}^l)} \log \frac{p_\Theta(\boldsymbol{X}^k, \boldsymbol{X}^l \mid \boldsymbol{z}_{(t)}) \sum_{\boldsymbol{c}} p(\boldsymbol{c}) p(\boldsymbol{z}_{(t)} \mid \boldsymbol{c})}{p(\boldsymbol{z}_{(t)} \mid \boldsymbol{X}^k, \boldsymbol{X}^l)} \\
=&\mathbb{E}_{q_{\Phi_t}(\boldsymbol{z}_{(t)} \mid \boldsymbol{X}^k, \boldsymbol{X}^l)} \log \left[ \frac{p_\Theta(\boldsymbol{X}^k, \boldsymbol{X}^l \mid \boldsymbol{z}_{(t)}) \sum_{\boldsymbol{c}} p(\boldsymbol{c}) p(\boldsymbol{z}_{(t)} \mid \boldsymbol{c})}{p(\boldsymbol{z}_{(t)} \mid \boldsymbol{X}^k, \boldsymbol{X}^l)} \cdot \frac{q_{\Phi_t}(\boldsymbol{z}_{(t)} \mid \boldsymbol{X}^k, \boldsymbol{X}^l)}{q_{\Phi_t}(\boldsymbol{z}_{(t)} \mid \boldsymbol{X}^k, \boldsymbol{X}^l)} \right] \\
=&\mathbb{E}_{q_{\Phi_t}(\boldsymbol{z}_{(t)} \mid \boldsymbol{X}^k, \boldsymbol{X}^l)} \log \frac{p_\Theta(\boldsymbol{X}^k, \boldsymbol{X}^l \mid \boldsymbol{z}_{(t)}) \sum_{\boldsymbol{c}} p(\boldsymbol{c}) p(\boldsymbol{z}_{(t)} \mid \boldsymbol{c})}{q_{\Phi_t}(\boldsymbol{z}_{(t)} \mid \boldsymbol{X}^k, \boldsymbol{X}^l)} \\
&+ D_{\mathrm{KL}} \left[ q_{\Phi_t}(\boldsymbol{z}_{(t)} \mid \boldsymbol{X}^k, \boldsymbol{X}^l) \| p(\boldsymbol{z}_{(t)} \mid \boldsymbol{X}^k, \boldsymbol{X}^l) \right].
\end{aligned}
\tag{26}
$$

According to the non-negativity of the Kullback-Leibler divergence, it can be obtained:

$$
\begin{aligned}
&\log p_\Theta(\boldsymbol{X}^k, \boldsymbol{X}^l) \\
\geq&\mathbb{E}_{q_{\Phi_t}(\boldsymbol{z}_{(t)} \mid \boldsymbol{X}^k, \boldsymbol{X}^l)} \log \frac{p_\Theta(\boldsymbol{X}^k, \boldsymbol{X}^l \mid \boldsymbol{z}_{(t)}) \sum_{\boldsymbol{c}} p(\boldsymbol{c}) p(\boldsymbol{z}_{(t)} \mid \boldsymbol{c})}{q_{\Phi_t}(\boldsymbol{z}_{(t)} \mid \boldsymbol{X}^k, \boldsymbol{X}^l)}.
\end{aligned}
\tag{27}
$$

Building on the virtual modality pair generator (Equation 23), we derive:

$$\log p_{\Theta}(\boldsymbol{X}^k, \boldsymbol{X}^l)$$

$$\geq \mathbb{E}_{q_{\Phi_t}(\boldsymbol{z}_{(t)}|\boldsymbol{X}^k, \boldsymbol{X}^l)} \log \frac{p_{\Theta}(\boldsymbol{X}^k, \boldsymbol{X}^l \mid \boldsymbol{z}_{(t)}) \sum_{\boldsymbol{c}} p(\boldsymbol{c}) p(\boldsymbol{z}_{(t)} \mid \boldsymbol{c})}{q_{\Phi_t}(\boldsymbol{z}_{(t)} \mid \boldsymbol{X}^k, \boldsymbol{X}^l)}$$

$$= \mathbb{E}_{q_{\Phi_t}(\boldsymbol{z}_{(t)}|\boldsymbol{X}^k, \boldsymbol{X}^l)} \log \frac{p_{\Theta}(\boldsymbol{X}^k \mid \boldsymbol{z}_{(t)}) p_{\Theta}(\boldsymbol{X}^l \mid \boldsymbol{z}_{(t)}) \sum_{\boldsymbol{c}} p(\boldsymbol{c}) p(\boldsymbol{z}_{(t)} \mid \boldsymbol{c})}{q_{\Phi_t}(\boldsymbol{z}_{(t)} \mid \boldsymbol{X}^k, \boldsymbol{X}^l)}$$

$$= \frac{1}{2} \sum_{h \in \{k,l\}} \mathbb{E}_{q_{\Phi_t}(\boldsymbol{z}_{(t)}|\boldsymbol{X}^h)} \left[ \log p_{\Theta}(\boldsymbol{X}^h \mid \boldsymbol{z}_{(t)}) + \log p_{\Theta}(\boldsymbol{X}^{\overline{h}} \mid \boldsymbol{z}_{(t)}) + \log \frac{\sum_{\boldsymbol{c}} p(\boldsymbol{c}) p(\boldsymbol{z}_{(t)} \mid \boldsymbol{c})}{q_{\Phi_t}(\boldsymbol{z}_{(t)} \mid \boldsymbol{X}^k, \boldsymbol{X}^l)} \right]$$

$$= \frac{1}{2} \sum_{h \in \{k,l\}} \left[ \underbrace{\mathbb{E}_{q_{\Phi_t}(\boldsymbol{z}_{(t)}|\boldsymbol{X}^h)} \sum_{n=1}^{M} \log p_{\theta_n}(\boldsymbol{x}_n^h \mid \boldsymbol{z}_{(t)})}_{\text{self-sample reconstruction likelihood term}} + \underbrace{\mathbb{E}_{q_{\Phi_t}(\boldsymbol{z}_{(t)}|\boldsymbol{X}^h)} \sum_{n=1}^{M} \log p_{\theta_n}(\boldsymbol{x}_n^{\overline{h}} \mid \boldsymbol{z}_{(t)})}_{\text{similar sample reconstruction likelihood term}} \right.$$

$$\left. + \underbrace{\mathbb{E}_{q_{\Phi_t}(\boldsymbol{z}_{(t)}|\boldsymbol{X}^h)} \log \frac{\sum_{\boldsymbol{c}} p(\boldsymbol{c}) p(\boldsymbol{z}_{(t)} \mid \boldsymbol{c})}{q_{\Phi_t}(\boldsymbol{z}_{(t)} \mid \boldsymbol{X}^k, \boldsymbol{X}^l)}}_{\text{Kullback-Leibler term}} \right].$$

$$(28)$$

Then, we can derive the evidence lower bounds for *the self-sample reconstruction likelihood term* and *the similar sample reconstruction term*, respectively. First, by invoking Jensen's inequality, we derive the evidence lower bound for *the similar sample reconstruction term* $\mathbb{E}_{q_{\Phi_t}(\boldsymbol{z}_{(t)}|\boldsymbol{X}^h)} \sum_{n=1}^{M} \log p_{\theta_n}(\boldsymbol{x}_n^{\overline{h}} \mid \boldsymbol{z}_{(t)})$ :

$$\mathbb{E}_{q_{\Phi_t}(\boldsymbol{z}_{(t)}|\boldsymbol{X}^h)} \sum_{n=1}^{M} \log p_{\theta_n}(\boldsymbol{x}_n^{\overline{h}} \mid \boldsymbol{z}_{(t)})$$

$$= \mathbb{E}_{q_{\Phi_t}(\boldsymbol{z}_{(t)}|\boldsymbol{X}^h)} \sum_{n=1}^{M} \log \mathbb{E}_{\substack{\widetilde{\boldsymbol{z}}_{(r)} \sim f(\boldsymbol{z}_{(r)}) \\ \widetilde{\boldsymbol{w}}_n \sim r_n(\boldsymbol{w}_n)}} p_{\theta_n}(\boldsymbol{x}_n^{\overline{h}} \mid \boldsymbol{z}_{(t)}, \widetilde{\boldsymbol{z}}_{(r)}, \widetilde{\boldsymbol{w}}_n)$$

$$\geq \mathbb{E}_{q_{\Phi_t}(\boldsymbol{z}_{(t)}|\boldsymbol{X}^h)} \sum_{n=1}^{M} \mathbb{E}_{\substack{\widetilde{\boldsymbol{z}}_{(r)} \sim f(\boldsymbol{z}_{(r)}) \\ \widetilde{\boldsymbol{w}}_n \sim r_n(\boldsymbol{w}_n)}} \log p_{\theta_n}(\boldsymbol{x}_n^{\overline{h}} \mid \boldsymbol{z}_{(t)}, \widetilde{\boldsymbol{z}}_{(r)}, \widetilde{\boldsymbol{w}}_n)$$

$$= \mathbb{E}_{\substack{q_{\Phi_t}(\boldsymbol{z}_{(t)}|\boldsymbol{X}^h) \\ \widetilde{\boldsymbol{z}}_{(r)} \sim f(\boldsymbol{z}_{(r)}) \\ \{\widetilde{\boldsymbol{w}}_n \sim r_n(\boldsymbol{w}_n)\}_{n=1}^{M}}} \sum_{n=1}^{M} \log p_{\theta_n}(\boldsymbol{x}_n^{\overline{h}} \mid \boldsymbol{z}_{(t)}, \widetilde{\boldsymbol{z}}_{(r)}, \widetilde{\boldsymbol{w}}_n).$$

$$(29)$$

For *the self-sample reconstruction likelihood term*, we can make the following derivation.

$$\mathbb{E}_{q_{\Phi_t}(\boldsymbol{z}_{(t)}|\boldsymbol{X}^h)} \sum_{n=1}^{M} \log p_{\theta_n}(\boldsymbol{x}_n^h \mid \boldsymbol{z}_{(t)}) = \mathbb{E}_{q_{\Phi_t}(\boldsymbol{z}_{(t)}|\boldsymbol{X}^h)} \log p_{\Theta}(\boldsymbol{X}^h \mid \boldsymbol{z}_{(t)})$$

$$= \mathbb{E}_{q_{\Phi_t}(\boldsymbol{z}_{(t)}|\boldsymbol{X}^h)} \left[ \mathbb{E}_{q_{\Phi_r}(\boldsymbol{z}_{(r)}|\boldsymbol{X}^h, \boldsymbol{z}_{(t)})} \log p_{\Theta}(\boldsymbol{X}^h \mid \boldsymbol{z}_{(t)}) \right]$$

$$= \mathbb{E}_{q_{\Phi_t}(\boldsymbol{z}_{(t)}, \boldsymbol{z}_{(r)}|\boldsymbol{X}^h)} \log p_{\Theta}(\boldsymbol{X}^h \mid \boldsymbol{z}_{(t)})$$

$$= \mathbb{E}_{q_{\Phi_t}(\boldsymbol{z}_{(t)}, \boldsymbol{z}_{(r)}|\boldsymbol{X}^h)} \log \left[ \frac{p_{\Theta}(\boldsymbol{X}^h \mid \boldsymbol{z}_{(t)}, \boldsymbol{z}_{(r)}) p(\boldsymbol{z}_{(r)})}{p(\boldsymbol{z}_{(r)} \mid \boldsymbol{X}^h)} \cdot \frac{q_{\Phi_r}(\boldsymbol{z}_{(r)} \mid \boldsymbol{X}^h)}{q_{\Phi_r}(\boldsymbol{z}_{(r)} \mid \boldsymbol{X}^h)} \right]$$

$$= \mathbb{E}_{q_{\Phi_t}(\boldsymbol{z}_{(t)}, \boldsymbol{z}_{(r)}|\boldsymbol{X}^h)} \log \left[ \frac{p_{\Theta}(\boldsymbol{X}^h \mid \boldsymbol{z}_{(t)}, \boldsymbol{z}_{(r)}) p(\boldsymbol{z}_{(r)})}{q_{\Phi_r}(\boldsymbol{z}_{(r)} \mid \boldsymbol{X}^h)} \right] + D_{\mathrm{KL}} \left[ q_{\Phi_r}(\boldsymbol{z}_{(r)} \mid \boldsymbol{X}^h) \| p(\boldsymbol{z}_{(r)} \mid \boldsymbol{X}^h) \right].$$

$$(30)$$

According to the non-negativity of the Kullback-Leibler divergence, equation 30 can be obtained:

$$
\mathbb{E}_{q_{\Phi_t}(\boldsymbol{z}_{(t)}|\boldsymbol{X}^h)} \sum_{n=1}^{M} \log p_{\theta_n}(\boldsymbol{x}_n^h \mid \boldsymbol{z}_{(t)})
$$

$$
=\mathbb{E}_{q_{\Phi_t}(\boldsymbol{z}_{(t)},\boldsymbol{z}_{(r)}|\boldsymbol{X}^h)} \log \left[ \frac{p_\Theta(\boldsymbol{X}^h \mid \boldsymbol{z}_{(t)},\boldsymbol{z}_{(r)})p(\boldsymbol{z}_{(r)})}{q_{\Phi_r}(\boldsymbol{z}_{(r)} \mid \boldsymbol{X}^h)} \right] + D_{\mathrm{KL}}\left[ q_{\Phi_r}(\boldsymbol{z}_{(r)} \mid \boldsymbol{X}^h) \| p(\boldsymbol{z}_{(r)} \mid \boldsymbol{X}^h) \right]
$$

$$
\geq \mathbb{E}_{q_{\Phi_t}(\boldsymbol{z}_{(t)},\boldsymbol{z}_{(r)}|\boldsymbol{X}^h)} \log p_\Theta(\boldsymbol{X}^h \mid \boldsymbol{z}_{(t)},\boldsymbol{z}_{(r)}) + \mathbb{E}_{q_{\Phi_t}(\boldsymbol{z}_{(r)}|\boldsymbol{X}^h)} \log \frac{p(\boldsymbol{z}_{(r)})}{q_{\Phi_r}(\boldsymbol{z}_{(r)} \mid \boldsymbol{X}^h)}
$$

$$
= \underbrace{\frac{1}{M}\sum_{m=1}^{M} \mathbb{E}_{q_{\phi_{\boldsymbol{z}_m}}(\boldsymbol{z}_{(t)},\boldsymbol{z}_{(r)}|\boldsymbol{x}_m^h)} \log p_{\theta_m}(\boldsymbol{x}_m^h \mid \boldsymbol{z}_{(t)},\boldsymbol{z}_{(r)})}_{\text{intra-modal reconstruction likelihood term}}
$$

$$
+ \underbrace{\frac{1}{M}\sum_{m=1}^{M} \mathbb{E}_{q_{\phi_{\boldsymbol{z}_m}}(\boldsymbol{z}_{(t)},\boldsymbol{z}_{(r)}|\boldsymbol{x}_m^h)} \sum_{n\neq m} \log p_{\theta_n}(\boldsymbol{x}_n^h \mid \boldsymbol{z}_{(t)},\boldsymbol{z}_{(r)})}_{\text{cross-modal reconstruction likelihood term}}
$$

$$
+ \underbrace{\mathbb{E}_{q_{\Phi_t}(\boldsymbol{z}_{(r)}|\boldsymbol{X}^h)} \log \frac{p(\boldsymbol{z}_{(r)})}{q_{\Phi_r}(\boldsymbol{z}_{(r)} \mid \boldsymbol{X}^h)}}_{\text{Kullback-Leibler term}} .
\tag{31}
$$

By combining equations 31, 20 and 21, we derive the evidence lower bound for *the self-sample reconstruction likelihood term*:

$$
\mathbb{E}_{q_{\Phi_t}(\boldsymbol{z}_{(t)}|\boldsymbol{X}^h)} \sum_{n=1}^{M} \log p_{\theta_n}(\boldsymbol{x}_n^h \mid \boldsymbol{z}_{(t)})
$$

$$
\geq \underbrace{\mathbb{E}_{q_{\Phi_t}(\boldsymbol{z}_{(r)}|\boldsymbol{X}^h)} \log \frac{p(\boldsymbol{z}_{(r)})}{q_{\Phi_r}(\boldsymbol{z}_{(r)} \mid \boldsymbol{X}^h)}}_{\text{Kullback-Leibler term}}
$$

$$
+ \underbrace{\frac{1}{M}\sum_{m=1}^{M} \mathbb{E}_{q_{\phi_{\boldsymbol{z}_m}}(\boldsymbol{z}_{(t)},\boldsymbol{z}_{(r)}|\boldsymbol{x}_m^h)} \log p_{\theta_m}(\boldsymbol{x}_m^h \mid \boldsymbol{z}_{(t)},\boldsymbol{z}_{(r)})}_{\text{intra-modal reconstruction likelihood term}}
$$

$$
+ \underbrace{\frac{1}{M}\sum_{m=1}^{M} \mathbb{E}_{q_{\phi_{\boldsymbol{z}_m}}(\boldsymbol{z}_{(t)},\boldsymbol{z}_{(r)}|\boldsymbol{x}_m^h)} \sum_{n\neq m} \log p_{\theta_n}(\boldsymbol{x}_n^h \mid \boldsymbol{z}_{(t)},\boldsymbol{z}_{(r)})}_{\text{cross-modal reconstruction likelihood term}}
\tag{32}
$$

$$
\geq \frac{1}{M}\sum_{m=1}^{M} \mathbb{E}_{\substack{q_{\phi_{\boldsymbol{z}_m}}(\boldsymbol{z}_{(t)},\boldsymbol{z}_{(r)}|\boldsymbol{x}_m^h) \\ q_{\phi_{\boldsymbol{w}_m}}(\boldsymbol{w}_m|\boldsymbol{x}_m^h)}} \log p_{\theta_m}(\boldsymbol{x}_m^h \mid \boldsymbol{z}_{(t)},\boldsymbol{z}_{(r)},\boldsymbol{w}_m)
$$

$$
+ \frac{1}{M}\sum_{m=1}^{M} \mathbb{E}_{\substack{q_{\phi_{\boldsymbol{z}_m}}(\boldsymbol{z}_{(t)},\boldsymbol{z}_{(r)}|\boldsymbol{x}_m^h) \\ \{\widetilde{\boldsymbol{w}}_n \sim r_n(\boldsymbol{w}_n)\}_{n=1}^{M}}} \sum_{n\neq m} \log p_{\theta_n}(\boldsymbol{x}_n^h \mid \boldsymbol{z}_{(t)},\boldsymbol{z}_{(r)},\widetilde{\boldsymbol{w}}_n)
$$

$$
+ \mathbb{E}_{q_{\Phi_t}(\boldsymbol{z}_{(r)}|\boldsymbol{X}^h)} \log \frac{p(\boldsymbol{z}_{(r)})}{q_{\Phi_r}(\boldsymbol{z}_{(r)} \mid \boldsymbol{X}^h)}
$$

$$
+ \frac{1}{M}\sum_{m=1}^{M} \mathbb{E}_{q_{\phi_{\boldsymbol{w}_m}}(\boldsymbol{w}_m|\boldsymbol{x}_m^h)} \log \frac{p(\boldsymbol{w}_m)}{q_{\phi_{\boldsymbol{w}_m}}(\boldsymbol{w}_m \mid \boldsymbol{x}_m^h)} .
$$

Then, we can derive the evidence lower bound for $\log p_\Theta(\boldsymbol{X}^k, \boldsymbol{X}^l)$ by combining equations 28, 29 and 32:

$$
\log p_\Theta(\boldsymbol{X}^k, \boldsymbol{X}^l)
$$

$$
\geq \frac{1}{2} \sum_{h \in \{k,l\}} \Bigg[ \underbrace{\mathbb{E}_{q_{\Phi_t}(\boldsymbol{z}_{(t)}|\boldsymbol{X}^h)} \sum_{n=1}^{M} \log p_{\theta_n}(\boldsymbol{x}_n^h \mid \boldsymbol{z}_{(t)})}_{\text{self-sample reconstruction likelihood term}} + \underbrace{\mathbb{E}_{q_{\Phi_t}(\boldsymbol{z}_{(t)}|\boldsymbol{X}^h)} \sum_{n=1}^{M} \log p_{\theta_n}(\boldsymbol{x}_n^{\overline{h}} \mid \boldsymbol{z}_{(t)})}_{\text{similar sample reconstruction likelihood term}}
$$

$$
+ \underbrace{\mathbb{E}_{q_{\Phi_t}(\boldsymbol{z}_{(t)}|\boldsymbol{X}^h)} \log \frac{\sum_{\boldsymbol{c}} p(\boldsymbol{c}) p(\boldsymbol{z}_{(t)} \mid \boldsymbol{c})}{q_{\Phi_t}(\boldsymbol{z}_{(t)} \mid \boldsymbol{X}^k, \boldsymbol{X}^l)}}_{\text{Kullback-Leibler term}} \Bigg]
$$

$$
\geq \frac{1}{2} \sum_{h \in \{k,l\}} \Bigg[ \frac{1}{M} \sum_{m=1}^{M} \mathbb{E}_{\substack{q_{\phi_{\boldsymbol{z}_m}}(\boldsymbol{z}_{(t)}, \boldsymbol{z}_{(r)}|\boldsymbol{x}_m^h) \\ q_{\phi_{\boldsymbol{w}_m}}(\boldsymbol{w}_m|\boldsymbol{x}_m^h)}} \log p_{\theta_m}(\boldsymbol{x}_m^h \mid \boldsymbol{z}_{(t)}, \boldsymbol{z}_{(r)}, \boldsymbol{w}_m)
$$

$$
\hspace{3cm} (33)
$$

$$
+ \frac{1}{M} \sum_{\substack{m=1 \\ \{\widetilde{\boldsymbol{w}}_n \sim r_n(\boldsymbol{w}_n)\}_{n=1}^{M}}}^{M} \mathbb{E}_{q_{\phi_{\boldsymbol{z}_m}}(\boldsymbol{z}_{(t)}, \boldsymbol{z}_{(r)}|\boldsymbol{x}_m^h)} \sum_{n \neq m} \log p_{\theta_n}(\boldsymbol{x}_n^h \mid \boldsymbol{z}_{(t)}, \boldsymbol{z}_{(r)}, \widetilde{\boldsymbol{w}}_n)
$$

$$
+ \mathbb{E}_{\substack{q_{\Phi_t}(\boldsymbol{z}_{(t)}|\boldsymbol{X}^h) \\ \widetilde{\boldsymbol{z}}_{(r)} \sim f(\boldsymbol{z}_{(r)}) \\ \{\widetilde{\boldsymbol{w}}_n \sim r_n(\boldsymbol{w}_n)\}_{n=1}^{M}}} \sum_{n=1}^{M} \log p_{\theta_n}(\boldsymbol{x}_n^{\overline{h}} \mid \boldsymbol{z}_{(t)}, \widetilde{\boldsymbol{z}}_{(r)}, \widetilde{\boldsymbol{w}}_n)
$$

$$
+ \mathbb{E}_{q_{\Phi_t}(\boldsymbol{z}_{(t)}|\boldsymbol{X}^h)} \log \frac{\sum_{\boldsymbol{c}} p(\boldsymbol{c}) p(\boldsymbol{z}_{(t)} \mid \boldsymbol{c})}{q_{\Phi_t}(\boldsymbol{z}_{(t)} \mid \boldsymbol{X}^k, \boldsymbol{X}^l)} + \mathbb{E}_{q_{\Phi_t}(\boldsymbol{z}_{(r)}|\boldsymbol{X}^h)} \log \frac{p(\boldsymbol{z}_{(r)})}{q_{\Phi_r}(\boldsymbol{z}_{(r)} \mid \boldsymbol{X}^h)}
$$

$$
+ \frac{1}{M} \sum_{m=1}^{M} \mathbb{E}_{q_{\phi_{\boldsymbol{w}_m}}(\boldsymbol{w}_m|\boldsymbol{x}_m^h)} \log \frac{p(\boldsymbol{w}_m)}{q_{\phi_{\boldsymbol{w}_m}}(\boldsymbol{w}_m \mid \boldsymbol{x}_m^h)} \Bigg].
$$

$\square$

# E  THE EFFECT OF THE FIRST TERM IN $\mathcal{R}_{\mathrm{PS}}(\boldsymbol{X}^k, \boldsymbol{X}^l)$

In this section, we analyze the effect of the first term in $\mathcal{R}_{\mathrm{PS}}$ (Equation 4), defined as:

$$
\begin{aligned}
&\mathcal{L}_{\mathrm{ssr}}(\boldsymbol{X}^k, \boldsymbol{X}^l) \\
&= \frac{1}{2} \sum_{h \in \{k,l\}} \frac{1}{M} \sum_{m=1}^{M} \mathbb{E}_{\substack{q_{\phi_t}(\boldsymbol{z}_{(t)}|\boldsymbol{x}_m^h) \\ \widetilde{\boldsymbol{z}}_{(r)} \sim f(\boldsymbol{z}_{(r)}) \\ \{\widetilde{\boldsymbol{w}}_n \sim r_n(\boldsymbol{w}_n)\}_{n \neq m}}} \left[ \sum_{n=1}^{M} \log p_{\theta_n}(\boldsymbol{x}_n^{\overline{h}} \mid \boldsymbol{z}_{(t)}, \widetilde{\boldsymbol{z}}_{(r)}, \widetilde{\boldsymbol{w}}_m) \right],
\end{aligned}
\tag{34}
$$

Assuming $\boldsymbol{z}_{(t)}^k$ and $\boldsymbol{z}_{(t)}^l$ represent features capturing the similarity between similarity sample pair $(\boldsymbol{X}^k, \boldsymbol{X}^l)$, then $\boldsymbol{X}^{\overline{h}}$ and $\boldsymbol{z}_{(t)} \sim p(\boldsymbol{z}_{(t)} \mid \boldsymbol{X}^h)$ ($h \in \{k,l\}$) should exhibit high mutual information:

$$
\mathcal{I}(\boldsymbol{X}^{\overline{h}}; \boldsymbol{z}_{(t)}), \quad \boldsymbol{z}_{(t)} \sim p(\boldsymbol{z}_{(t)} \mid \boldsymbol{X}^h).
\tag{35}
$$

We can maximize this objective by maximizing its lower bound. Thus, we derive a lower bound for equation 36:

$$
\begin{aligned}
&\mathcal{I}(\boldsymbol{X}^{\overline{h}}; \boldsymbol{z}_{(t)}) \\
&= \mathbb{E}_{p_\Theta(\boldsymbol{X}^{\overline{h}}, \boldsymbol{z}_{(t)})} \log \frac{p_\Theta(\boldsymbol{X}^{\overline{h}} \mid \boldsymbol{z}_{(t)})}{p_\Theta(\boldsymbol{X}^{\overline{h}})} \\
&\approx \mathbb{E}_{\substack{p_\Theta(\boldsymbol{X}^{\overline{h}}) \\ q_{\Phi_t}(\boldsymbol{z}_{(t)}|\boldsymbol{X}^h)}} \log p_\Theta(\boldsymbol{X}^{\overline{h}} \mid \boldsymbol{z}_{(t)}) - \mathbb{E}_{p_\Theta(\boldsymbol{X}^{\overline{h}})} p_\Theta(\boldsymbol{X}^{\overline{h}}) \\
&= \mathbb{E}_{\substack{p_\Theta(\boldsymbol{X}^{\overline{h}}) \\ q_{\Phi_t}(\boldsymbol{z}_{(t)}|\boldsymbol{X}^h)}} \log p_\Theta(\boldsymbol{X}^{\overline{h}} \mid \boldsymbol{z}_{(t)}) - \boldsymbol{H}(\boldsymbol{X}^{\overline{h}}) \\
&\geq \mathbb{E}_{p_\Theta(\boldsymbol{X}^{\overline{h}})} \log p_\Theta(\boldsymbol{X}^{\overline{h}} \mid \boldsymbol{z}_{(t)}) \\
&= \mathbb{E}_{p_\Theta(\boldsymbol{X}^{\overline{h}})} \left[ \mathbb{E}_{q_{\Phi_t}(\boldsymbol{z}_{(t)}|\boldsymbol{X}^h)} \log p_\Theta(\boldsymbol{X}^{\overline{h}} \mid \boldsymbol{z}_{(t)}) \right].
\end{aligned}
\tag{36}
$$

The approximately equal employs the relation:

$$
p_\Theta(\boldsymbol{X}^{\overline{h}}, \boldsymbol{z}_{(t)}) = p_\Theta(\boldsymbol{X}^{\overline{h}}) \cdot p_\Theta(\boldsymbol{z}_{(t)} \mid \boldsymbol{X}^h) \approx p_\Theta(\boldsymbol{X}^{\overline{h}}) \cdot q_{\Phi_t}(\boldsymbol{z}_{(t)} \mid \boldsymbol{X}^h).
\tag{37}
$$

Critically, note that $p_\Theta(\boldsymbol{X}^{\overline{h}} \mid \boldsymbol{z}_{(t)})$ in $\log p_\Theta(\boldsymbol{X}^{\overline{h}} \mid \boldsymbol{z}_{(t)})$ must be interpreted as the function $p_\Theta(\boldsymbol{X} \mid \boldsymbol{z}_{(t)})$ evaluated at $\boldsymbol{X} = \boldsymbol{X}^{\overline{h}}$, denoted as $p_\Theta(\boldsymbol{X} \mid \boldsymbol{z}_{(t)})|_{\boldsymbol{X}=\boldsymbol{X}^{\overline{h}}}$. Consequently, the term $\log p_\Theta(\boldsymbol{X}^{\overline{h}} \mid \boldsymbol{z}_{(t)})$ cannot be decomposed by separating $\boldsymbol{X}^k$ and $\boldsymbol{z}_{(t)}$ as done in expectation operations.

From equation 29, we derive:

$$
\begin{aligned}
&\mathcal{I}(\boldsymbol{X}^{\overline{h}}; \boldsymbol{z}_{(t)}) \\
&\geq \mathbb{E}_{p_\Theta(\boldsymbol{X}^{\overline{h}})} \left[ \mathbb{E}_{q_{\Phi_t}(\boldsymbol{z}_{(t)}|\boldsymbol{X}^h)} \log p_\Theta(\boldsymbol{X}^{\overline{h}} \mid \boldsymbol{z}_{(t)}) \right] \\
&= \mathbb{E}_{p_\Theta(\boldsymbol{X}^{\overline{h}})} \left[ \mathbb{E}_{\substack{q_{\Phi_t}(\boldsymbol{z}_{(t)}|\boldsymbol{X}^h) \\ \widetilde{\boldsymbol{z}}_{(r)} \sim f(\boldsymbol{z}_{(r)}) \\ \{\widetilde{\boldsymbol{w}}_n \sim r_n(\boldsymbol{w}_n)\}_{n=1}^{M}}} \sum_{n=1}^{M} \log p_{\theta_n}(\boldsymbol{x}_n^{\overline{h}} \mid \boldsymbol{z}_{(t)}, \widetilde{\boldsymbol{z}}_{(r)}, \widetilde{\boldsymbol{w}}_n) \right] \\
&= \mathbb{E}_{p_\Theta(\boldsymbol{X}^{\overline{h}})} \left[ \frac{1}{M} \sum_{m=1}^{M} \mathbb{E}_{\substack{q_{\phi_t}(\boldsymbol{z}_{(t)}|\boldsymbol{x}_m^h) \\ \widetilde{\boldsymbol{z}}_{(r)} \sim f(\boldsymbol{z}_{(r)}) \\ \{\widetilde{\boldsymbol{w}}_n \sim r_n(\boldsymbol{w}_n)\}_{n \neq m}}} \left[ \sum_{n=1}^{M} \log p_{\theta_n}(\boldsymbol{x}_n^{\overline{h}} \mid \boldsymbol{z}_{(t)}, \widetilde{\boldsymbol{z}}_{(r)}, \widetilde{\boldsymbol{w}}_m) \right] \right].
\end{aligned}
\tag{38}
$$

Consequently, maximizing

$$\frac{1}{M} \sum_{m=1}^{M} \mathbb{E}_{\substack{q_{\phi_t}(\boldsymbol{z}_{(t)}|\boldsymbol{x}_m^h) \\ \widetilde{\boldsymbol{z}}_{(r)} \sim f(\boldsymbol{z}_{(r)}) \\ \{\widetilde{\boldsymbol{w}}_n \sim r_n(\boldsymbol{w}_n)\}_{n \neq m}}} \left[ \sum_{n=1}^{M} \log p_{\theta_n}(\boldsymbol{x}_n^{\overline{h}} \mid \boldsymbol{z}_{(t)}, \widetilde{\boldsymbol{z}}_{(r)}, \widetilde{\boldsymbol{w}}_m) \right] \tag{39}$$

maximizes $\mathcal{I}(\boldsymbol{X}^{\overline{h}}; \boldsymbol{z}_{(t)})$, $\boldsymbol{z}_{(t)} \sim p(\boldsymbol{z}_{(t)} \mid \boldsymbol{X}^h)$. Recalling the objective $\mathcal{L}_{\mathrm{ssr}}(\boldsymbol{X}^k, \boldsymbol{X}^l)$, we establish that maximizing $\mathcal{L}_{\mathrm{ssr}}(\boldsymbol{X}^k, \boldsymbol{X}^l)$ maximizes

$$\frac{1}{2}(\mathcal{I}(\boldsymbol{X}^k; \boldsymbol{z}_{(t)}^l) + \mathcal{I}(\boldsymbol{X}^l; \boldsymbol{z}_{(t)}^k)). \tag{40}$$

This reaffirms that $\mathcal{R}_{\mathrm{PS}}$ facilitates the extraction of similarity information between positive sample pairs.

# F ANALYSIS OF $\mathcal{I}_{\mathrm{CCMI}}(\boldsymbol{z}_{(t)}^i; \boldsymbol{z}_{(t)}^j)$

**Definition 1** (Common-Cause Mutual Information (CCMI)). *Assume that $\boldsymbol{z}_{(t)}^i$ and $\boldsymbol{z}_{(t)}^j$ are two observations, which are generated conditional on $\boldsymbol{c}^i$ and $\boldsymbol{c}^j$ respectively, where $\boldsymbol{c}^i$ and $\boldsymbol{c}^j$ are derived from the same underlying semantic space. Then, we define the CCMI between $\boldsymbol{z}_{(t)}^i$ and $\boldsymbol{z}_{(t)}^j$ as*

$$\mathcal{I}_{CCMI}(\boldsymbol{z}_{(t)}^i; \boldsymbol{z}_{(t)}^j) = \mathbb{E}_{p(\boldsymbol{z}_{(t)}^i)p(\boldsymbol{z}_{(t)}^j)} \left[ \log \frac{\mathbb{E}_{\boldsymbol{c}}\left[p(\boldsymbol{z}_{(t)}^i \mid \boldsymbol{c}) \cdot p(\boldsymbol{c} \mid \boldsymbol{z}_{(t)}^j)\right]}{\mathbb{E}_{\boldsymbol{c}}\left[p(\boldsymbol{z}_{(t)}^i \mid \boldsymbol{c})\right]} \right]. \tag{41}$$

## F.1 SYMMETRY

At first, $\mathcal{I}_{\mathrm{CCMI}}(\boldsymbol{z}_{(t)}^i; \boldsymbol{z}_{(t)}^j)$ and $\mathcal{R}_{\mathrm{NS}}(\boldsymbol{X}^i, \boldsymbol{X}^j)$ exhibits symmetry. Based on Bayes' formula:

$$p(\boldsymbol{c} \mid \boldsymbol{z}_{(t)}^j) = \frac{p(\boldsymbol{z}_{(t)}^i \mid \boldsymbol{c}) \cdot p(\boldsymbol{c})}{\mathbb{E}_{\boldsymbol{c}} p(\boldsymbol{z}_{(t)}^i \mid \boldsymbol{c})} \tag{42}$$

we can reformulate both $\mathcal{I}_{\mathrm{CCMI}}(\boldsymbol{z}_{(t)}^i; \boldsymbol{z}_{(t)}^j)$ and $\mathcal{R}_{\mathrm{NS}}(\boldsymbol{X}^i, \boldsymbol{X}^j)$ as follows:

$$
\begin{aligned}
\mathcal{I}_{\mathrm{CCMI}}(\boldsymbol{z}_{(t)}^i; \boldsymbol{z}_{(t)}^j) &= \mathbb{E}_{p(\boldsymbol{z}_{(t)}^i)p(\boldsymbol{z}_{(t)}^j)} \left[ \log \frac{\mathbb{E}_{\boldsymbol{c}}\left[p(\boldsymbol{z}_{(t)}^i \mid \boldsymbol{c}) \cdot p(\boldsymbol{c} \mid \boldsymbol{z}_{(t)}^j)\right]}{\mathbb{E}_{\boldsymbol{c}}\left[p(\boldsymbol{z}_{(t)}^i \mid \boldsymbol{c})\right]} \right] \\
&= \mathbb{E}_{p(\boldsymbol{z}_{(t)}^i)p(\boldsymbol{z}_{(t)}^j)} \left[ \log \frac{\mathbb{E}_{\boldsymbol{c}}\left[p(\boldsymbol{z}_{(t)}^i \mid \boldsymbol{c}) \cdot p(\boldsymbol{z}_{(t)}^j \mid \boldsymbol{c}) \cdot p(\boldsymbol{c})\right]}{\mathbb{E}_{\boldsymbol{c}}\left[p(\boldsymbol{z}_{(t)}^i \mid \boldsymbol{c})\right] \cdot \mathbb{E}_{\boldsymbol{c}}\left[p(\boldsymbol{z}_{(t)}^j \mid \boldsymbol{c})\right]} \right].
\end{aligned}
\tag{43}
$$

$$
\begin{aligned}
\mathcal{R}_{\mathrm{NS}}(\boldsymbol{X}^i, \boldsymbol{X}^j) &= \mathbb{E}_{\substack{\boldsymbol{z}_{(t)}^i \sim q_{\Phi_t}(\boldsymbol{z}_{(t)}|X^i) \\ \boldsymbol{z}_{(t)}^j \sim q_{\Phi_t}(\boldsymbol{z}_{(t)}|X^j)}} \log \left[ 1 - \frac{\mathbb{E}_{\boldsymbol{c}}\left[p(\boldsymbol{z}_{(t)}^i \mid \boldsymbol{c}) \cdot p(\boldsymbol{c} \mid \boldsymbol{z}_{(t)}^j)\right]}{\mathbb{E}_{\boldsymbol{c}}\left[p(\boldsymbol{z}_{(t)}^i \mid \boldsymbol{c})\right]} \right] \\
&= \mathbb{E}_{\substack{\boldsymbol{z}_{(t)}^i \sim q_{\Phi_t}(\boldsymbol{z}_{(t)}|X^i) \\ \boldsymbol{z}_{(t)}^j \sim q_{\Phi_t}(\boldsymbol{z}_{(t)}|X^j)}} \log \left[ 1 - \frac{\mathbb{E}_{\boldsymbol{c}}\left[p(\boldsymbol{z}_{(t)}^i \mid \boldsymbol{c}) \cdot p(\boldsymbol{z}_{(t)}^j \mid \boldsymbol{c}) \cdot p(\boldsymbol{c})\right]}{\mathbb{E}_{\boldsymbol{c}}\left[p(\boldsymbol{z}_{(t)}^i \mid \boldsymbol{c})\right] \cdot \mathbb{E}_{\boldsymbol{c}}\left[p(\boldsymbol{z}_{(t)}^j \mid \boldsymbol{c})\right]} \right]
\end{aligned}
\tag{44}
$$

Therefore, $\mathcal{I}_{\mathrm{CCMI}}(\boldsymbol{z}_{(t)}^i; \boldsymbol{z}_{(t)}^j) = \mathcal{I}_{\mathrm{CCMI}}(\boldsymbol{z}_{(t)}^j; \boldsymbol{z}_{(t)}^i)$ and $\mathcal{R}_{\mathrm{NS}}(\boldsymbol{X}^i, \boldsymbol{X}^j) = \mathcal{R}_{\mathrm{NS}}(\boldsymbol{X}^j, \boldsymbol{X}^i)$.

## F.2 A MORE INTUITIVE SEMANTIC INTERPRETATION FOR $\mathcal{I}_{\mathrm{CCMI}}(\boldsymbol{z}_{(t)}^i; \boldsymbol{z}_{(t)}^j)$

As established in Section 2.2.2, a higher $\mathcal{I}_{\mathrm{CCMI}}(\boldsymbol{z}_{(t)}^i; \boldsymbol{z}_{(t)}^j)$ value indicates stronger evidence that $\boldsymbol{z}_{(t)}^i$ and $\boldsymbol{z}_{(t)}^j$ encode the same underlying semantic $\boldsymbol{c}$; this relationship becomes particularly clear when $\boldsymbol{c}$ is modeled as a discrete random variable. In this section, we conduct a comprehensive analysis of CCMI and provide its formal semantic interpretation.

*When $\boldsymbol{c}$ is a continuous random variable.* Based on equation 43 and Bayes' formula, $\mathcal{I}_{\text{CCMI}}$ can be further developed as:

$$
\begin{aligned}
\mathcal{I}_{\text{CCMI}}(\boldsymbol{z}_{(t)}^i; \boldsymbol{z}_{(t)}^j) &= \mathbb{E}_{p(\boldsymbol{z}_{(t)}^i)p(\boldsymbol{z}_{(t)}^j)} \left[ \log \frac{\mathbb{E}_{\boldsymbol{c}} \left[ p(\boldsymbol{z}_{(t)}^i \mid \boldsymbol{c}) \cdot p(\boldsymbol{z}_{(t)}^j \mid \boldsymbol{c}) \cdot p(\boldsymbol{c}) \right]}{\mathbb{E}_{\boldsymbol{c}} \left[ p(\boldsymbol{z}_{(t)}^i \mid \boldsymbol{c}) \right] \cdot \mathbb{E}_{\boldsymbol{c}} \left[ p(\boldsymbol{z}_{(t)}^j \mid \boldsymbol{c}) \right]} \right] \\
&= \mathbb{E}_{p(\boldsymbol{z}_{(t)}^i)p(\boldsymbol{z}_{(t)}^j)} \left[ \log \frac{\int_{\boldsymbol{c}} p(\boldsymbol{z}_{(t)}^i \mid \boldsymbol{c}) \cdot p(\boldsymbol{z}_{(t)}^j \mid \boldsymbol{c}) \cdot p(\boldsymbol{c}) \cdot p(\boldsymbol{c}) d\boldsymbol{c}}{\mathbb{E}_{\boldsymbol{c}} \left[ p(\boldsymbol{z}_{(t)}^i \mid \boldsymbol{c}) \right] \cdot \mathbb{E}_{\boldsymbol{c}} \left[ p(\boldsymbol{z}_{(t)}^j \mid \boldsymbol{c}) \right]} \right] \\
&= \mathbb{E}_{p(\boldsymbol{z}_{(t)}^i)p(\boldsymbol{z}_{(t)}^j)} \log \int_{\boldsymbol{c}} \frac{p(\boldsymbol{z}_{(t)}^i \mid \boldsymbol{c}) \cdot p(\boldsymbol{c})}{\mathbb{E}_{\boldsymbol{c}^i} \left[ p(\boldsymbol{z}_{(t)}^i \mid \boldsymbol{c}^i) \right]} \cdot \frac{p(\boldsymbol{z}_{(t)}^j \mid \boldsymbol{c}) \cdot p(\boldsymbol{c})}{\mathbb{E}_{\boldsymbol{c}^j} \left[ p(\boldsymbol{z}_{(t)}^j \mid \boldsymbol{c}^j) \right]} d\boldsymbol{c} \\
&= \mathbb{E}_{p(\boldsymbol{z}_{(t)}^i)p(\boldsymbol{z}_{(t)}^j)} \log \int_{\boldsymbol{c}} p(\boldsymbol{c} \mid \boldsymbol{z}_{(t)}^i) \cdot p(\boldsymbol{c} \mid \boldsymbol{z}_{(t)}^j) d\boldsymbol{c} \\
&= \mathbb{E}_{p(\boldsymbol{z}_{(t)}^i)p(\boldsymbol{z}_{(t)}^j)} \log P(\boldsymbol{c}^i = \boldsymbol{c}^j \mid \boldsymbol{z}_{(t)}^i, \boldsymbol{z}_{(t)}^j),
\end{aligned}
\tag{45}
$$

where $P(\boldsymbol{c}^i = \boldsymbol{c}^j \mid \boldsymbol{z}_{(t)}^i, \boldsymbol{z}_{(t)}^j)$ denotes the conditional probability that the underlying semantic $\boldsymbol{c}^i$ and underlying semantic $\boldsymbol{c}^j$ share identical values, given the observed representations $\boldsymbol{z}_{(t)}^i$ and $\boldsymbol{z}_{(t)}^j$.

*When $\boldsymbol{c}$ is a discrete random variable.* Based on equation 43 and Bayes' formula, $\mathcal{I}_{\text{CCMI}}$ can be further developed as:

$$
\begin{aligned}
\mathcal{I}_{\text{CCMI}}(\boldsymbol{z}_{(t)}^i; \boldsymbol{z}_{(t)}^j) &= \mathbb{E}_{p(\boldsymbol{z}_{(t)}^i)p(\boldsymbol{z}_{(t)}^j)} \left[ \log \frac{\mathbb{E}_{\boldsymbol{c}} \left[ p(\boldsymbol{z}_{(t)}^i \mid \boldsymbol{c}) \cdot p(\boldsymbol{z}_{(t)}^j \mid \boldsymbol{c}) \cdot p(\boldsymbol{c}) \right]}{\mathbb{E}_{\boldsymbol{c}} \left[ p(\boldsymbol{z}_{(t)}^i \mid \boldsymbol{c}) \right] \cdot \mathbb{E}_{\boldsymbol{c}} \left[ p(\boldsymbol{z}_{(t)}^j \mid \boldsymbol{c}) \right]} \right] \\
&= \mathbb{E}_{p(\boldsymbol{z}_{(t)}^i)p(\boldsymbol{z}_{(t)}^j)} \left[ \log \frac{\sum_{\boldsymbol{c}} p(\boldsymbol{z}_{(t)}^i \mid \boldsymbol{c}) \cdot p(\boldsymbol{z}_{(t)}^j \mid \boldsymbol{c}) \cdot p(\boldsymbol{c}) \cdot p(\boldsymbol{c})}{\mathbb{E}_{\boldsymbol{c}} \left[ p(\boldsymbol{z}_{(t)}^i \mid \boldsymbol{c}) \right] \cdot \mathbb{E}_{\boldsymbol{c}} \left[ p(\boldsymbol{z}_{(t)}^j \mid \boldsymbol{c}) \right]} \right] \\
&= \mathbb{E}_{p(\boldsymbol{z}_{(t)}^i)p(\boldsymbol{z}_{(t)}^j)} \log \sum_{\boldsymbol{c}} \frac{p(\boldsymbol{z}_{(t)}^i \mid \boldsymbol{c}) \cdot p(\boldsymbol{c})}{\mathbb{E}_{\boldsymbol{c}^i} \left[ p(\boldsymbol{z}_{(t)}^i \mid \boldsymbol{c}^i) \right]} \cdot \frac{p(\boldsymbol{z}_{(t)}^j \mid \boldsymbol{c}) \cdot p(\boldsymbol{c})}{\mathbb{E}_{\boldsymbol{c}^j} \left[ p(\boldsymbol{z}_{(t)}^j \mid \boldsymbol{c}^j) \right]} \\
&= \mathbb{E}_{p(\boldsymbol{z}_{(t)}^i)p(\boldsymbol{z}_{(t)}^j)} \log \sum_{\boldsymbol{c}} p(\boldsymbol{c} \mid \boldsymbol{z}_{(t)}^i) \cdot p(\boldsymbol{c} \mid \boldsymbol{z}_{(t)}^j) \\
&= \mathbb{E}_{p(\boldsymbol{z}_{(t)}^i)p(\boldsymbol{z}_{(t)}^j)} \log P(\boldsymbol{c}^i = \boldsymbol{c}^j \mid \boldsymbol{z}_{(t)}^i, \boldsymbol{z}_{(t)}^j),
\end{aligned}
\tag{46}
$$

where $P(\boldsymbol{c}^i = \boldsymbol{c}^j \mid \boldsymbol{z}_{(t)}^i, \boldsymbol{z}_{(t)}^j)$ denotes the conditional probability that the underlying semantic $\boldsymbol{c}^i$ and underlying semantic $\boldsymbol{c}^j$ share identical values, given the observed representations $\boldsymbol{z}_{(t)}^i$ and $\boldsymbol{z}_{(t)}^j$.

In summary, $\mathcal{I}_{\text{CCMI}}(\boldsymbol{z}_{(t)}^i; \boldsymbol{z}_{(t)}^j)$ serves as a robust indicator of the capability of $\boldsymbol{z}_{(t)}^i$ and $\boldsymbol{z}_{(t)}^j$ to encode identical underlying semantics $\boldsymbol{c}$, regardless of whether $\boldsymbol{c}$ is modeled as a continuous or discrete random variable.

## G DETAILS ON DATASETS, BASELINES AND IMPLEMENTATION

### G.1 DATASETS

In this section, we provide comprehensive details regarding the three datasets employed in our study.

The first dataset is the **Double-Digit MNIST MultiModal Dataset** (DDMNISTMM), a semi-synthetic yet challenging benchmark specifically designed for multimodal disentanglement learning and representation learning. DDMNISTMM comprises three distinct modalities, each depicting two completely independent MNIST digits (left/right position). These digits are systematically patched onto randomly cropped regions derived from three unique background images (one per modality). As illustrated in Figure 4, while the digit labels (left/right) are shared across modalities in DDMNISTMM, the actual digit instances are sampled independently within each digit class, and background crops originate from different source images. Consequently, the left and right digit labels constitute the only inter-modal shared information, whereas both background content and handwriting characteristics exhibit modality-specific variations across data points. The formal synthesis procedure for DDMNISTMM is detailed in Algorithm 1.

---

**Algorithm 1** DDMNISTMM Synthesis Protocol

**Require:** MNIST dataset, target sample size $N$, background images $\{img_1, img_2, img_3\}$
**Ensure:** Generated DDMNISTMM dataset

1: **for** $n = 1$ **to** $N$ **do**
2:     **for** $m = 1$ **to** $3$ **do**
3:         Sample $label_{\text{left}}, label_{\text{right}} \sim \mathcal{U}\{0, 9\}$ with replacement
4:         Select $d_{\text{left}}$ randomly from MNIST class $label_{\text{left}}$
5:         Select $d_{\text{right}}$ randomly from MNIST class $label_{\text{right}}$
6:         Crop $56 \times 56$ sub-image $sub\_img_m$ from $img_m$ at random position
7:         Generate position masks $mask_{\text{left}}, mask_{\text{right}}$ from $d_{\text{left}}, d_{\text{right}}$
8:         Apply color inversion to left/right regions of $sub\_img_m$ using $mask_{\text{left}}, mask_{\text{right}}$
9:     **end for**
10: **end for**

---

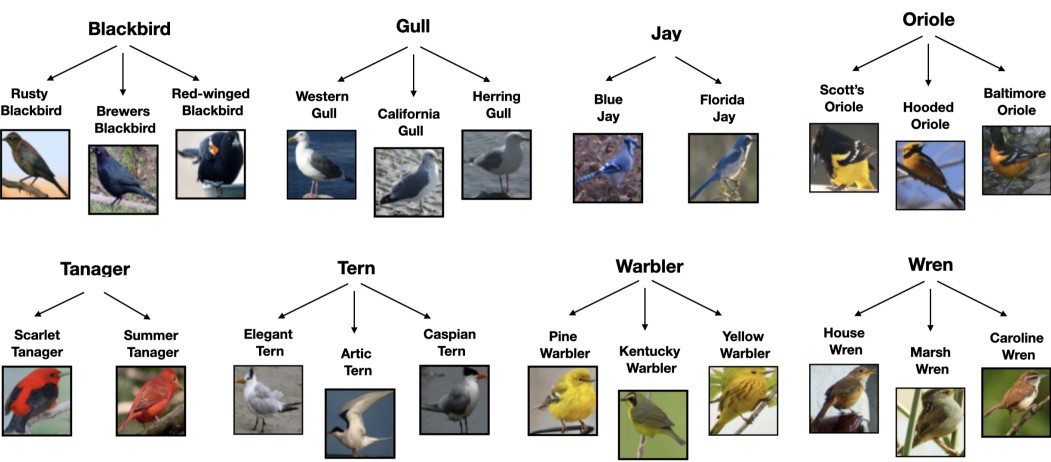

Figure 9: CUBICC dataset structure. Bird sub-species are grouped together in a single category. Images taken from Palumbo et al. (2024).

The second dataset, **CUB Image-Captions for Clustering** (CUBICC) dataset (Palumbo et al. (2024)), constitutes a modified variant of the CUB Image-Captions dataset (Wah et al. (2011); Shi et al. (2019)). The original CUB Image-Captions dataset contains fine-grained bird images paired with descriptive captions. CUBICC was specifically developed to evaluate multimodal clustering methods under more realistic conditions. As illustrated in Figure 9, this modification employs hierarchical

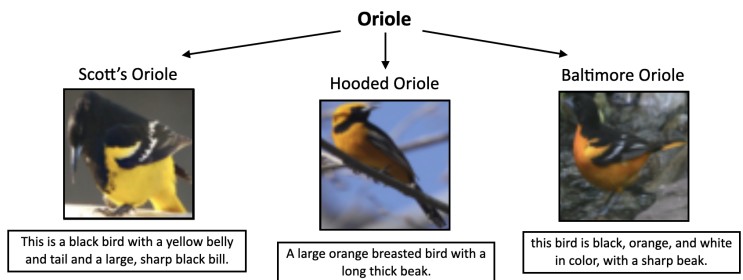

Figure 10: Illustrative samples for the CUBICC: Example of the variability within a single bird class, Oriole. Here we show the images (modality 1) with their corresponding captions (modality 2). Images taken from Palumbo et al. (2024).

grouping of avian subspecies into consolidated species categories, resulting in 8 broad taxonomic classes. This grouping strategy significantly amplifies intra-class variability by incorporating diverse morphological variations within each species cluster, thereby presenting substantially greater challenges for clustering algorithms. Figure 10 exemplifies this characteristic variability through a representative sampling within the *Oriole* bird class.

To further validate DL$^2$'s capacity for targeted semantic disentanglement, we introduce CelebA-HQ (Lee et al. (2020)) as the third benchmark dataset. CelebA-HQ comprises dual modalities: facial images paired with corresponding textual descriptions. Notably, the dataset provides annotations for multiple attributes, each representing distinct high-level semantic concepts. Specifically, we conduct experiments using two attribute labels: *gender* (binary) and *smile intensity* (6-point scale). By evaluating clustering performance across these disparate semantic targets, we establish a robust assessment framework for DL$^2$'s ability to disentangle specified high-level semantics. Representative samples under both attribute conditions are visualized in Figure 11.

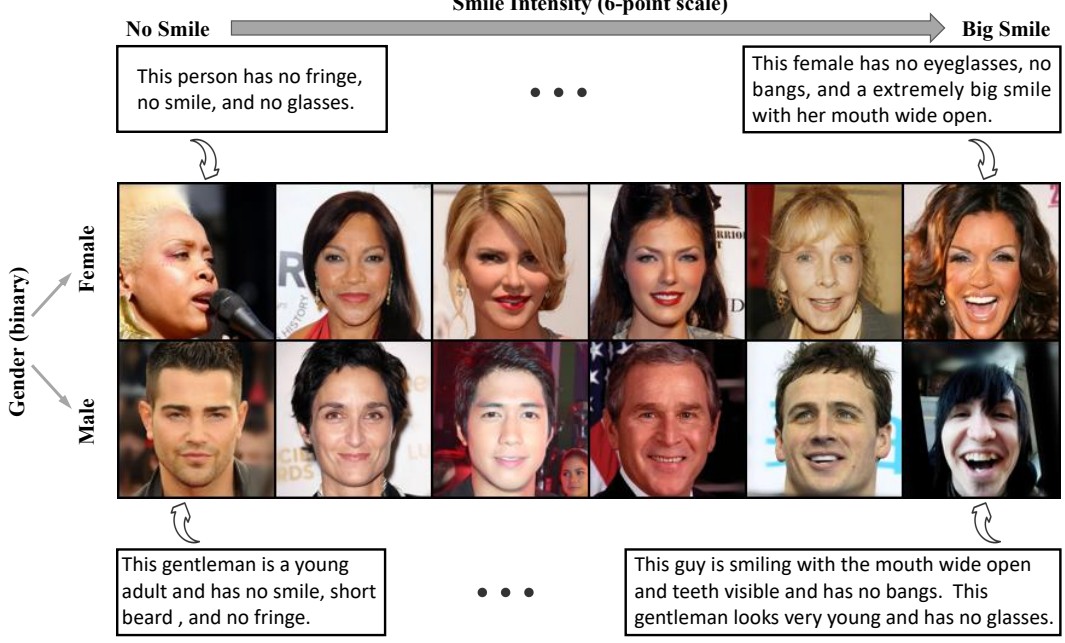

Figure 11: Illustrative samples for the CelebA-HQ: The first row illustrates six female-labeled samples, while the second row displays six male-labeled samples. Each column corresponds to a distinct smile intensity level, with expression magnitude progressively intensifying from left to right (i.e., spanning *no smile* to *big smile*).

## G.2 BASELINES

We compare our approach with the following state-of-the-art deep clustering methods, which cover a wide range of settings including single-modal and multi-modal, as well as unsupervised and weakly-supervised paradigms.

Single-Modal Methods:

- **VaDE (Jiang et al. (2016)).** An unsupervised deep clustering method based on the Variational Autoencoder (VAE) framework, which models the latent variables with a Gaussian Mixture Model (GMM) prior.

- **DC-GMM (Manduchi et al. (2021)).** A weakly-supervised deep clustering method also built upon the VAE framework. It introduces a Conditional Gaussian Mixture prior for the latent variables that is guided by pairwise constraints, integrating domain knowledge in a probabilistic generative manner.

- **SDEC (Ren et al. (2019)).** A weakly-supervised deep clustering method based on an Autoencoder (AE) architecture. It enhances the Deep Embedded Clustering (DEC) framework by integrating pairwise constraints to simultaneously learn feature representations and perform cluster assignments.

- **C-IDEC (Zhang et al. (2019)).** A weakly-supervised deep clustering framework based on an Autoencoder (AE) structure. It introduces a flexible constraint framework to incorporate various forms of prior knowledge, such as pairwise constraints, instance difficulty constraints, and triplet constraints.

- **VolMaxDCC (Nguyen et al. (2023)).** A weakly-supervised deep clustering method with a unique encoder-only architecture (i.e., no decoder). It learns discriminative representations by maximizing the volume of the latent space through a geometric regularization approach, specifically designed to handle incomplete and noisy pairwise annotations in a provably identifiable manner.

Multi-Modal Method:

- **CMVAE (Palumbo et al. (2024)).** An unsupervised multi-modal clustering method based on the VAE framework, designed to learn a common latent representation from multiple modalities for clustering.

- **MMVAE+ (Palumbo et al. (2023)).** A weakly-supervised multimodal generative model based on the Mixture-of-Experts Multimodal VAE (MMVAE) framework. It enhances generative quality by modeling shared and modality-specific information in separate latent subspaces, while maintaining high semantic coherence through a novel objective that uses auxiliary priors to avoid shortcuts during cross-modal reconstruction.

## G.3 IMPLEMENTATION DETAILS

**DDMNISTMM (results in Section 4.1 and Appendix G)**     MNIST

For all datasets, we employed identical encoder and decoder networks for all comparative methods (CMVAE (Palumbo et al. (2024)), MMVAE+ (Palumbo et al. (2023)), VaDE (Jiang et al. (2016)), DC-GMM (Manduchi et al. (2021)), SDEC (Ren et al. (2019)), and C-IDEC (Zhang et al. (2019))) based on the encoder-decoder architecture. For VolMaxDCC (Nguyen et al. (2023)), which only requires projecting samples into the category space, the same encoder followed by a two-layer MLP was used during training to project samples first into a feature space and then into the category space.

**DDMNISTMM**     On the DDMNISTMM dataset, the dimensionalities of DL2's task-relevant shared latent subspace, residual shared latent subspace, and modality-specific latent subspaces were set to 32, 32, and 64, respectively. For CMVAE and MMVAE+, the shared and modality-specific latent subspace dimensions were set to 64 and 64. The latent space dimensionality for VaDE, DC-GMM, SDEC, and C-IDEC was set to 128, while VolMaxDCC's feature space dimensionality was set to 128 to ensure a fair comparison. All methods were trained for 250 epochs with a learning rate of 5e-4 on DDMNISTMM, following the best practices from their original works but without any pretraining (e.g., for VaDE and SDEC). Specifically for SDEC, the reconstruction loss from its pretrained feature extraction encoder was incorporated into the training objective for end-to-end

training. For VAE-based methods, the ELBO was estimated using the IWAE estimator with one sample.

**CUBICC** On the CUBICC dataset, DL2 used ResNet and convolutional encoder/decoder networks for image and text modalities, respectively. The dimensionalities of its task-relevant shared, residual shared, and modality-specific latent subspaces were set to 48, 16, and 32. For CMVAE and MMVAE+, the corresponding dimensions were 64 and 32; for VaDE, DC-GMM, SDEC, and C-IDEC, the latent dimensionality was 96; and for VolMaxDCC, the feature space dimensionality was 96. All methods were trained for 150 epochs with a learning rate of 1e-3, again without pretraining. On the CUBICC dataset, we followed the practice of Palumbo et al. (2024). for training VAE-based methods. Specifically, the loss was estimated using 10 samples, and the DReG estimator (Tucker et al. (2018)) was applied for gradient calculation.

**CelebA-HQ** The training setup for the CelebA-HQ dataset was largely consistent with that for CUBICC. For the smile intensity discrimination task, DL2's task-relevant shared, residual shared, and modality-specific latent subspace dimensions were set to 32, 32, and 32; for the gender discrimination task, they were set to 48, 16, and 32. The comparative methods used the same dimensionalities as on CUBICC. All models were trained for 250 epochs with a learning rate of 1e-3. For VAE-based methods, the IWAE estimator with one sample was used.

Finally, for DDMNISTMM, the dataset is partitioned into 60,000 training samples, 5,000 validation samples, and 5,000 test samples. For CUBICC, it comprises 11,834 training, 638 validation, and 659 test samples. For CelebA-HQ, the dataset consists of 20,999 training, 3,000 validation, and 6,000 test samples.

Note that during training on all datasets, the correct number of clusters was provided. Consequently, the results for CMVAE on CUBICC differ from those reported in the original paper.

## H  Evaluation Metrics

In this work, we employ a comprehensive set of metrics to evaluate the performance of our model from both generative and clustering perspectives. Below we provide detailed definitions and formulations for each metric.

### H.1  Evaluation Metrics of Generative capacity

First, regarding generative capabilities, we focus on both generation quality and generation consistency. This is because an effective MVAE should not only possess the ability to produce diverse samples but also ensure high semantic consistency across different modalities.

#### H.1.1  Fréchet Inception Distance (FID)

The FID score measures the similarity between the distribution of generated images and the distribution of real images (Heusel et al. (2017)). It is computed by modeling the feature representations of both sets of images using a pre-trained Inception-v3 network (Szegedy et al. (2016)). A lower FID score indicates higher quality and diversity of the generated images, meaning the generated distribution is closer to the real data distribution (Benny et al. (2021)).

Let $\mu_r$, $\mu_g$ be the mean feature vectors, and $\Sigma_r$, $\Sigma_g$ be the covariance matrices of the real and generated data distributions, respectively. The FID is calculated asHeusel et al. (2017):

$$\text{FID} = \|\mu_r - \mu_g\|_2^2 + \text{Tr}(\Sigma_r + \Sigma_g - 2(\Sigma_r \Sigma_g)^{1/2})$$

where Tr denotes the trace of a matrix.

#### H.1.2  Generative Coherence

To quantitatively evaluate generative coherence across the two multimodal datasets (DDMNISTMM and CelebA) used in our study, we adopt task-specific coherence metrics that align with the semantic structure of each dataset. Specifically, we employ modality-specific classifiers trained to identify shared semantic content. This approach is consistent with conventional practices in multimodal generative modeling (e.g., Shi et al. (2019); Palumbo et al. (2023; 2024)), which utilize classifiers as proxies for assessing semantic preservation during generation.

**Generative coherence: DDMNISTMM**    To quantitatively evaluate generative coherence on the DDMNISTMM dataset, we adopt a strategy that aligns with the structure of the data. Since each modality depicts two independent MNIST digits (left/right) against modality-specific backgrounds, and the digit labels represent the shared information across modalities, we train six separate classifiers—one for each digit position (left or right) within each modality. All classifiers achieve an accuracy exceeding 98.97% on the test set, ensuring reliable assessment. For cross-modal generation, we compute the coherence for left and right digits separately by comparing the classifier-predicted labels with the ground-truth labels. The overall shared consistency is then defined as the product of the left and right coherence scores. For unconditional generation, we generate samples for all modalities and apply the corresponding classifiers to verify whether the predicted digit labels are consistent across modalities.

**Generative coherence: CelebA**    In the CelebA dataset, we focus on two attributes: gender (binary) and smile intensity (6-point scale). We train four classifiers in total: for each modality (image and text), we train one classifier for gender and one for smile intensity. The classifiers achieve accuracies of 100% (text-gender), 100% (text-smile), 97.82% (image-gender), and 83.17% (image-smile) on the test set. For cross-modal generation, we compute attribute-level coherence by comparing the classifier predictions for the generated sample with the true labels. The overall coherence is defined as the product of gender coherence and smile coherence. For unconditional generation, we check whether the predictions for both attributes are consistent across modalities.

**Rationale and Validity**    This evaluation strategy is designed to ensure a rigorous and semantically meaningful assessment of generative coherence. By leveraging highly accurate modality-specific classifiers, we establish a reliable proxy for measuring whether the generated content preserves the underlying shared information—be it digit labels or semantic attributes. The use of separate classifiers for distinct components (e.g., left/right digits in DDMNISTMM) or attributes (e.g., gender and

smile in CelebA) allows for fine-grained and interpretable evaluation. Moreover, the multiplicative aggregation of coherence scores emphasizes the necessity of holistic consistency across all relevant factors. This approach is particularly suitable for evaluating multimodal generative models where high-dimensional raw data (e.g., images or text) must be evaluated in terms of discrete semantic concepts.

**Remark 3** (For the CUBICC dataset, this paper does not calculate the generative coherence). *On the DDMNISTMM dataset, we selected the left-digit labels as the clustering target. Thus, the task-relevant information corresponds to the left digits, while the residual shared information corresponds to the right digits. On the CelebA-HQ dataset, when smiling intensity was chosen as the target, the task-relevant information captured smiling intensity, and the residual information contained other attributes. Although we lack complete labels for the residual information, gender is included within it, so we used gender as a proxy. Similarly, when gender was selected as the target, the residual information contained attributes such as smiling intensity, which served as the proxy. In contrast, for the CUBICC dataset, we completely lack annotated labels for the residual shared information. Therefore, consistency measurements were omitted on this dataset.*

### H.2 EVALUATION METRICS OF CLUSTERING PERFORMANCE

To quantitatively assess the clustering performance, we adopt three widely-used metrics: Accuracy (ACC), Normalized Mutual Information (NMI) and Adjusted Rand Index (ARI) (Xu & Tian (2015)). Given the set of ground-truth labels $L$ and the set of cluster assignments $C$ obtained from the model, these metrics are defined as follows.

#### H.2.1 ACCURACY (ACC)

Clustering Accuracy is defined as the maximum matching accuracy between cluster assignments and ground-truth labels. It is computed by finding the optimal one-to-one mapping between clusters and classes using the Hungarian algorithm.

$$\text{ACC} = \max_m \frac{1}{N} \sum_{i=1}^{N} \mathbb{I}(l_i = m(c_i))$$

where $N$ is the total number of samples, $l_i$ is the ground-truth label of sample $i$, $c_i$ is its assigned cluster label, $m$ is a permutation mapping function that maps each cluster index to a class index, and $\mathbb{I}(\cdot)$ is the indicator function.

#### H.2.2 NORMALIZED MUTUAL INFORMATION (NMI)

NMI quantifies the mutual dependence between the cluster assignments and the true labels, normalized by the entropy of each (Strehl & Ghosh (2002)). It measures the amount of statistical information shared by the two sets of distributions.

$$\text{NMI}(L, C) = \frac{2 \cdot I(L; C)}{H(L) + H(C)}$$

where $I(L; C)$ is the mutual information between $L$ and $C$, and $H(\cdot)$ is the entropy.

#### H.2.3 ADJUSTED RAND INDEX (ARI)

The ARI is a chance-adjusted version of the Rand Index (RI) (Hubert & Arabie (1985)). It compares the similarity between two data clusterings while accounting for the similarity that would occur by random chance. An ARI score of 1 indicates perfect clustering, and a score around 0 indicates random labeling.

Let $a$ be the number of pairs of samples that are in the same cluster in $C$ and in the same class in $L$ (TP), $b$ be the number of pairs in the same cluster in $C$ but not in the same class in $L$ (FP), $c$ be the number of pairs not in the same cluster in $C$ but in the same class in $L$ (FN), and $d$ be the number of pairs in different clusters and different classes (TN). The total number of pairs is $n = \binom{N}{2}$.

The Rand Index (RI) is:

$$\text{RI} = \frac{a + d}{a + b + c + d} = \frac{a + d}{n}$$

The Adjusted Rand Index is then computed as (Hubert & Arabie (1985)):

$$\text{ARI} = \frac{\text{RI} - \mathbb{E}[\text{RI}]}{\max(\text{RI}) - \mathbb{E}[\text{RI}]}$$

A common formulation based on the contingency table is:

$$\text{ARI} = \frac{\sum_{ij} \binom{n_{ij}}{2} - \frac{[\sum_i \binom{a_i}{2} \sum_j \binom{b_j}{2}]}{\binom{n}{2}}}{\frac{1}{2}[\sum_i \binom{a_i}{2} + \sum_j \binom{b_j}{2}] - \frac{[\sum_i \binom{a_i}{2} \sum_j \binom{b_j}{2}]}{\binom{n}{2}}}$$

where $n_{ij}$ is an entry in the contingency table (the number of samples common to cluster $i$ and class $j$), $a_i$ is the sum of the $i$-th row (size of cluster $i$), and $b_j$ is the sum of the $j$-th column (size of class $j$).

# I ADDITIONAL RESULTS

Table 3: The generative capabilities of $DL^2$,CMVAE and MMVAE+ on the CelebA-HQ (Gender) dataset. Bold and underline denote best and second-best results, respectively.

| Method | U FID ($\downarrow$) | C FID ($\downarrow$) | CC ($\uparrow$) |
|---|---|---|---|
| MMVAE+ | 55.58(1.11) | 58.67(1.32) | 0.41(0.03) |
| CMVAE | 54.25(1.58) | 60.58(3.32) | 0.38(0.04) |
| $DL^2$ | **53.50(1.52)** | **56.75(1.95)** | **0.45(0.00)** |

Abbreviations: U FID = Unconditional FID; C FID = Conditional FID; CC = Conditional Coherence; UCC = Unconditional Coherence.

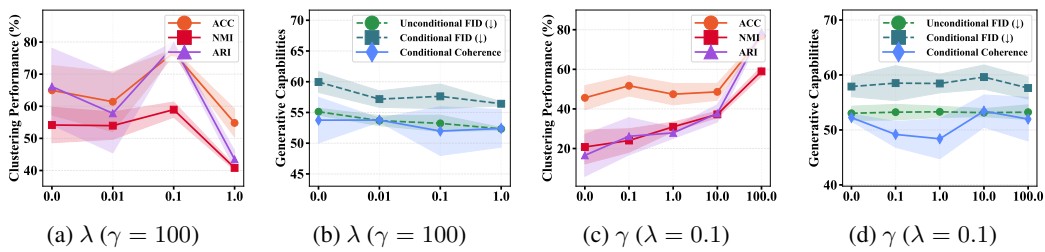

   (a) $\lambda$ ($\gamma = 100$)      (b) $\lambda$ ($\gamma = 100$)      (c) $\gamma$ ($\lambda = 0.1$)      (d) $\gamma$ ($\lambda = 0.1$)

Figure 12: The task-adaptive disentanglement performance of $DL^2$ on CelebA-HQ with varying values of $\lambda$ and $\gamma$.

## I.1 ANALYSIS AND EXPERIMENTAL VERIFICATION OF THE TASK-GENERATION DILEMMA AND THE EFFECT OF EXPLICITLY MODEL RESIDUAL SHARED INFORMATION

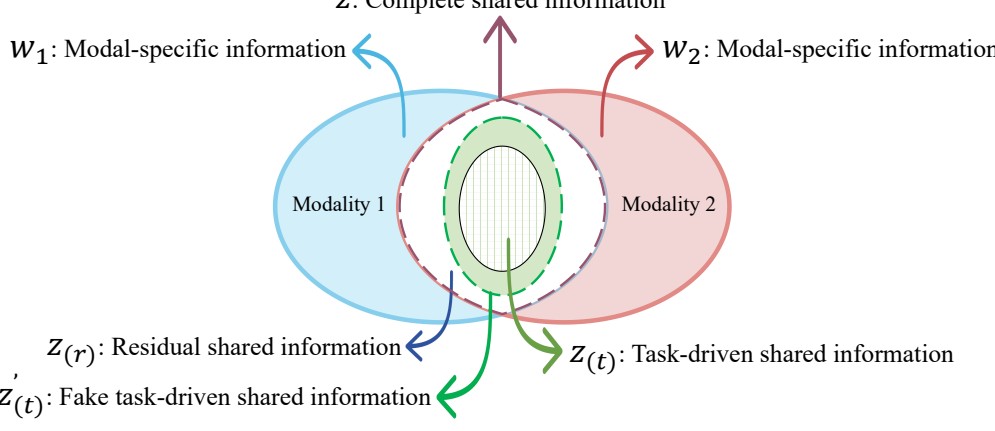

Figure 13: Schematic diagram of information structure. The blue and red ellipses represent two distinct modalities. The non-overlapping regions, filled with solid blue and red colors, correspond to modality-specific information ($w_1$ and $w_2$). Their overlapping area, outlined by a purple dashed line, denotes the complete shared information $z$. Within this shared region, an ellipse with a black border and green mesh filling represents the task-relevant shared information $z_{(t)}$. The remaining area (i.e., the complete shared information excluding the task-relevant part) constitutes the residual shared information $z_{(r)}$. Note that the task-relevant shared information is contained within a green dashed ellipse, which indicates the pseudo task-relevant shared information $z'_{(t)}$, as formally defined in Appendix I.1.

This section begins with a theoretical dissection of the intrinsic conflict in the task–generation dilemma within multimodal data. Subsequently, we validate this theoretical analysis through a controlled experiment, ultimately demonstrating that the dual-level disentanglement (DL$^2$) framework serves as a critical mechanism for resolving this fundamental conflict (or, **Dual-Level Disentan-gLement is necessary for the task-generation dilemma in multimodal data**).

We first introduce a variant of DL$^2$, referred to as DL, which performs only single-level disentanglement by reducing the dimensionality of $z_{(r)}$ to zero and setting the dimension of $z_{(t)}$ to the sum of the original $z_{(r)}$ and $z_{(t)}$ dimensions. Thus, we have $z = z_{(t)}$ in DL.

**Analysis Verification of the Task-Generation Dilemma** Figure 13 shows the information structure in a multimodal data with two modalities. The primary objective $\mathcal{L}$DL (Equation 2) encourages the separation of modality-specific information and shared information, enabling $\mathbf{z}$ to capture the complete shared information. The regularization terms $\mathcal{R}_{\text{PS}}$ (Equation 4) and $\mathcal{R}_{\text{NS}}$ (Equation 9) promote the disentanglement of task-relevant shared information from residual shared information, thereby guiding $z_{(t)}$ to encode primarily task-related semantics. In DL, since $z = z_{(t)}$, an inherent conflict arises: $\mathcal{L}_{\text{DL}}$ pushes $z$ to capture all shared information, while the contrastive losses constrain it to only task-relevant information. This leads to a suboptimal equilibrium where $z$ encodes neither complete shared information nor purely task-specific information, but an intermediate representation (denoted as "Fake task-driven shared information $z_{(t)}^{'}$" in Figure 13). Our analysis therefore supports the following inference: while DL improves clustering over unsupervised baselines such as CMVAE, it is expected to underperform relative to DL$^2$ and exhibit reduced generative consistency due to partial information loss.

**Experimental Verification of the Task-Generation Dilemma** We trained DL$^2$, CMVAE, and DL on the CelebA-HQ dataset using smiling intensity as the target label. Results in Figure 1 show that both DL$^2$ and DL achieve better clustering than CMVAE, confirming the benefit of weak supervision. However, DL$^2$ surpasses DL in clustering accuracy, indicating higher purity of task-relevant information in $z_{(t)}$. Moreover, DL exhibits significantly lower conditional and unconditional coherence compared to DL$^2$ and CMVAE, confirming the loss of shared information under single-level disentanglement. These results collectively indicate that DL's representation is neither sufficiently pure for task relevance nor complete for generation.

In summary, our theoretical and experimental results are consistent and mutually reinforcing, demonstrating that dual-level disentanglement is necessary to resolve the task–generation dilemma in multimodal data. This approach can be naturally extended to unimodal settings or general information purification tasks: directly regularizing the full encoder output may be suboptimal. Instead, explicitly modeling residual information provides a "reservoir" for irrelevant content, avoiding internal optimization conflicts and enabling both high purity and integrity in the representations.

I.2 A KEY IN RESOLVING THE TASK–GENERATION DILEMMA IS MAINTAINING GENERATIVE CAPABILITY WHILE ACHIEVING DISENTANGLEMENT.

A key in addressing the task–generation dilemma is to achieve disentanglement without compromising generative capability, or equivalently, to ensure minimal loss of overall information during task-adaptive disentanglement. We argue that simply incorporating weak supervision in a straightforward manner to identify task-relevant information may be inadequate to resolve this dilemma, as it can impair generative quality. To validate this claim, we conducted an ablation experiment that also addresses the question raised in Remark 2: why a reversed $\mathcal{R}_{\text{NS}}$ (or equivalently, CCMI) was not applied to positive signals to unify the framework.

$$\mathcal{R}_{\text{PS}}^*(z_{(t)}^k; z_{(t)}^l) = \mathcal{I}_{\text{CCMI}}(z_{(t)}^k; z_{(t)}^l) = \mathbb{E}_{p(z_{(t)}^k)p(z_{(t)}^l)} \left[ \log \frac{\mathbb{E}_{\boldsymbol{c}} \left[ p(z_{(t)}^k \mid \boldsymbol{c}) \cdot p(\boldsymbol{c} \mid z_{(t)}^l) \right]}{\mathbb{E}_{\boldsymbol{c}} \left[ p(z_{(t)}^k \mid \boldsymbol{c}) \right]} \right]. \quad (47)$$

Specifically, we replaced the proposed $\mathcal{R}_{\text{PS}}$ with $\mathcal{R}_{\text{PS}}^*$ (as defined in equation 47), resulting in a variant named DL$^2*$ that is trained with the loss function in equation 48. DL$^2*$ can be viewed as a purely weakly-supervised model that attracts positive sample pairs and repels negative pairs in the $\mathbf{z}_{(t)}$ space.

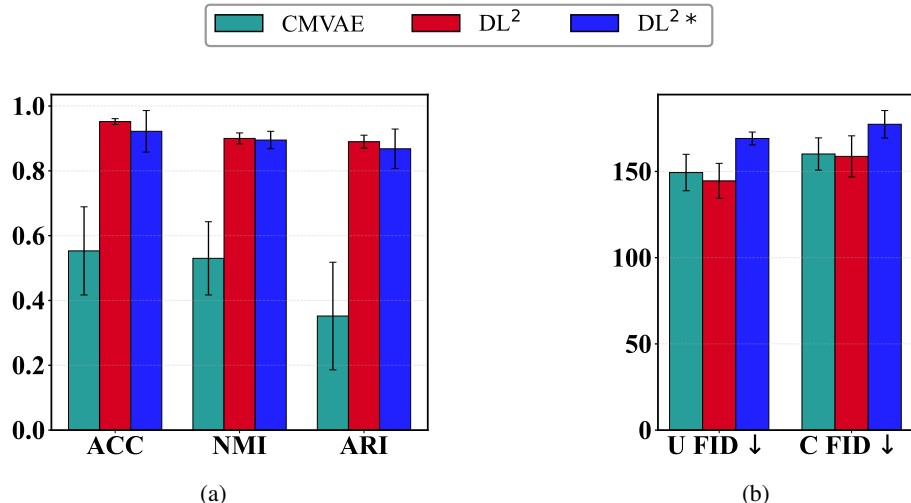

Figure 14: Experimental results of CMVAE, $DL^2$, $DL^2*$ on the CUBICC dataset demonstrate that while simple weakly-supervised techniques can enhance disentanglement performance, they often do so at the cost of generative quality. ***Note:*** For a fair comparison, we set the number of components in the prior distribution of CMVAE's $z$ to match the number of clusters, rather than using a large value as in the original implementation. Therefore, the results on CUBICC differ from those reported in the original paper.

We trained $DL^2$, $DL^2*$, and CMVAE on the CUBICC dataset; results are shown in Figure 14. Both $DL^2$ and $DL^2*$ substantially outperform CMVAE in clustering performance, confirming their effectiveness in leveraging contrastive signals. And $DL^2$ slightly surpasses $DL^2*$ in clustering accuracy, it demonstrates that our $\mathcal{R}_{PS}$ design, while less intuitive than $\mathcal{R}_{PS}^*$, is no less effective in identifying task-relevant information. Moreover, while $DL^2$ matches CMVAE in generative quality, $DL^2*$ exhibits significantly worse performance. This suggests that although simplistic weak supervision may enhance the purity of task-specific components, it can adversely affect generative performance. In contrast, $DL^2$ achieves the best clustering results without degrading generative quality.

$$\mathcal{L}_{DL^2*}(\boldsymbol{X}, \mathbb{M}, \mathbb{C}) = \mathcal{L}_{DL}(\boldsymbol{X}) + \frac{\lambda}{|\mathbb{M}|} \sum_{(\boldsymbol{X}^k, \boldsymbol{X}^l) \in \mathbb{M}} \mathcal{R}_{PS}^*(\boldsymbol{X}^k, \boldsymbol{X}^l) + \frac{\gamma}{|\mathbb{C}|} \sum_{(\boldsymbol{X}^i, \boldsymbol{X}^j) \in \mathbb{C}} \mathcal{R}_{NS}(\boldsymbol{X}^i, \boldsymbol{X}^j).$$

(48)

### I.3 QUALITATIVE COMPARISON OF GENERATED SAMPLES

To complement the quantitative metrics (FID and Coherence) presented in the main text, we report a qualitative comparison of generated samples from $DL^2$, CMVAE, and MMVAE+ on the DDM-NISTMM dataset. For a fair comparison, all three methods were evaluated using the same $\beta$ values on each dataset: $\beta = 2.5$ for DDMNIST, and $\beta = 1.0$ for both CUBICC and CelebA-HQ. The visual results as shown in Figure 15, Figure 16, Figure 17, Figure 18, Figure 19 and Figure 20.

We begin by examining the conditional generation outcomes on the DDMNISTMM dataset, as presented in Figure 15, Figure 16 and Figure 17. While the overall sample quality appears broadly comparable across all three methods, closer inspection reveals that digits generated by $DL^2$ and MMVAE+ exhibit noticeably sharper handwriting strokes and more consistent digit labels compared to those produced by CMVAE. We further analyze the unconditional generation results on the DDMNISTMM dataset, illustrated in Figure 18, Figure 19 and Figure 20. For this experiment, 100 samples were drawn from each component of the latent variable $z$ for both MMVAE+ and CM-VAE, and from each component of $z_{(t)}$ for $DL^2$. Given that the prior distributions of $z$ in CMVAE and $z_{(t)}$ in $DL^2$ are modeled as mixtures of 10 components, this sampling strategy yielded 1000

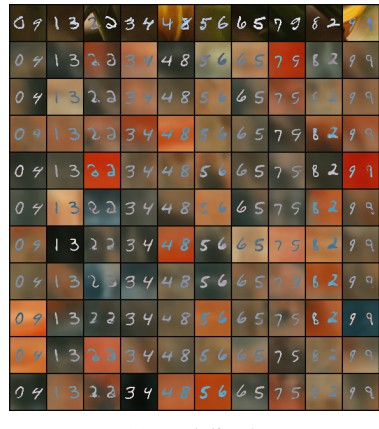

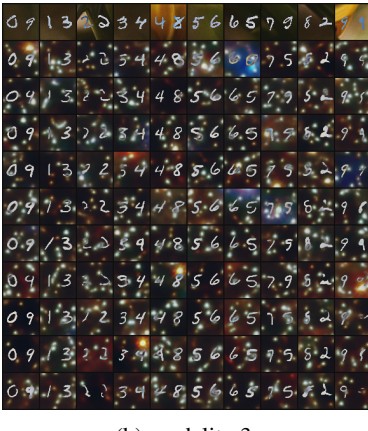

(a) modality 2                                          (b) modality 3

Figure 15: Conditional generation from the first modality to the remaining ones for DL$^2$ trained with $\beta = 2.5$ on the DDMNISTMM dataset. In each image, on the top row are starting samples, and below ten instances of conditional generation for the corresponding target modality. Qualitative results complement the analysis in Table 2.

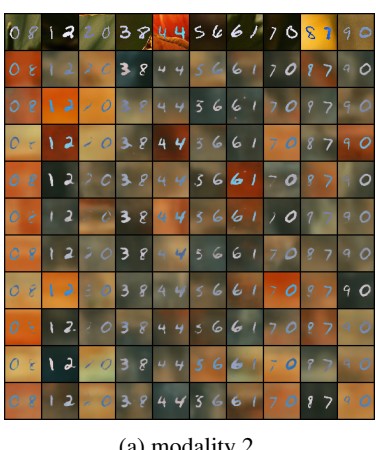

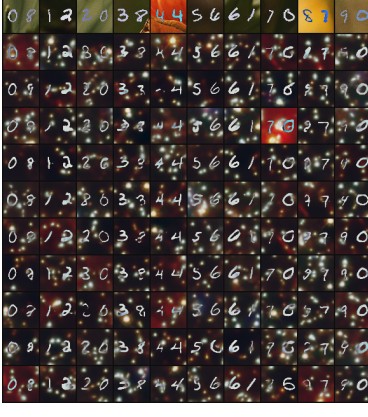

(a) modality 2                                          (b) modality 3

Figure 16: Conditional generation from the first modality to the remaining ones for CMBVAE trained with $\beta = 2.5$ on the DDMNISTMM dataset. In each image, on the top row are starting samples, and below ten instances of conditional generation for the corresponding target modality. Qualitative results complement the analysis in Table 2.

unconditional generated samples for each of these two methods. A clear visual hierarchy emerges: DL$^2$ produces samples of superior perceptual quality, outperforming MMVAE+ and substantially surpassing CMVAE. The latter not only demonstrates poorer visual effect but also generates digits with reduced sharpness and cross-modal consistency.

A particularly telling observation is the detectable correlation between modal backgrounds and specific components of $z$ in CMVAE's generated samples. For instance, in the first modality, certain components yield samples with predominantly black backgrounds, while others produce samples with yellow backgrounds. This phenomenon provides tangible evidence that CMVAE fails to adequately disentangle modality-specific information (e.g., background) from the shared information, resulting in the leakage of modal variations into the shared latent representation $z$.

This finding highlights a fundamental limitation common to MVAE-based clustering approaches. The framework often requires a predefined and potentially incomplete prior structure for the shared latent space $z$. In cases like DDMNISTMM, where prior knowledge might only extend to the num-

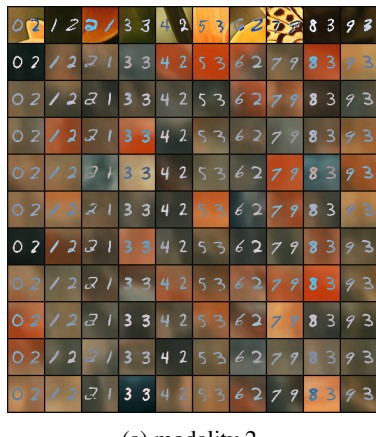

(a) modality 2

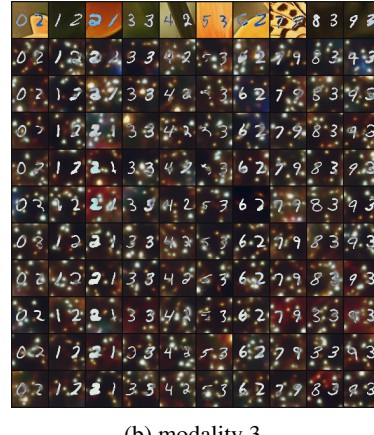

(b) modality 3

Figure 17: Conditional generation from the first modality to the remaining ones for MMVAE+ trained with $\beta = 2.5$ on the DDMNISTMM dataset. In each image, on the top row are starting samples, and below ten instances of conditional generation for the corresponding target modality. Qualitative results complement the analysis in Table 2.

ber of classes for the left-digit (the target clusters), but remain absent for other shared factors (e.g., the right-digit label), the model is forced to compress the entirety of the shared information into an inadequately structured space (e.g., a 10-component mixture). This overly restrictive prior can make it infeasible for the model to properly structure the entire shared latent space, consequently impairing its ability to disentangle shared and modality-specific factors, which in turn degrades generative performance. Furthermore, this aligns with the established understanding in disentanglement literature (Locatello et al. (2019)) that unsupervised identification and isolation of factors is challenging without appropriate inductive biases. When the shared latent space encompasses information beyond the target clustering structure, unsupervised multimodal clustering methods lack the necessary signals to identify the intended factors, leading to nearly random-level clustering performance for CMVAE on DDMNISTMM, as quantitatively confirmed in Table 2.

These qualitative analyses complement the quantitative findings in Table 2, demonstrating that $DL^2$ produces samples of comparable visual fidelity and semantic clarity. Crucially, they confirm that its superior disentanglement performance does not come at the cost of degraded generative capability—that is, $DL^2$ achieves more disentangled representations without compromising the ability to generate high-quality data, thereby providing visual and explanatory support for the efficacy of the proposed task-adaptive disentanglement paradigm and the $DL^2$ framework.

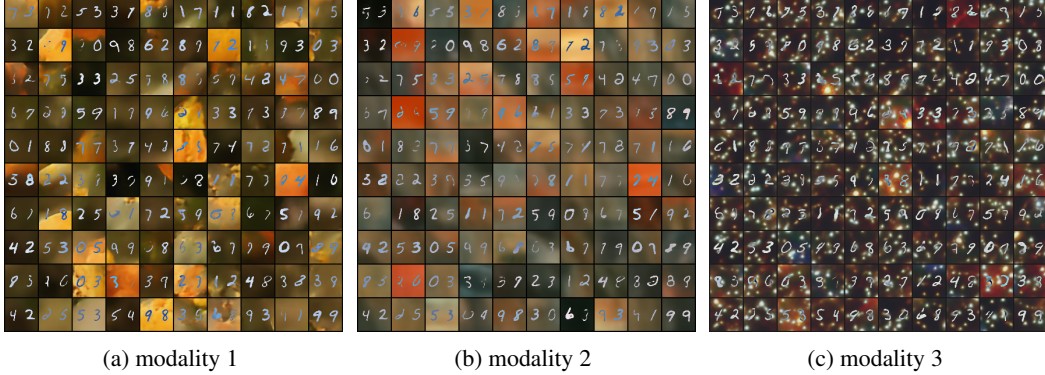

(a) modality 1    (b) modality 2    (c) modality 3

Figure 18: Unconditional generation for MMVAE+ traind with $\beta = 2.5$ on the DDMNISTMM dataset. A hundred instances across modalities are shown. Qualitative results complement the analysis in Table 2.

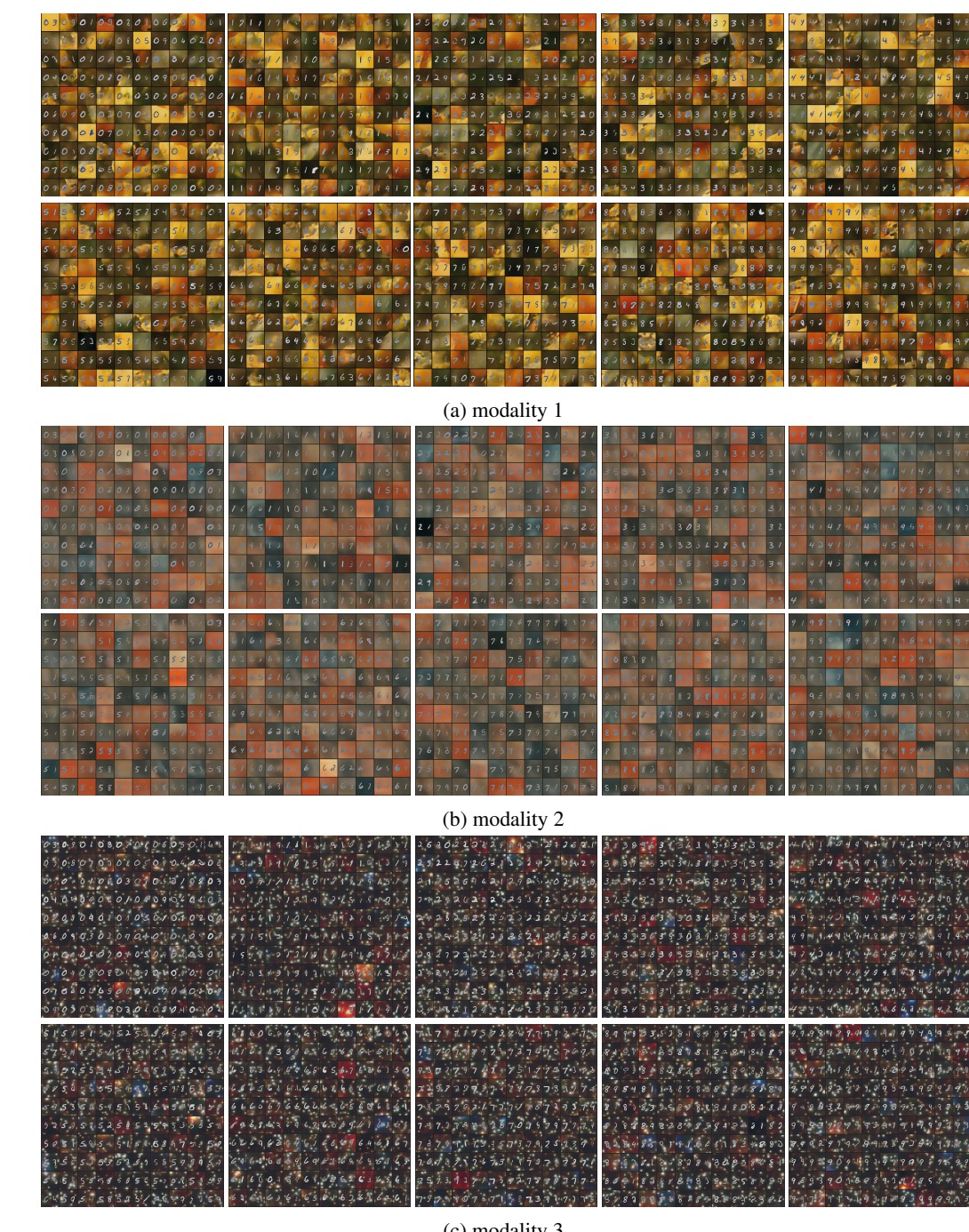

(a) modality 1

(b) modality 2

(c) modality 3

Figure 19: Unconditional generation for DL$^2$ trained with $\beta = 2.5$ on the DDMNISTMM dataset. A thousand instances across modalities are shown. Qualitative results complement the analysis in Table 2.

## I.4 LATENT SPACE INTERPOLATION AND SEMANTIC TRAVERSALS

To more intuitively understand what information is encoded in the task-driven encoding $z_{(t)}$, the residual encoding $z_{(r)}$, and the modality-private encoding $\mathbf{w}$, we performed latent space interpolation and generation experiments on DDMNIST. The results are shown in Figure 21, Figure 22,

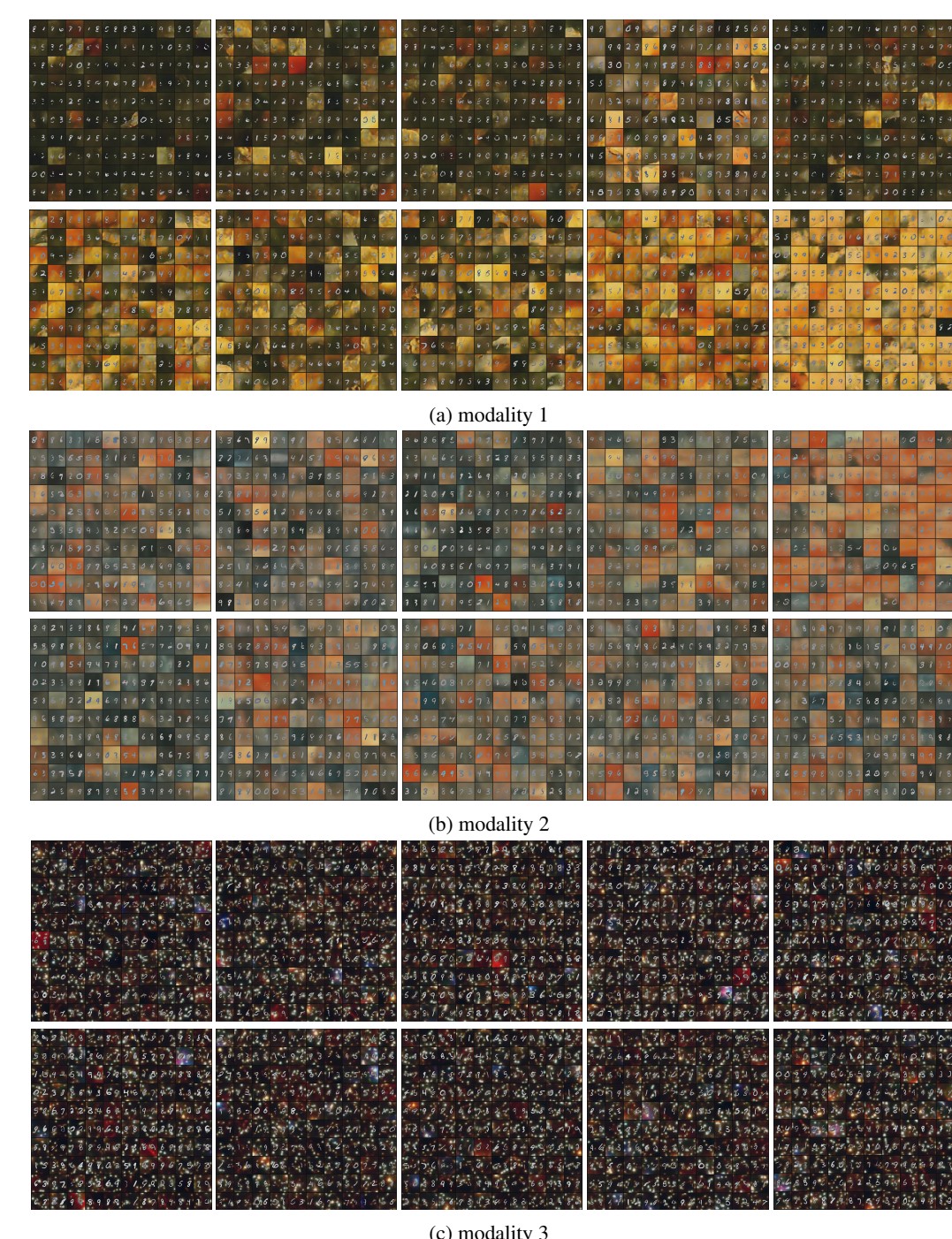

(a) modality 1

(b) modality 2

(c) modality 3

Figure 20: Unconditional generation for CMVAE traind with $\beta = 2.5$ on the DDMNISTMM dataset. A thousand instances across modalities are shown. Qualitative results complement the analysis in Table 2.

Figure 23 and Figure 24. Each semantic traversal grid displays a 10×10 image matrix. The first and last rows of each grid use $z_{(t)}$, $z_{(r)}$, and $w$ sampled from the prior distribution, while the middle eight rows are generated by linearly interpolating a specified latent code while keeping the others fixed. All latent variables remain unchanged in the corresponding columns of the grid, except for the latent encoding being traversed.

We observe that when interpolating the modality-private encoding $\mathbf{w}$, only the background and handwriting style change across the three modalities, while the central digit label remains almost unchanged (Figure 21). When interpolating the task-driven encoding $\mathbf{z}_{(t)}$, the background and the right digit remain largely consistent, with only the left digit varying gradually (Figure 22). Similarly, interpolating the residual encoding $\mathbf{z}_{(r)}$ leaves the background and left digit mostly unchanged, with smooth variation only in the right digit (Figure 23). Finally, Figure 24 shows the semantic traversal when interpolating the entire $\mathbf{z} = (\mathbf{z}_{(t)}, \mathbf{z}_{(r)})$, where both digits change simultaneously while the background remains stable.

Thus, by interpolating different components of the latent space and analyzing the resulting semantic traversals, we conclude that: the modality-private encoding $\mathbf{w}$ captures modality-specific information; the task-driven encoding $\mathbf{z}_{(t)}$ represents the left digit label (the target indicated by the CSs); and the residual encoding $\mathbf{z}_{(r)}$ adaptively encodes the right digit label, enabling $\mathbf{z} = (\mathbf{z}_{(t)}, \mathbf{z}_{(r)})$ to fully capture the shared information. These results demonstrate that DL$^2$ successfully achieves task-adaptive disentanglement.

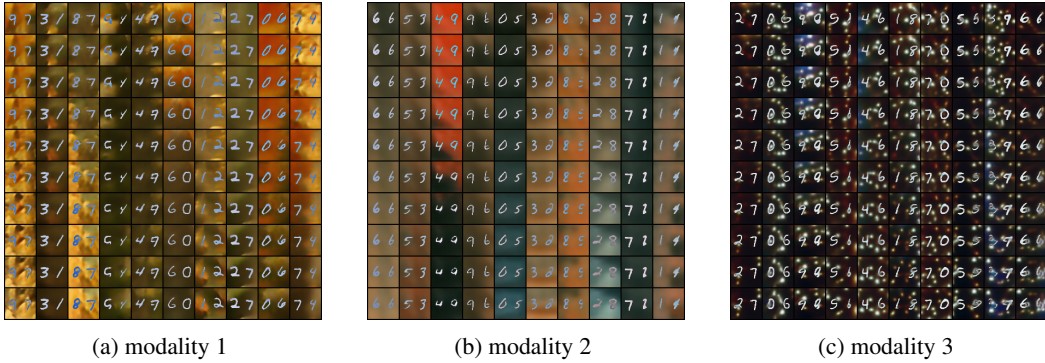

(a) modality 1          (b) modality 2          (c) modality 3

Figure 21: The semantic traversal images generated by interpolation in the latent space of $\mathbf{w}$.

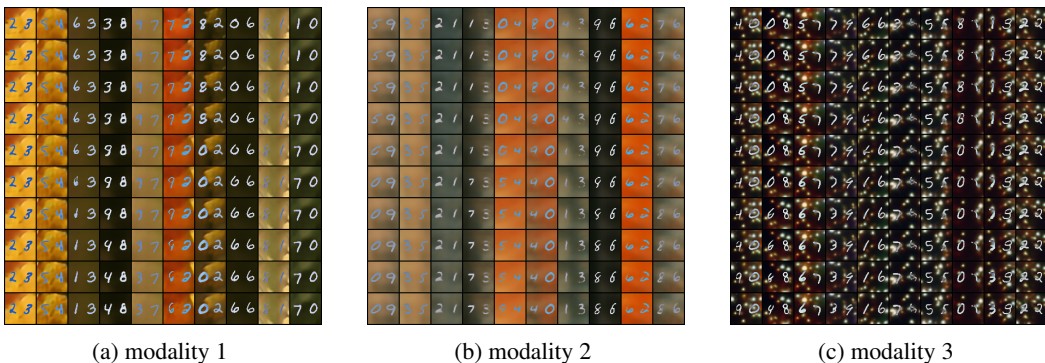

(a) modality 1          (b) modality 2          (c) modality 3

Figure 22: The semantic traversal images generated by interpolation in the latent space of $\mathbf{z}_{(t)}$.

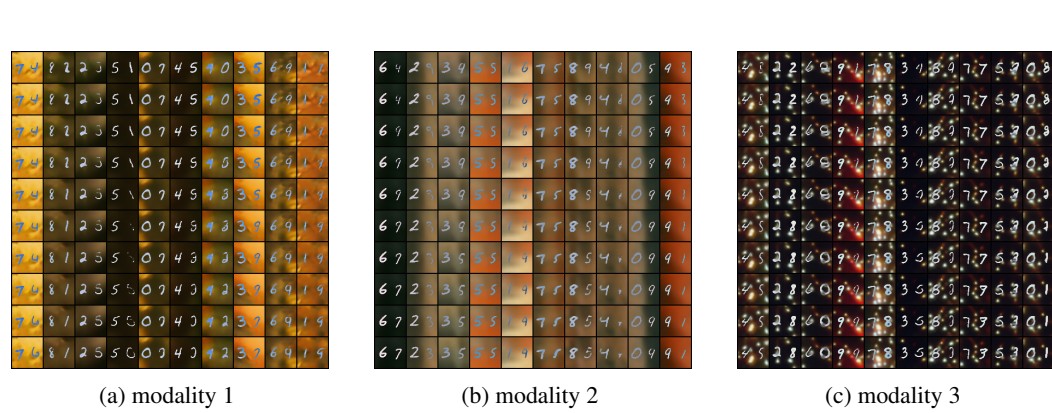

(a) modality 1              (b) modality 2              (c) modality 3

Figure 23: The semantic traversal images generated by interpolation in the latent space of $z_{(r)}$.

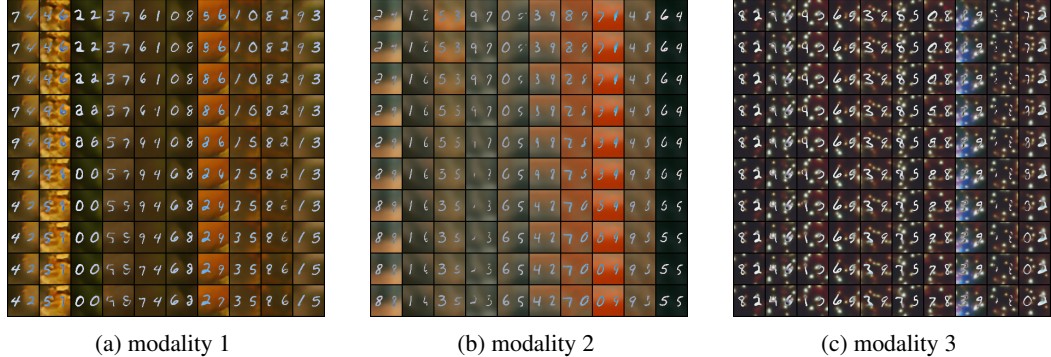

(a) modality 1              (b) modality 2              (c) modality 3

Figure 24: The semantic traversal image generated by interpolation in the latent space of $z$.

