# OpenReview forum: "Dual-Level DisentangLement ($\text{DL}^2$): Task-Adaptive Disentanglement for Resolving the Task-Generation Dilemma"
_ICLR.cc/2026/Conference — ICLR 2026 Conference Withdrawn Submission_

### Official Review · Reviewer_yCKW · 2025-10-29

**Soundness:** 2
**Presentation:** 1
**Contribution:** 2
**Rating:** 2
**Confidence:** 3

**Summary:**

This paper addresses some discrepancies between discriminative and generative model performance in multi-modal latent variable models. Under a specific assumption of tasks responding to data clusters arising from the shared information in multi-modal data, they argue that the latent spaces of generative models perform poorly on discriminative tasks and that (predominantly discussing VAE-based methods) disentanglement methods fail at generation. Authors propose a two-level disentanglement framework that first splits the representation into shared and modality-specific components via distinct encoders and then uses contrastive losses ensuring the clustering of a subset of the shared latents to align with the provided labels (with the remaining latents ensuring completeness). The method achieves strong performance on clustering and reconstruction tasks compared to benchmarks.

**Strengths:**

1) The paper has a good problem formulation, and effectively identifies and articulates the "task-generation dilemma", providing a reasonable motivation for developing models that can handle both discriminative and generative objectives well.

2) The paper also has good experimental methodology. The authors perform experiments on several datasets. They include ablation studies, analyses of hyperparameter sensitivity, and the impact of supervision quantity, adding robustness to the empirical validation.

**Weaknesses:**

1) The paper claims significant novelty but primarily combines existing concepts (MVAE structure, contrastive learning) with specific, newly formulated regularizers. Crucially, it fails to discuss or compare against a large body of highly relevant prior work on supervised/weakly-supervised disentanglement in VAEs and multimodal learning. Examples include:

a) GMM priors: Methods using Gaussian Mixture Model priors in the latent space for clustering/disentanglement (e.g., GMVAE https://arxiv.org/abs/1611.02648 and multiDGD https://www.nature.com/articles/s41467-024-53340-z) are directly relevant given the clustering task and generative assumptions (and multiDGD is even a multi-modal model, but without modality-specific spaces).

b) Adversarial disentanglement: Several works use adversarial methods for disentangling shared/specific or other factors in multimodal contexts (e.g., https://arxiv.org/abs/2506.13658, https://pmc.ncbi.nlm.nih.gov/articles/PMC7773223/ ), offering an alternative approach not discussed

c) Hierarchical/structural VAEs: Methods like treeVAE (https://arxiv.org/abs/2306.08984) explore structured disentanglement, relevant to the idea of decomposing the shared space.

d) Other contrastive disentanglement approaches: Various works combine contrastive learning with VAEs/GANs for disentanglement (e.g., https://arxiv.org/pdf/2203.11284 , https://proceedings.neurips.cc/paper_files/paper/2023/hash/d5470483dd38f71f7bd9e68ce1b94145-Abstract-Conference.html , https://proceedings.mlr.press/v240/tu24a.html , https://openaccess.thecvf.com/content/WACV2023/papers/Mo_Representation_Disentanglement_in_Generative_Models_With_Contrastive_Learning_WACV_2023_paper.pdf ), which should really be situated relative to the contrastive approach used in DL².

Although none of these methods provide the exact combination of multimodal spared and private spaces with disentanglement in the shared space, without proper comparison to these related areas, the claims of SOTA performance are unreliable.

2) The core TADL mechanism relies entirely on task-specific contrastive signals (CSs). This inherently limits the model's applicability to scenarios where such supervision is readily available and only increases a models’ versatility for clustering and generation wrt the pretrained task. The paper should acknowledge this limitation more directly.

3) Strong and specific generative assumptions: The described generative process (Sec 2.1) assumes a specific hierarchical structure (cluster -> z_{(t)}, independent z_{(r)}, independent w_m) and distributional forms. This makes the derived ELBO and regularizers potentially sensitive to violations of these assumptions, limiting the method's robustness to diverse real-world data structures. The paper also does not acknowledge that this generative process is a very strong assumption, but rather states it as known fact of multi-modal data generation.

**Questions:**

Can the authors further clarify the generative assumption (i.e. the hierarchical structure) and explain how realistic these are in real-world scenarios.

---

### Official Review · Reviewer_2tKB · 2025-11-01

**Soundness:** 2
**Presentation:** 3
**Contribution:** 3
**Rating:** 6
**Confidence:** 3

**Summary:**

The article addresses what it calls the task–generation dilemma in multimodal learning, *i.e.*, the trade-off between achieving discriminative task performance (requiring pure task features) and maintaining generative fidelity (requiring complete shared information).
To resolve this, the authors propose a new paradigm, named Task-Adaptive Disentanglement (TADL), and instantiate it with DL^2 (Dual-Level Disentanglement) built on an MVAE.
The results show that DL^2 reports good clustering metrics on DDMNISTMM (semi-synthetic), CUBICC (image–text birds), and CelebA-HQ (image–text), while matching or slightly improving the generative quality against selected competitors.

**Strengths:**

The article is well-written and structured.
The proposed method is described in sufficient detail.

The DL^2 model extends the MVAE with rigorously formulated probabilistic components, including explicit variational bounds (Eqs. 2, 4, 9).
Furthermore, the dual-level decomposition provides a clean factorization between the relevant task and the residual information.

The Common-Cause Mutual Information (CCMI) metric is conceptually new and offers a new perspective for modeling negative pair dependencies.

The authors compare DL^2 with a comprehensive suite of baselines (VaDE, DC-GMM, SDEC, C-IDEC, VolMaxDCC, CMVAE, MMVAE+) that cover both discriminative metrics (ACC, NMI, ARI) and generative metrics (FID, coherence).

The Appendix contains detailed proofs, dataset descriptions, and qualitative analyzes.

**Weaknesses:**

The mutual independence assumption between $z^{(t)}$ and $z^{(r)}$ may be too strong, as dependencies between residual and private features often persist in real multimodal data, and no empirical verification (*e.g.*, independence tests) is provided.

With respect to experimental results, recent multimodal SOTA disentanglement methods such as MoPoE-VAE (Sutter et al., 2021) and MVTCAE (Hwang et al., 2021) are discussed in the state-of-the-art, but not compared from an experimental point of view.

Furthermore, the authors omit potential competitive representation-learning baselines that are highly relevant for modern image–text clustering, such as CLIP/OpenCLIP features with simple clustering heads (*e.g.*, k-means/constrained k-means or HDBSCAN).

Contrastive signal construction uses “$N$ unique CSs” derived from labels, but the sampling protocol (*e.g.*, how pairs are chosen, noise levels) is not specified in the main text.

**Questions:**

The authors should:
1) Add experiments where the “residual” factors are correlated with the task factor to test whether $z^{(t)}$ truly protects generation without leaking discriminative signals back into $z^{(t)}$. Report how $R_{PS}$ and $R_{NS}$ behave under such confounding.
2) Add competitive multimodal representation baselines such as CLIP/OpenCLIP + k-means/constrained k-means or HDBSCAN. This will better characterize whether DL^2’s gains arise from disentanglement per se or could be matched by strong off‑the‑shelf features.
3) For unimodal baselines, report both modalities (or the mean across them) rather than “best‑modal”, and justify deviations. For CMVAE, at least explain the prior choice change across datasets; where possible, run both the original and modified settings to show sensitivity.

---

### Official Review · Reviewer_Y6VT · 2025-11-01

**Soundness:** 3
**Presentation:** 2
**Contribution:** 2
**Rating:** 2
**Confidence:** 5

**Summary:**

This paper first identifies a task-generation dilemma that trade-off may exist in a single model. Accordingly, this paper designs a task-adaptive disentanglement (TADL) paradigm that dynamically disentangles representations guided by task-specific supervised signals with the dual-level disentanglement (DL2) framework that leverages contrastive signals as a practical and efficient form of weak supervision. Several experiments are carried out based on multimodal clustering tasks.

**Strengths:**

1. this paper is clearly written and well-structured.

2. the motivation and clear, also the technical design connects closely to the motivation.

**Weaknesses:**

1. The paper is not very carefully organized. For example, the paragraph at line 62 and line 71 has the exact same beginning sentences, "Resolving the task-generation dilemma requires disentangling task-relevant information. Although
disentangled representation learning (DRL) provides a framework for factorizing data, its mainstream paradigms are unsuitable for this challenge". This obvious mistake shows the incompleteness and unorganized structure of this work.

2. The problem setting is unclear. First of all, what is the definition of the shared cross-modal information? This seems to rely on a clear causal structure, where each modality is generated from some shared variables and modality-specific variables. But such definitions are not formulated clearly in this paper. Figure 2 directly addresses the shared representations, without even explaining how such representations come from the original multi-modal data.

3. The analysis of related works mainly involves unsupervised disentanglement tasks. While task-relevant discrimitive representation learning is itself a supervised process.

4. More baselines could be included in the experiments.

5. Overall, this paper uses too many abbreviations, which may make it difficult to understand.

**Questions:**

Please refer to the weakness.

---

### Official Review · Reviewer_xA93 · 2025-11-03

**Soundness:** 2
**Presentation:** 2
**Contribution:** 3
**Rating:** 4
**Confidence:** 3

**Summary:**

This paper introduces Dual-Level Disentanglement (DL2), a multimodal variational framework designed to address competition between the task objective and generation quality: discriminative tasks benefit from compact, task-relevant representations, whereas generative tasks require richer, shared information. DL2 extends multimodal VAEs with two hierarchical latent decompositions. The first level aims at separating modality specific information from shared one, while the second level adaptively splits the shared representation into a task-specific component and a residual subspace.
Weak supervision is introduced through contrastive signals on positive and negative pairs, combined with a Common-Cause Mutual Information (CCMI) regularizer that encourages disentanglement of unrelated factors.
Experiments on a proposed synthetic dataset variant of MNIST (named  DDMNISTMM), CUBICC, and CelebA-HQ report improved clustering and competitive generative metrics, supporting the idea that adaptive decomposition can balance discriminative purity and generative completeness.

**Strengths:**

* **Clear motivation and problem framing**
  The paper addresses the interesting and less explored compatibility between discriminative and generative objectives in multimodal VAEs. The task–generation dilemma is well articulated and aligns with recent discussions on shared-factor learning and information bottlenecks in representation learning (Palumbo et al., 2024; Wu et al 2025).

* **Empirically principled model design**
  The two-level decomposition is logically motivated: Level-1 mirrors multi-level VAEs (Bouchacourt et al., 2018; Sutter et al., 2020), while Level-2 introduces task-aware factorization, conceptually extending adaptive disentanglement frameworks (Locatello et al., 2020).

* **Empirical strength and robustness trends**
  The model consistently outperforms CMVAE and MMVAE+ on clustering while maintaining comparable FID scores on the benchmarks tested. Sensitivity studies on regularization parameters and supervision density show predictable behavior, supporting stability under good quality moderate supervision.

**Weaknesses:**

* **Lack of identifiability**. Identifiability guarantees of the latent variables are not discussed, together with assumptions on the data generating process (DGP). Under suitable assumptions on the DGP the method it would be interesting to check whether the method can be proved identifiable, possibly with a similar proof strategy of (Locatello 2020). This would support the necessity of the dual-level structure from the theoretical perspective.
From my understanding the model assumes mutual independence among latent components (task, residual, and modality-specific), which is rarely satisfied in realistic multimodal data with nonlinear cross-modal correlations (Hwang et al., 2021; Wu & Goodman, 2018). This may limit identifiability and robustness.

* **Experiments limited to semisynthetic settings** Experiments are limited to one semi-synthetic and two image–text datasets, which restricts claims about scalability and generality. Tests on larger and more diverse corpora (e.g., MS-COCO, AudioSet) would strengthen the empirical case. Demonstrating DL2 on larger or more diverse modalities (audio–video, text–audio, multimodal scientific datasets) would illustrate its flexibility beyond controlled settings.

* **CCMI regularizer** The Common-Cause Mutual Information regularizer looks only intuitively motivated to me. Its connection to standard mutual information estimators such as InfoNCE (Chen et al., 2020) or JS-MI (Poole et al., 2019) is not discussed, and considerations on convergence guarantees are lacking. See questions for more details.

* **Additional related work** There are a few unmentioned papers that would benefit from discussion. For example: [a] proposes an information‐theoretic Minimum Necessary Information criterion for controlled multimodal disentanglement; [b]  enforces sparse sufficiency and feature sharing across tasks with identifiability guarantees and strong results in real-world multi domain settings; [c] shows that just a few labeled examples suffice to steer unsupervised models toward disentanglement under mild inductive biases.


_[a] Wang et al (2024). An Information Criterion for Controlled Disentanglement of Multimodal Data. ICLR 2024._

_[b] Fumero et al. (2023). Leveraging Sparse and Shared Feature Activations for Disentangled Representation Learning. NeurIPS 2023._

_[c] Locatello et al. (2020). Disentangling Factors of Variation Using Few Labels. ICLR 2020_


**Lack of limitations**: the paper doesn't have a limitations section or paragraph. I believe highlighting the limitations of the work would help understanding better the paper and putting it into context.

**Questions:**

* Could the authors describe the estimator used for CCMI, i.e. how it is computed in practice, and clarify its relation with known divergences like InfoNCE or JS-MI as well with practical objectives such as the one employed in (Locatello 2020)?

* Under which assumptions on the latent generative process is the task-specific component identifiable up to permutation or scaling?

* Have the authors measured mutual information between the task and residual subspaces to verify the assumed independence? Some qualitative or traversal analyses could clarify whether the residual captures meaningful semantics or noise.

* What is the rationale behind introducing the DDMNISTMM as opposed as using e.g. PolyMNIST as a synthetic benchmark, which is already employed by the community ?

* How would the method perform in case of noisy data pairings (e.g. by corrupting a few pairs)? It seems from Fig. 6 that the method's success depends on the quality of weakly supervised pairs. The observed degradation under reduced supervision (Fig. 6) may suggest sensitivity to label consistency, consistent with prior work on partially supervised disentanglement (Locatello et al., 2020).

* Reporting considerations about parameter counts, runtime per epoch, and GPU memory use would clarify computational efficiency relative to CMVAE and MMVAE+.

* Reported metrics (ACC, NMI, ARI, FID, Coherence) assess clustering and generation but not disentanglement directly. Could standard disentanglement metrics such as MIG, DCI, or SAP (Chen et al., 2018; Eastwood & Williams, 2018) be employed?

---

### Note · Authors · 2025-11-21

I have read and agree with the venue's withdrawal policy on behalf of myself and my co-authors.